# Impact of temperature and water availability on microwave-derived gross primary production

Irene Eva Teubner[1,2], Matthias Forkel[3], Benjamin Wild[1], Leander Mösinger[1], and Wouter Arnoud Dorigo[1]

[1]Department of Geodesy and Geoinformation, TU Wien, Wiedner Hauptstraße 8, 1040 Vienna, Austria
[2]Zentralanstalt für Meteorologie und Geodynamik (ZAMG), Hohe Warte 38, 1190 Vienna, Austria
[3]Environmental Remote Sensing Group, Institute of Photogrammetry and Remote Sensing, Technische Universität Dresden, Helmholtzstraße 10, 01069 Dresden, Germany

**Correspondence:** Irene E. Teubner (irene.teubner@zamg.ac.at)

**Abstract.** Vegetation optical depth (VOD) from microwave satellite observations has received much attention in global vegetation studies in recent years due to its relationship to vegetation water content and biomass. We recently have shown that VOD is related to plant productivity, i.e. gross primary production (GPP). Based on this relationship between VOD and GPP we developed a theory-based machine learning model to estimate global patterns of GPP from passive microwave VOD retrievals. The VOD-GPP model generally showed good agreement with site observations and other global data sets in temporal dynamic but tended to overestimate annual GPP across all latitudes. We hypothesized that the reason for the overestimation is the missing effect of temperature on autotrophic respiration in the theory-based machine learning model. Here we aim to further assess and enhance the robustness of the VOD-GPP model by including the effect of temperature on autotrophic respiration within the machine learning approach and by assessing the interannual variability of the model results with respect to water availability. We used X-band VOD from the VOD Climate Archive (VODCA) data set for estimating GPP and used global state-of-the art GPP data sets from FLUXCOM and MODIS to assess residuals of the VOD-GPP model with respect to drought conditions as quantified by the Standardized Precipitation and Evaporation Index (SPEI).

Our results reveal an improvement in model performance for correlation when including the temperature dependency of autotrophic respiration (average correlation increase of 0.18). This improvement in temporal dynamic is larger for temperate and cold regions than for the tropics. For ubRMSE and bias, the results are regionally diverse and are compensated in the global average. Improvements are observed in temperate and cold regions while decreases in performance are obtained mainly in the tropics. The overall improvement when adding temperature was less than expected and thus may only partly explain previously observed differences between the global GPP datasets. On interannual time scales, estimates of the VOD-GPP model agree well with GPP from FLUXCOM and MODIS. We further find that the residuals between VOD-based GPP estimates and the other data sets do not significantly correlate with SPEI which demonstrates that the VOD-GPP model can capture responses of GPP to water availability even without including additional information on precipitation, soil moisture or evapotranspiration. Exceptions from this rule were found in some regions: significant negative correlations between VOD-GPP residuals and SPEI were observed in the US corn belt, Argentina, Eastern Europe, Russia and China, while significant positive correlations were

obtained in South America, Africa and Australia. In these regions, the significant correlations may indicate different plant strategies for dealing with variations in water availability.

Overall, our findings support the robustness of global microwave-derived estimates of gross primary production for large-scale studies on climate-vegetation interactions.

*Copyright statement.* TEXT

## 1 Introduction

Vegetation optical depth (VOD) from microwave satellite observations provides the opportunity for studying large-scale vegetation dynamics due to its sensitivity to the vegetation water content and above-ground biomass. Different studies have employed VOD for deriving various plant properties or vegetation characteristics that can be related to the plant's water content, including biomass estimation (Liu et al., 2015; Brandt et al., 2018; Rodríguez-Fernández et al., 2018; Chaparro et al., 2019; Fan et al., 2019; Frappart et al., 2020; Wigneron et al., 2020; Li et al., 2021), crop yield (Chaparro et al., 2018), tree mortality (Rao et al., 2019; Sapes et al., 2019), analysis of burned area (Forkel et al., 2019), ecosystem-scale isohydricity (Konings and Gentine, 2017), plant water uptake during dry downs (Feldman et al., 2018) and plant water storage (Tian et al., 2018). VOD, or microwave satellite observations in general, are also analyzed for its potential in detecting the impact of drought (Song et al., 2019; Crocetti et al., 2020). Despite the sensitivity of VOD to vegetation water content, the relationship between VOD and GPP has not yet been analyzed with regard to how the relationship responds to varying conditions of dry- or wetness.

Recently, we have shown that VOD is related to plant productivity, i.e. gross primary production (GPP) (Teubner et al., 2018). Based on these findings, we developed a theory-guided machine learning model to estimate GPP from VOD (VOD-GPP model) and trained the model using eddy covariance estimates of GPP from the FLUXNET network (Teubner et al., 2019). The VOD-GPP model relies on estimating carbon sink terms, i.e. net primary production (NPP) and autotrophic respiration (Ra), based on VOD as a proxy for above-ground living biomass. The VOD-GPP model thus represents a carbon sink-driven approach. Since the VOD-GPP model uses biomass as main input, the estimation of GPP does not rely on input variables that are commonly used in source-driven approaches, e.g. absorption of photosynthetically active radiation as primary input term or vapor pressure deficit as controlling factor for stomatal conductance (Running et al., 2000; Turner et al., 2005; Goodrich et al., 2015; Zhang et al., 2016, 2017). Although different studies are tackling the question of how much information on biomass is actually contained in the VOD signal (Momen et al., 2017; Vreugdenhil et al., 2018; Zhang et al., 2019), it might be worth noting that the water content can be seen as an important aspect in our model approach since it presents the living part of the vegetation and only living cells, which contain water, are able to respire. We have shown that the VOD-GPP model can well represent temporal dynamics of GPP but that it overestimates GPP especially in temperate and boreal regions (Teubner et al., 2019). We hypothesize that this overestimation may be caused by a missing representation of temperature dependency of autotrophic respiration in the VOD-GPP model.

Ra is the process through which chemical energy that was stored by building up carbohydrates during photosynthesis is gained by converting carbohydrates back into carbon dioxide. It is generally known that Ra is a temperature-dependent process (e.g., Atkin and Tjoelker, 2003). Modelling the response of Ra to temperature, however, is complex due to the existence of thermal acclimation (Atkin and Tjoelker, 2003). Ra is commonly represented through an exponential function with Q10 as base which is multiplied with a basal respiration rate (e.g., Smith and Dukes, 2013). The base value Q10 describes how much

Ra changes when temperature changes by 10°C (e.g., Atkin et al., 2008). Although global models often use constant values for either one parameter or both parameters (Gifford, 2003; Smith and Dukes, 2013), studies have shown that both basal respiration rate and Q10 may vary with temperature (Tjoelker et al., 2001; Wythers et al., 2013). The implementation of such temperature acclimation yields a functional representation that decreases again at higher temperatures and thus takes into account that respiration may decrease outside an optimum temperature range (Smith and Dukes, 2013).

Here we aim to assess the impact of the temperature dependency of Ra in the VOD-GPP model and if it can improve model performance. Furthermore, we will test the plausibility of the model by comparing the estimated interannual variability of GPP with independent state-of-the art global data sets of GPP and by assessing model residuals with respect to variations in climatological water availability as represented by the Standardized Precipitation and Evaporation Index (SPEI). Since source- (GPP) and sink-terms (NPP + Ra) should theoretically be in balance, any differences between the two approaches that are

related to variations in water availability may give insight into different plant strategies for dealing with dry or wet conditions and thus may be of interest for ecological or plant-physiological studies at large-scale.

## 2 Data and methods

### 2.1 Choice of microwave frequency

The VOD-GPP model relies on biomass as input. Nevertheless, the choice of microwave frequency for estimating GPP may

look counterintuitive. On the one hand, VOD from low microwave frequencies like L-band has been demonstrated to be better suited as proxy for mapping total above-ground biomass than high frequency VOD, i.e. X-band VOD, as L-band VOD saturates less at high biomass values (Chaparro et al., 2019; Frappart et al., 2020; Li et al., 2021). On the other hand, previous analyses demonstrated that X-band VOD shows a closer agreement with GPP (Teubner et al., 2018, 2019; Kumar et al., 2020). In Figure A1 we further corroborated this observation by a correlation analysis between in situ GPP and VOD from L- and X-

band, respectively (for details about the single sensor VOD datasets, see Teubner et al., 2018). Despite the high fraction (38%) of forest pixels used for this computation, higher correlations were obtained for X-band than for L-band. An explanation could be that whole plant biomass was found to be less suited for estimating GPP as opposed to biomass of metabolically active plant parts like leaves and fine roots (Litton et al., 2007). Based on these findings, we concluded that higher frequency VOD appears to be better suited for estimating GPP and therefore we used X-band VOD in our analysis.

## 2.2 Data sets

We analyzed different GPP data sets derived from microwave and optical sensors as well as SPEI. As input to the VOD-GPP model, we used X-band VOD data from the VOD Climate Archive (VODCA). Since global coverage for VODCA X-band data starts in 2003 (Moesinger et al., 2020) and SPEI data are available through 2015, we used the common period from 2003 to 2015 for our analysis. Temporal median maps for the global GPP data sets are displayed in the supplement (Figure A2).

### 2.2.1 VODCA

VOD retrievals from single sensors often span only a certain period in time, which may hamper the analysis of longer periods. To overcome this problem, we used a merged single frequency VOD from the VOD Climate Archive (VODCA; Moesinger et al., 2020) as input to our model. VODCA (Moesinger et al., 2020) X-band (VODCAX) contains nighttime observations of passive VOD derived from TMI (10.7 GHz; variable overpass time), AMSR-E (10.7 GHz; descending 1:30 am), WindSat (10.7 GHz; descending 6:00 am) and AMSR2 (10.7 GHz; descending 1:30 am). The VOD input data are obtained from the Land Parameter Retrieval Model (LPRM; van der Schalie et al., 2017). The use of nighttime observations on the one hand meets the LPRM assumption of homogeneous temperature conditions (Owe et al., 2001) and on the other hand is better suited as proxy for plant water status than daytime observations. Due to diurnal differences in plant water status and the refilling during the night (El Hajj et al., 2019; Konings and Gentine, 2017), nighttime observations are closer to the predawn water potential which is commonly used as estimator for the daily vegetation water status (Konings and Gentine, 2017; Konings et al., 2019). During the processing of VODCAX, data are masked for radio frequency interference (RFI) (Moesinger et al., 2020) since RFI can introduce spurious retrievals (Li et al., 2004; Njoku et al., 2005). Data are available at daily resolution and 0.25° grid spacing.

### 2.2.2 Independent global GPP data sets

The MOD17A2H v006 product provides global estimates of GPP which are derived from surface reflectances (Running et al., 2004, 2015). The algorithm is based on the light-use efficiency concept by Monteith (1972) and uses the fraction of Photosynthetically Absorbed Radiation for deriving plant productivity (Running et al., 1999, 2000). Data are produced as 8-daily GPP estimates at 500 m resolution.

FLUXCOM presents an upscaling of GPP from eddy covariance measurements using an ensemble of machine learning approaches (Jung et al., 2020). The data set is available at 8-daily resolution and 10 km grid spacing. FLUXCOM estimates are produced in two setups: the FLUXCOM RS is based on remote sensing data as input to the machine learning models and the FLUXCOM RS+METEO uses meteorological data and only the mean seasonal cycle of remote sensing data (Jung et al., 2020). Since our approach is mainly based on remote sensing data, i.e. VOD observations, we used FLUXCOM RS in our analysis. The FLUXCOM algorithm uses the following MODIS variables as input: Enhanced Vegetation Index, Leaf Area Index, MODIS band 7 - Middle Infrared Reflectance, Normalized Difference Vegetation Index and Normalized Difference Water Index.

### 2.2.3 In situ GPP estimation from FLUXNET

The Fluxnet2015 data set (Gilberto et al., 2020) provides daily in situ estimates of carbon, water and heat fluxes, which are determined using the eddy covariance technique. GPP estimates are available for two flux partitioning methods, i.e. daytime and nighttime partitioning method. We used the mean of both partitioning methods, as suggested in (Gilberto et al., 2020), with variable friction velocity threshold (GPP_DT_VUT_REF, GPP_NT_VUT_REF) from the freely available station data set (Tier1 v1). Since data are available until 2014, we used data for the period from 2003 to 2014 as training data for estimating GPP based on VOD. An overview of the FLUXNET sites is given in Figure A3 and Table A1.

### 2.2.4 SPEI

For analyzing the impact of variations in water availability, we used SPEI from the SPEIbase (Beguería et al., 2017; Vicente-Serrano et al., 2010). The climatological water balance is calculated on different time scales ranging from 1 up to 48 months. Since drought can act on different time scales, we used SPEI at two different aggregations, 3- and 12-month, for investigating the response to dry and wet conditions. The 3-month SPEI (SPEI03) represents short-term effects, while the 12-month SPEI (SPEI12) relates to dry or wet conditions at annual time scale. Although SPEI cannot be used to express actual water shortage for plants, it allows to indicate relative deviations from mean conditions. Because of the use of both precipitation and temperature, SPEI further enables the comparison between different biomes (Vicente-Serrano et al., 2010). The SPEI data has monthly resolution and a grid spacing of $0.5°$.

### 2.2.5 ERA5-Land

ERA5-Land produced by the European Centre for Medium-Range Weather Forecasts (ECMWF) (C3S, 2019; Muñoz-Sabater, 2019) provides a reanalysis data set of meteorological parameters. ERA5 uses a 4D variational data assimilation scheme and a Simplified Extended Kalman Filter (Hersbach et al., 2020). We used skin temperature and snow data for masking VOD. In the VOD-GPP model, we incorporated 2m air temperature ($T2M$) for representing the temperature dependency of autotrophic respiration. $T2M$ was used in our analysis, since this parameter is most common for describing the temperature dependency of autotrophic respiration for above-ground vegetation (e.g., Ryan et al., 1997; Running et al., 2000; Ceschia et al., 2002; Drake et al., 2016). The data has hourly resolution and 9 km spatial sampling.

### 2.3 Data processing

VODCAX data were masked for low temperature (skin temperature $< 0°$C) and snow cover (snow depth $> 0$cm) and then aggregated to 8-daily estimates by computing the mean over 8 days to match the temporal resolution of GPPmodis and GPPfluxcom. These 8-daily values were then used as input to the VOD-GPP model and for further analysis throughout the study. GPPfluxcom and GPPmodis were aggregated to $0.25°$ to match the spatial sampling of VODCAX. For the comparison with SPEI, 8-daily GPP estimates were further resampled to monthly resolution while SPEI was spatially resampled to $0.25°$ using the nearest neighbour method.

## 2.4 GPP estimation based on VOD

The approach of estimating GPP based on microwave radiation and the corresponding equations are described in detail in Teubner et al. (2019). In short, the VOD-GPP model uses VOD as a proxy of above-ground living biomass (Equation 1). It determines GPP by estimating sinks for carbohydrates, i.e. the sum of NPP and Ra, which are represented through different VOD-derived variables: 1) time series of the bulk VOD signal ($VOD$; 8-daily aggregated native VOD time series), 2) time series of the temporal change in VOD ($\Delta VOD$; $\Delta VOD_t = VOD_t - VOD_{t-1}$ computed from the smoothed 8-daily aggregated VOD time series) and 3) the grid cell median of VOD ($mdnVOD$; calculated over the entire VOD time series of the grid cell; used as a proxy for vegetation cover). While NPP is related to $\Delta VOD$, Ra is related to both $VOD$ and $\Delta VOD$ using the concept proposed by Ryan et al. (1997) of dividing Ra into maintenance and growth respiration (Equation 2). By assuming that belowground biomass terms are proportional to above-ground biomass (i.e. biomass $B$ can be expressed through above ground biomass $AGB$) and by adding a static term $c$ supporting the conversion in Equation 2, GPP can be represented through a differential equation with VOD as input (Equation 3).

$$AGB = f(VOD) = \widetilde{VOD} \tag{1}$$

$$GPP = NPP + Ra = \left( \frac{dB}{dt} + \text{loss terms} \right) + \left( a_0 \, \frac{dB}{dt} + b_0 \, B \right) \approx a \, \frac{dB}{dt} + b \, B \tag{2}$$

$$GPP = a \, \frac{d\widetilde{VOD}}{dt} + b \, \widetilde{VOD} + c \tag{3}$$

The formulation in GAM for this previous model, which uses only VOD variables as input (GPPvod; Equation 4), then reads:

$$GPPvod = s(VOD) + s(\Delta VOD) + s(mdnVOD) \tag{4}$$

where $s$ denotes spline terms for representing the functions between each input variable and the response variable GPP in the 2-dimensional space.

For adding the temperature dependency of Ra, we are considering the two terms of Ra, i.e. maintenance and growth respiration. Since the temperature sensitivity mainly applies to the maintenance term (Ryan et al., 1997), we are only incorporating an interaction term with temperature for the maintenance part of the model formulation. Although all terms potentially may be dependent on temperature due to the general temperature dependency of enzymatic activity, the temperature dependency

for modelling growth related sink terms (growth respiration and net primary production) may be of less importance. For the current model formulation (GPPvodtemp; Equation 5), we now introduced an interaction term between $VOD$ and temperature:

$$\text{GPPvodtemp} = te(VOD, T2M) + s(\Delta VOD) + s(mdnVOD) \tag{5}$$

where $te$ stands for a tensor term, which represents the interaction between $VOD$ and temperature and spans a surface in the 3-dimensional space.

Consistent with our previous model, we used GAM as regression method for deriving GPP. The pyGAM (Servén and Brummitt, 2018) version 0.8.0 provides the possibility of adding an interaction term. An advantage of GAM is that the relationships between input variables and response variable are not required to be known beforehand, but instead can be estimated from the data itself (Hastie and Tibshirani, 1987). Since the relationship between VOD and GPP as well as its relationship with temperature is difficult to determine a priori, this method is well suited for our approach.

In GAM, a number of basis spline functions are fitted to the data and the resulting function is further smoothed to obtain the final response function (Servén and Brummitt, 2018). The degree of smoothing is determined by the smoothing factor, which yields strong smoothing for high values and low smoothing for low values. For the current models we used a smoothing factor of 2, which is lower than for the model in Teubner et al. (2019). This was done since the response function for the tensor term was too smooth using the default number of 10 splines for tensor terms and resulted in unrealistically high GPP values at high $VOD$. For $\Delta$VOD, the default number of 20 splines for spline terms were used, while for $mdnVOD$ we reduced the number of splines to 5 in order to obtain a smooth relationship.

## 2.5 Statistical analysis

For model comparison, we computed Pearson correlation, unbiased Root Mean Square Error (ubRMSE) and bias. For studying the error characteristics, ubRMSE was used instead of RMSE to exclude the impact of bias, which was observed during our analysis. In addition, cross validation was computed for the above metrics using the leave-site-out method, where the model performance is evaluated at each site by omitting the respective site data from model training and then using the left-out data for computing the statistics. The analysis was carried out for the full signal and the anomalies from the mean seasonal cycle.

In case of analyzing annual GPP anomalies as a measure for interannual variability and residuals of the VOD-GPP model, we based our analysis on standardized annual or 8-daily time series data (z-scores). This was done in order to analyze GPP data in the absence of systematic differences between the data sets. The standardization for the 8-daily or the annual data was applied to each grid cell time series by subtracting the mean and dividing by the standard deviation.

For generating the smoothed time series in the calculation of $\Delta VOD$ and for aiding visual comparison in time series plots, we applied a Savitzky-Golay filter with window size of 11 data points.

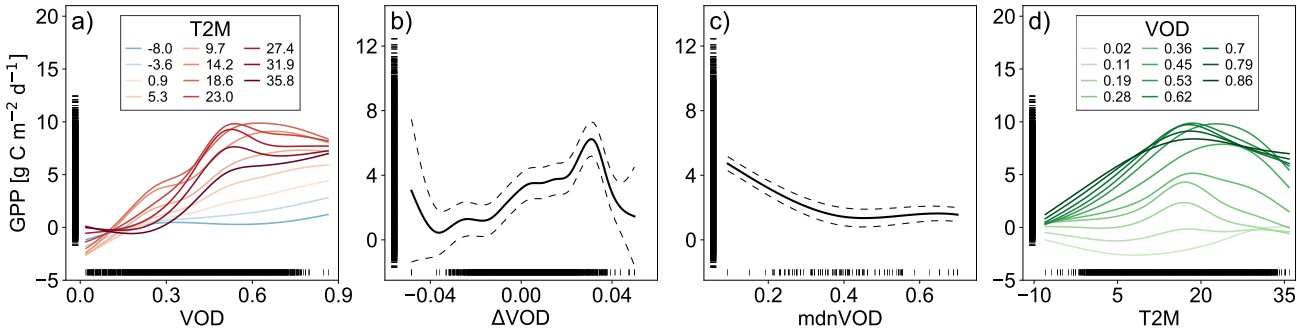

**Figure 1.** Partial dependency plot for GPPvodtemp for each input variable: (a) $VOD$, (b) $\Delta VOD$, (c) $mdnVOD$ and (d) $T2M$. The model was trained with data from the period 2003-2014. Dashed lines in (b) and (c) denote the 95% confidence interval. The interaction between $VOD$ and $T2M$ (a,d), which represents a surface in the 3-dimensional space, is displayed as projection on the 2D plane for each of the two input variables. For this, the parameter space was divided into 10 equally spaced bins between minimum and maximum of the respective variable. The bin edges are displayed as colored lines as indicated in the legend.

## 3 Results

### 3.1 Model representation of temperature dependency

We find that the sensitivity of $VOD$ to GPP increases with temperature as shown by the partial dependency plots (Figure 1). For low temperatures, the sensitivity of the VOD-GPP-relationship is relatively low (Figure 1a). As temperature increases, the sensitivity also increases and further exhibits an optimum behavior. At high temperatures, however, the maxima of the curves are lower than for moderate temperatures. The partial dependency for $T2M$ (Figure 1d) shows an optimum behavior with a peak around 20°C, which slightly differs between the $VOD$ values. The partial dependencies for $\Delta VOD$ and $mdnVOD$ (Figure 1b,c) are consistent with the previous model and yield an increasing relationship with GPP for $\Delta VOD$ in the middle part of the value range and a general decreasing relationship for $mdnVOD$.

In addition to identifying the underlying relationships, we can further assess the magnitude of the contribution to GPP for the input variables based on the data range in the partial dependency plots. The main contribution to GPP in the model comes from the interaction term between $VOD$ and $T2M$ with a range of about 12 gC m$^{-2}$ d$^{-1}$, which is followed by $\Delta VOD$ with a range of about 6 gC m$^{-2}$ d$^{-1}$ and $mdnVOD$ with a range of about 4 gC m$^{-2}$ d$^{-1}$. The contribution of the maintenance part, as represented through the interaction term, thus, is higher than for $\Delta VOD$ which represents the sum of NPP and the growth term in Ra.

### 3.2 Evaluation at site-level

At FLUXNET in situ stations, global GPP datasets overall show similar results (Figure 2). GPPvod exhibits a slight accumulation of GPP values at around 4 g C m$^{-2}$ d$^{-1}$, while the density for GPPvodtemp is relatively smooth and comparable to

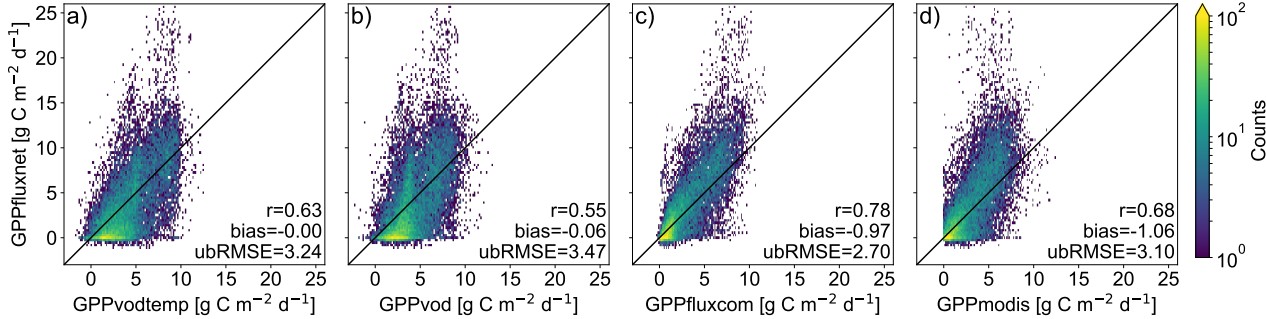

**Figure 2.** Scatter plots of 8-daily in situ GPPfluxnet versus global GPP data sets (a) GPPvodtemp, (b) GPPvod, (c) GPPfluxcom and (d) GPPmodis for the period 2003-2014.

GPPfluxcom and GPPmodis. Both GPPvod and GPPvodtemp show a relatively high number of non-zero GPP at around zero GPPfluxnet, which is less pronounced for GPPvodtemp than for GPPvod. Cross validation results in Table A2 further confirm

a higher performance of GPPvodtemp compared to GPPvod. For the full signal as well as for the anomalies from the mean cycle, correlation, ubRMSE and bias generally yield higher performance for GPPvodtemp. The increase in performance is more pronounced for the full signal than for the anomalies. Despite an overall agreement of GPPvodtemp, GPPfluxcom and GPPmodis with in situ GPP, all three data sets exhibit an underestimation of GPP at high values of GPP compared with in situ GPPfluxnet. At annual time scale, the difference with GPPfluxnet at high GPP becomes much lower for GPPvodtemp

compared to GPPfluxcom and GPPmodis (Figure A4), which indicates on the one hand that GPPvodtemp is able to match the in situ training data and on the other hand suggests that differences in GPP already exist between the training data set used in our study and the independent global GPP data sets, which may contribute to differences at global scale. The observed overestimation of GPP for GPPvodtemp at low in situ GPP can also be observed at annual time scale. This may be an explanation for the general tendency for overestimation of microwave-derived GPP estimates and appears not to be entirely related to the

temperature sensitivity of Ra, since it is still present for GPPvodtemp.

### 3.3  Impact of adding temperature dependency at the global scale

Performance metrics for GPPvod and GPPvodtemp were assessed with respect to both GPPfluxcom and GPPmodis. Since the results for GPPfluxcom and GPPmodis are similar, we are only showing results for GPPfluxcom.

Correlations with GPPfluxcom (Figure 3a) reveal widespread strongly positive values with a global mean of 0.63. Some areas

in the tropics and in the Australian desert exhibit an inverse temporal dynamic with GPPfluxcom. Compared with GPPvod, correlations increase in large parts of the world (Figure 3b) with a global average difference of 0.18. Regions that benefit most from adding temperature as input are temperate and cold regions, which could be expected since these regions per definition are strongly controlled by temperature. Tropics and subtropics, however, mainly show only minor changes in correlation coefficient

with a few exceptions of decreasing correlations. Since the annual temperature amplitude in these regions is low, the model's sensitivity to temperature is also low, which makes the interaction term mainly controlled by VOD.

The global average for ubRMSE between GPPvodtemp and GPPfluxcom (Figure 3c) yields a value of 1.20. Consistent with the increase in performance for the correlation, areas in the temperate and cold region show an improvement in error, i.e. a decrease of ubRMSE compared to GPPvod (Figure 3d). Other regions, however, exhibit an increase in ubRMSE. The global average of the difference between results for GPPvodtemp and GPPvod is -0.05. Therefore, gains and losses in error are largely compensated at the global scale.

The bias between GPPvodtemp and GPPfluxcom (Figure 3c) is generally positive everywhere with a global average of 1.64. This finding is also evident from the higher range in the median maps for GPPvodtemp compared with GPPfluxcom and GPPmodis (Figure A2). Comparing the results for GPPvod and GPPvodtemp, the addition of temperature shows an increase in bias mainly in the tropics (Figure 3d), which is also evident for the difference of the median maps (Figure A2e). Despite this increase in the tropics, also regions with a reduction in bias exist, which are mainly found in temperate and cold regions. On the global scale, decreases and increases in bias compensate and yield an average difference of -0.05.

The latitudinal distribution of annual GPP (Figure 4a) further demonstrates that the addition of temperature yields a reduction of GPP mainly for regions outside -35°N and +60°N. The reduction in the zonal mean, however, is smaller than may have been expected probably due to compensating effects. For the region between +30°N and +60°N, where reductions in bias were observed on the global map, positive and negative values for the bias appear to compensate yielding no net reduction in the zonal mean. In the tropical region, the increase in bias for GPPvodtemp compared with GPPvod is again evident. When considering the latitudinal distribution of annual GPP relative to the latitudinal maximum, however, the distribution for GPPvodtemp is actually closer to the independent datasets than GPPvod (Figure 4b). This suggests that although the bias largely increases in the tropics, the relative distribution between tropics and temperate to boreal regions is better represented by the setup that includes temperature.

For a region in Europe (5 to 15°E and 46 to 51°N), where we generally did observe an increase in all three performance metrics, we find that for GPPvod mainly winter time estimates of GPP are too high compared to GPPfluxcom and GPPmodis (Figure 5). By adding temperature as input to the model, winter observations are markedly dampened and summer observations are only slightly increased. Nevertheless, even when including the temperature dependency, winter GPP estimates are still slightly higher for GPPvodtemp than for GPPfluxcom or GPPmodis. A similar behavior is observed for other temperate regions (Figure A5).

In the remaining study, due to the observed bias (both at site-level and global scale), we are analyzing relative rather than absolute values for comparing interannual variability and the impact of water availability. In addition, we are focusing our further analysis on GPPvodtemp since this setup overall showed higher performance than GPPvod. Results for GPPvod are displayed in the supplement for comparison with GPPvodtemp.

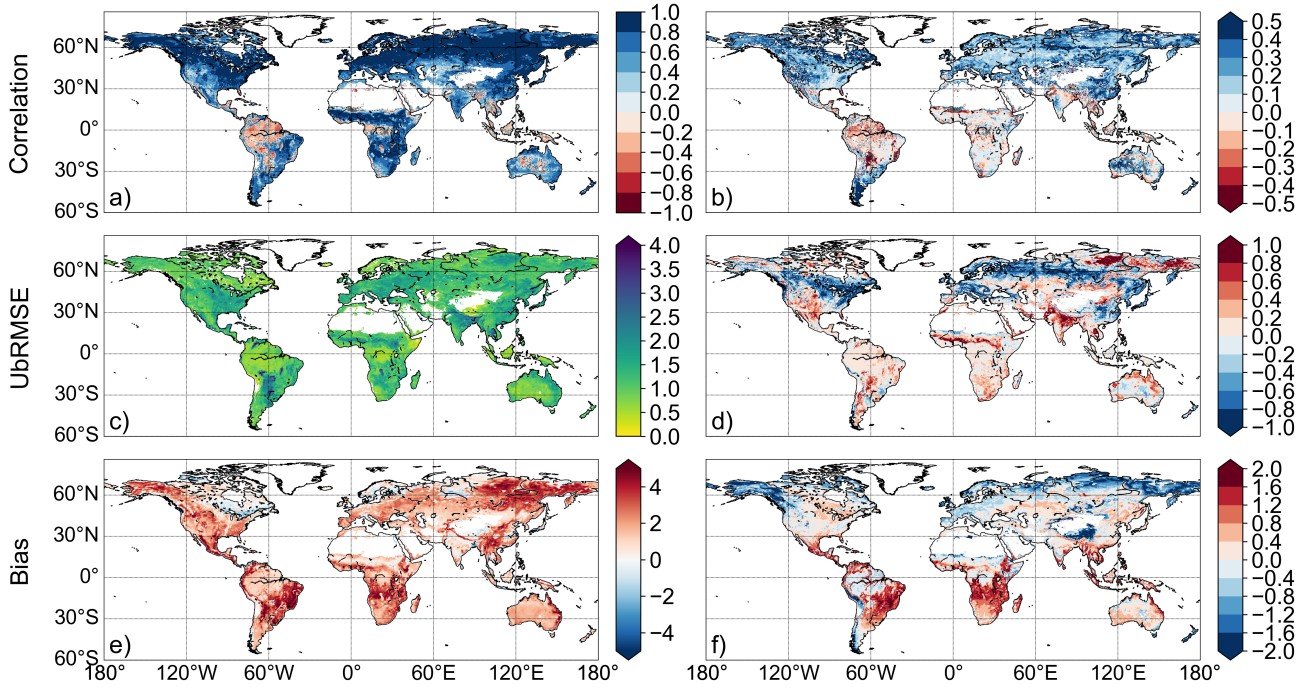

**Figure 3.** (a): Pearson correlation between GPPvodtemp and GPPfluxcom. (b): Difference between GPPvodtemp and GPPvod for Pearson correlation with GPPfluxcom. (c): ubRMSE between GPPvodtemp and GPPfluxcom. (d): Difference between GPPvodtemp and GPPvod for ubRMSE with GPPfluxcom. (e): Bias between GPPvodtemp and GPPfluxcom. (f): Difference between GPPvodtemp and GPPvod for the bias with GPPfluxcom. The unit for ubRMSE and bias is g C m$^{-2}$ d$^{-1}$. Areas with non-significant correlations in (a) and (b) are marked in grey. The analysis is computed over the whole study period (2003-2015).

## 3.4 Interannual variability and varying conditions of water availability

The latitudinal distribution of annual GPP anomalies reveals a general agreement between the GPP datasets (Figures 6 and A6). Although differences exist between all data sets, key features are observed among all data sets, such as the positive anomalies at -55°N in 2003, at -30°N in 2011 or at +75°N in 2012 and the negative anomalies at +75°N in 2003 and 2015 and at around -40° in 2009 and 2011. Despite the fact that these key features are found in all data sets, we also observe that the magnitude of the anomalies often differs between the data sets, which thus yields a generally relatively high variability between all data sets. In terms of the overall latitudinal pattern, it appears that GPPvodtemp is more similar to GPPmodis than to GPPfluxcom.

For the correlation of the residuals between standardized GPP (GPPvodtemp-GPPfluxcom or GPPvodtemp-GPPmodis) and SPEI, we find that large areas show no significant correlation with SPEI03 (Figure 7a,b). For the long-term climatological water balance, i.e. SPEI12 (Figure 7c,d), these areas with non-significant correlations further increase. In terms of model applicability, the non-significant correlations are the desired result. Given that correlations between GPPvodtemp and GPPfluxcom or GPPmodis are high in these regions, this demonstrates that GPPvodtemp shows a similar behavior as GPPfluxcom or GPPmodis in

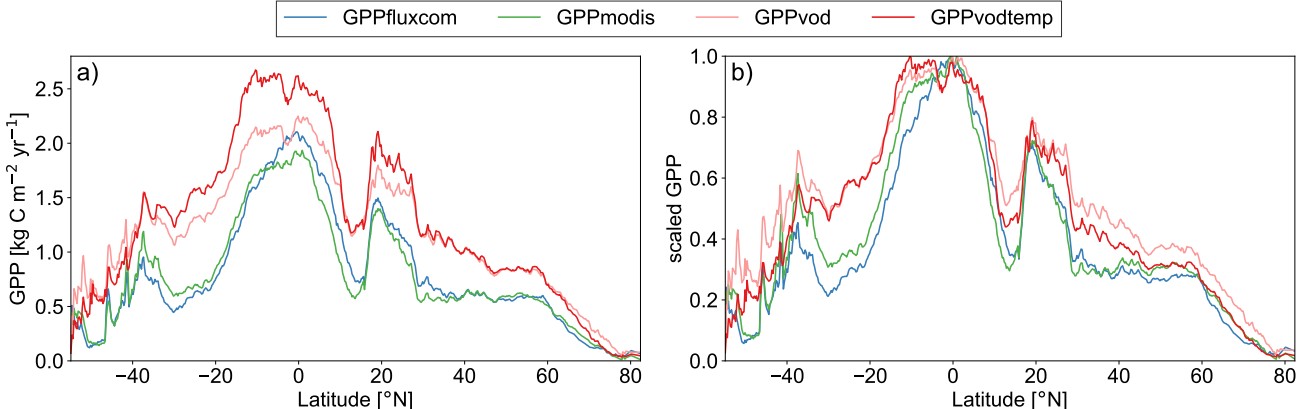

**Figure 4.** Zonal mean of annual GPP for GPPfluxcom, GPPmodis, GPPvodtemp and GPPvod for the study period 2003-2015. (a): Absolute latitudinal distribution. (b): Scaled latitudinal distribution. To obtain zonal means, data were averaged over all grid points of the same latitude. Scaled data were computed by dividing the latitudinal distribution by the maximum of the latitudinal distribution for each data set.

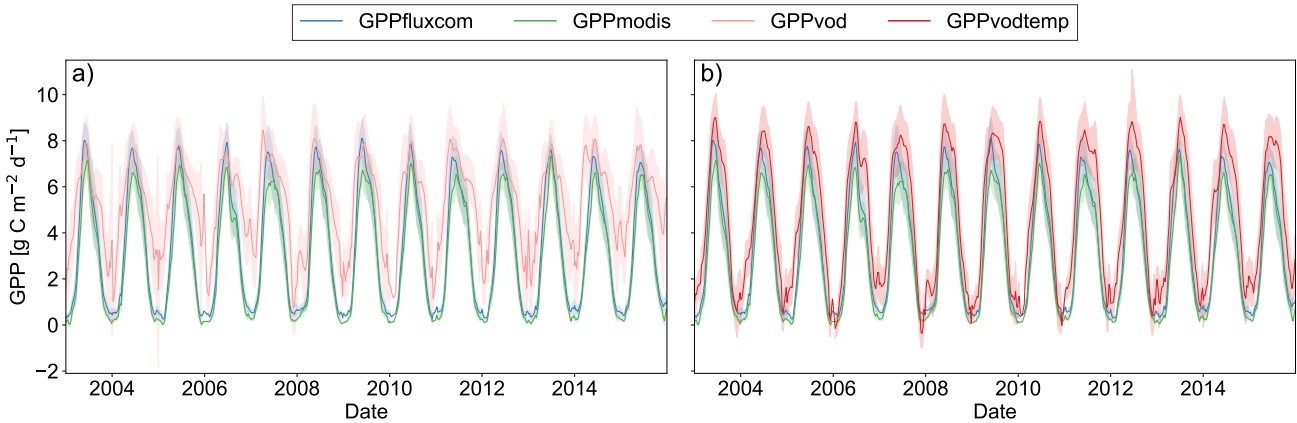

**Figure 5.** Time series plot of spatially aggregated GPP estimates for GPPfluxcom, GPPmodis and (a) GPPvod or (b) GPPvodtemp over the whole study period (2003-2015). Shaded areas indicate the standard deviation over the aggregated grid cells. The region is located in Europe, 5 to 15°E and 46 to 51°N, and was selected as an example where the correlation analysis between GPP residuals and SPEI largely yield no significant correlations. 8-daily data were smoothed to aid visual comparison.

response to variations in dry or wet conditions. This finding thus provides a strong indication that the VOD-GPP-relationship generally remains similar under varying conditions of water availability.

Apart from the widespread areas with non-significant correlation, some significant correlations, both positive and negative, occur at both time scales. Negative correlations indicate that during dry conditions GPPvodtemp is higher relative to the reference GPP than during wet conditions, while positive correlations mean that during dry conditions GPPvodtemp is lower

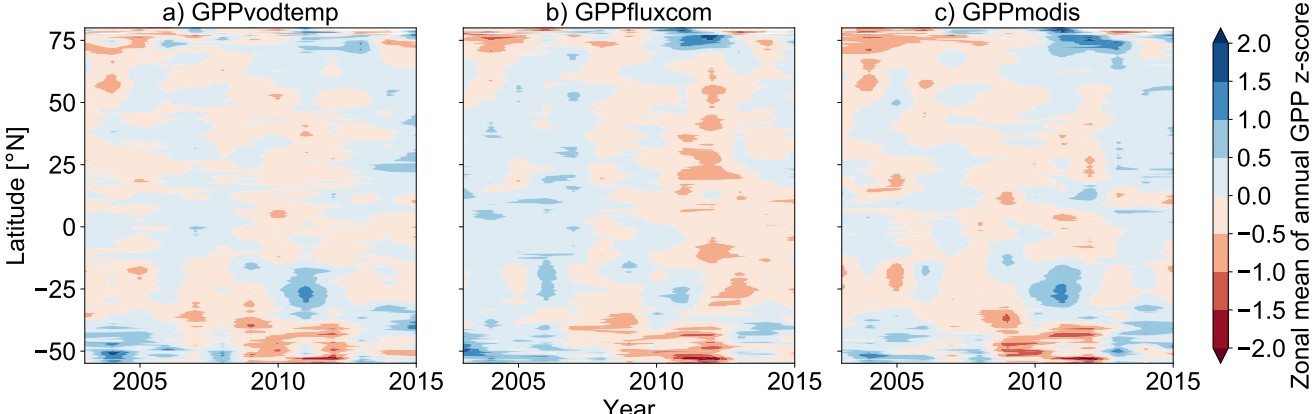

**Figure 6.** Hovmöller diagramm for zonal means of annual GPP anomalies (z-scores) for (a) GPPvodtemp, (b) GPPfluxcom and (c) GPPmodis over the study period. Zonal means were calculated by averaging data over all grid points of the same latitude.

relative to the reference GPP than during wet conditions. The spatial distribution of these significant correlations is largely consistent between GPPfluxcom and GPPmodis. For the short-term response to SPEI (Figure 7a,b), negative correlations are more frequent than positive correlations, indicating that the response to short-term drought events is often a reduction of source-driven GPP relative to sink-driven GPP. Negative correlations are mainly observed in the US corn belt, Argentina, Eastern Europe, Russia and China, with the strongest negative correlations being in the US, Argentina and Russia. Positive correlations are obtained mainly over South America, Africa and Australia. For the long-term response to SPEI (Figure 7c,d), the number of positive correlations increase. Similar to the short-term response, positive correlations are mainly found over South America, Africa and Australia.

The analysis of GPPvod residuals reveals a similar result as for GPPvodtemp (Figure A7). For GPPvod, however, the number of grid cells with non-significant correlations in the four analyses is lower by about 2 to 4 % than for GPPvodtemp, while the global average correlation is nearly identical. The higher number of non-significant correlations for GPPvodtemp than for GPPvod is expected, because the addition of temperature accounts for some variation in the VOD-based GPP estimation.

For specific regions, which are indicated in Figure 7, we analyzed the time series of the standardized GPP (Figure 8) and the response to SPEI categories (Figure A8) in order to inspect under which situations negative or positive correlations with SPEI occur.

For the region in the US corn belt (Figure 8a), where we found moderately negative correlations with SPEI, all three GPP data sets show a reduction in summer GPP in 2006 and 2012. Compared with other years, however, the reduction of GPPvodtemp tends to be less than for GPPfluxcom and GPPmodis. This behavior can be verified by considering the residuals along the SPEI12 gradient (Figure A8a). During dry conditions, the residuals are higher than during wet conditions. Since higher residuals indicate that GPPvodtemp is higher relative to the reference data sets, this result confirms the findings for the time series.

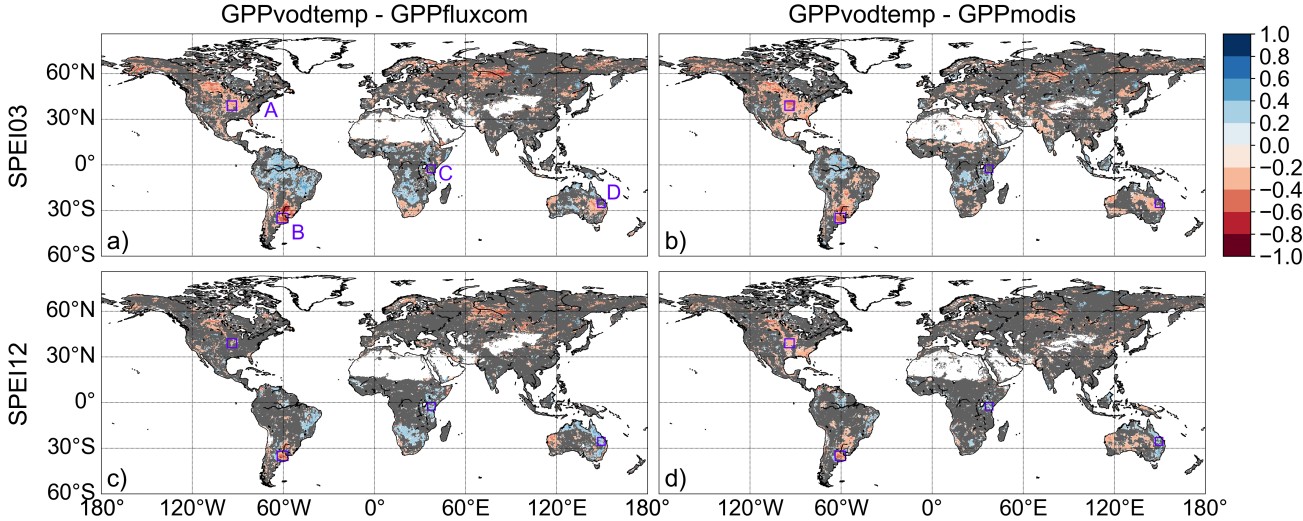

**Figure 7.** Correlation between residuals of standardized GPP (GPPvodtemp-GPPfluxcom and GPPvodtemp-GPPmodis) and SPEI. Non-significant correlations are indicated in grey. (a,c): GPPvodtemp-GPPfluxcom, (b,d): GPPvodtemp-GPPmodis, (a,b): SPEI03 (short-term response), (c,d): SPEI12 (long-term response). Regions A-D: US cornbelt (A), Argentina (B), Eastern Africa (C) and Eastern Australia (D). The analysis is based on the whole study period (2003-2015).

In Argentina (Figure 8b), we observed strongly negative correlations for the analysis with SPEI. For this region, a pronounced dry condition is observed at the end of 2008 and beginning of 2009. In this period, GPPfluxcom and GPPmodis are reduced more strongly than GPPvodtemp. In the first following year, the GPPvodtemp peak is slightly lower than for GPPfluxcom and GPPmodis at the end of 2009. In the second following year, end of 2011, GPPvodtemp is similar as for GPPfluxcom and GPPmodis again. This result is further supported by the pronounced decrease of the residuals with SPEI12 in Figure 8b. In addition to the interannual variability, we also find that the spring peak is more pronounced in GPPfluxcom and GPPmodis than in GPPvodtemp, which might point towards a surplus of carbohydrates in spring that are incorporated for building up biomass later in the year or may be related to differences in land cover.

For the example in Africa (Figure 8c), where correlations with SPEI12 were positive, GPPvodtemp generally appears to be a bit higher relative to GPPfluxcom and GPPmodis at the end of each growing period. In face of dry conditions, however, GPPvodtemp shows a stronger reduction in GPP than GPPfluxcom and GPPmodis at the end of the growing season, as observed in 2006 and 2009. Despite some differences in the time series between GPPvodtemp and the reference data sets, the temporal dynamic is generally similar between the data sets. This indicates that the sink-driven GPP shows a slightly different response to changes in environmental conditions for this region, which then results in the observed positive correlations with SPEI. Considering the residuals along the SPEI12 gradient for this region, we find that the residuals increase with SPEI12 for all categories except for very wet conditions (Figure A8c).

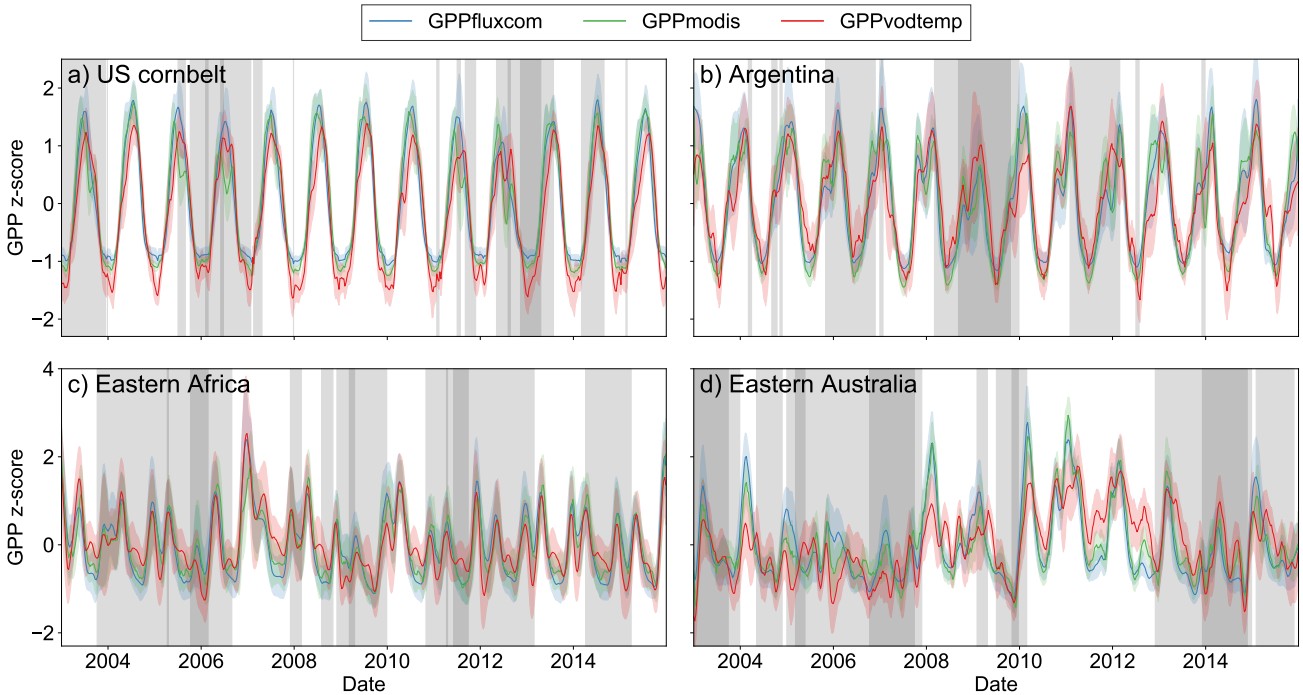

**Figure 8.** Regional mean of standardized GPP values for regions as indicated in Figure 7 over the study period. Shaded areas denote the standard deviation for the regional aggregated time series. Vertical grey areas indicate periods with different levels of dryness conditions for regional aggregated SPEI12: SPEI12<-1 (dark grey), -1<=SPEI12<0 (light grey) and SPEI12>=0 (white areas). Data were smoothed to aid visual comparison.

The time series for Australia (Figure 8d) shows that GPPvodtemp is generally reduced during dry conditions and increases relative to GPPfluxcom and GPPmodis during wet conditions. The increase in GPPvodtemp relative to the reference data sets appears to be strongest for the period following one year after long-term dry conditions, i.e. in 2009, 2011 and 2012. Consistently, the residuals show a clear increase along the SPEI12 categories (Figure A8d).

## 4   Discussion

**4.1   Impact of adding temperature as model input**

The performance of the VOD-GPP model was shown to improve with the addition of an interaction term between $VOD$ and temperature mainly in terms of temporal dynamic. Our results showed that the improvement in temporal dynamic was mainly observed for temperate and cold regions. Since the growing season in these regions is largely controlled by temperature, this indicates that the improvement may largely be a seasonal effect. When analyzing the temperature response of respiration across

biomes, both spatial and temporal differences resulting from thermal acclimation need to be taken into account (Vanderwel et al., 2015). On the spatial scale, temperature sensitivity largely varies with mean annual temperature across biomes (Piao et al., 2010; Vanderwel et al., 2015). On the temporal scale, temperature-corrected respiration rates, as observed for stem respiration of deciduous trees or for needle-leave evergreen trees, exhibit a seasonal variation leading to higher respiration rates during summer than during winter (Maier et al., 1998; Ceschia et al., 2002; Vose and Ryan, 2002; Zha et al., 2004).

Consistently, we observed a dampening of GPPvodtemp during winter compared to GPPvod. The addition of temperature thus seems to enable the model to reflect differences in basal respiration rates between growing and dormant periods in these regions. Although the temporal component of thermal acclimation of respiration appears to be the dominant contribution, the resulting dependency on temperature represents the cumulative effect of spatial and temporal thermal acclimation of respiration as the relationship for the temperature dependency was estimated from the data without a priori assumptions.

In addition to the temperature dependency, Ra also varies with tissue nitrogen content (Maier et al., 1998; Ceschia et al., 2002; Vose and Ryan, 2002; Tjoelker et al., 2008), which may thus contribute to uncertainties in the GPP estimation derived from VOD. Ra is also known to vary between plant tissues (Vose and Ryan, 2002; Gifford, 2003). The respiration of woody tissue is generally lower than for leaves (Vose and Ryan, 2002). Since VOD generally increases with the fraction of woody vegetation (Chaparro et al., 2019), using the median of VOD as model input may potentially compensate at least partly for

differences in respiration rates of stems and branches versus leaves within a grid cell.

## 4.2 Bias between GPP data sets

The addition of temperature dependency revealed contrasting results for the bias. While reductions in bias were observed for temperate and cold regions, a strong increase in bias was found for the tropics. Since the interaction term between $VOD$ and $T2M$ represents a relationship in the 3-dimensional space, certain combinations of $VOD$ and $T2M$ intervals in the parameter

space may not be well represented by the training data. FLUXNET stations are not evenly distributed around the globe, as the majority of stations are located in the temperate region. This may have caused the model to be not well constrained in certain regions, e.g. where temperature and $VOD$ are very high, and thus might have contributed to the increase in bias in the tropics. Therefore, additional FLUXNET stations might help to better constrain the VOD-GPP model. Nevertheless, differences between the dataset were already evident at the site-level, which suggests that the observed difference at global scale may at

least partly be caused by differences in the training dataset. In general, the agreement in annual GPP estimates is lowest in the tropics (Anav et al., 2015). Estimates for the FLUXCOM RS setup, which was used in our study, were reported to yield lower global estimates than the FLUXCOM RS+METEO setup or GPP estimates from vegetation models (Jung et al., 2020). Similarly, MODIS was found to underestimate GPP in the tropics (Turner et al., 2006). The need for better constraints for GPP estimates especially in the tropics is well recognized (MacBean et al., 2018) and tackled in different studies (e.g., MacBean

et al., 2018; Sun et al., 2018; Wu et al., 2020) but is usually hampered by the availability of in situ estimates.

### 4.3 Implications of possible saturation of VOD at high biomass

The choice of microwave frequency for the estimation of GPP may have certain implications. Different studies have demonstrated that L-band VOD yields more robust estimates of total above-ground biomass than X-band VOD, as low frequency VOD does not saturate at high biomass values (Chaparro et al., 2019; Frappart et al., 2020; Li et al., 2021). Nonetheless, the impact of such potential saturation with biomass on the estimation of GPP is less trivial, especially with regard to densely vegetated areas like the tropics. Non-linearity in the conversion between VOD and AGB should ideally be reflected in the partial dependency plot of GAM, which was also the reason for choosing this type of modelling approach. Scatterplots of the resulting GPPvodtemp estimates did not show clear signs of saturation at high in situ GPP. The FLUXNET training data set, however, only has few stations in the tropics and thus the robustness of the model may be limited by the availability of in situ stations. Apart from this, the relationship between VOD and GPP has been found to be in closer agreement for X-band VOD than for L-band (Teubner et al., 2018, 2019; Kumar et al., 2020), which was also observed for the correlation with in situ FLUXNET GPP (Figure A1). At first glance, this might appear contradictory to the above-mentioned better performance of L-band VOD for biomass estimation. A comparison of biomass estimates from different plant components with GPP, however, demonstrated that large structural components, which make up a large fraction of the total biomass, may contribute less to GPP than metabolically active plant parts (Litton et al., 2007). Since high frequency VOD is more sensitive to small plant parts like leaves and twigs (Woodhouse, 2017), this could be an explanation why X-band VOD might be better suited for the estimation of GPP and why saturation at high total above-ground biomass may be less of an issue here.

### 4.4 Independence of global GPP data sets

For the comparison with VOD-based GPP estimates, we used independent global data set from FLUXCOM and MODIS. Both data sets include to some extent information from FLUXNET data. FLUXCOM has been trained against FLUXNET data (Tramontana et al., 2016; Jung et al., 2020), however, with a larger number of stations than in the freely available Tier 1 data set that was used for our model. Also, MODIS has been partly calibrated to some FLUXNET stations (Running et al., 1999). Therefore, the FLUXCOM and MODIS may not be fully independent from our VOD-based GPP estimates. Nevertheless, there is no alternative to constrain absolute GPP estimates at global scale than by using FLUXNET data. In addition, the agreement between GPP and VOD-based GPP estimates was also confirmed at site level using leave-site-out cross validation. Since this analysis is independent from the comparison with global data sets, it supports the use of VOD for deriving GPP.

### 4.5 The "zero-GPP problem" and non-structural carbohydrates

For GPPvodtemp, we observed that winter GPP values for an example over Europe were slightly higher compared to GPPfluxcom and GPPmodis. This issue of estimating GPP values close to zero was also observed in the scatter plots between GPPvodtemp and in situ GPPfluxnet. The reason for the overestimation at low GPP may be on the one hand an artefact related to the rehydration of plant residues after rain events and on the other hand may be explained by the sink-driven nature of our approach. In the latter case, the non-zero GPPvodtemp values may be caused by perennial vegetation. Both evergreen and de-

ciduous vegetation are respiring throughout the dormant period (Maier et al., 1998; Vose and Ryan, 2002) and concurrently are containing water. In turn, this presence of vegetation water content is detected through microwave sensors leading to non-zero

GPPvodtemp estimates. It thus may point towards the existence of a storage term. In plants, photosynthetic assimilates can be stored in the form of non-structural carbohydrates (NSC), which can be converted back to plant usable sugars to support respiration during the dormant period and growth at the start of the growing season (e.g., Martínez-Vilalta et al., 2016). For tropical forest plots, the balancing of plot level measurements of source and sink terms showed a decoupling between the two in response to drought which the authors attributed to the existence of NSC (Doughty et al., 2015). Therefore, such a storage term

can thus support a temporary imbalance between sources and sinks of carbon, which may translate into differences between source- and sink-driven GPP.

### 4.6    Magnitude of input terms

Based on the partial dependency plots, we found that for the maintenance-related term, i.e. the interaction term between $VOD$ and $T2M$, the value range is higher than for $\Delta VOD$. Although our model represents the sum of NPP and growth Ra and not just

growth Ra, the magnitude of the two input terms is consistent with studies that analyzed the contribution of maintenance and growth to Ra. For whole plants as well as for stem respiration of boreal needle-leave trees, maintenance respiration was shown to play the dominant role for Ra with a contribution 70% (Chambers et al., 2004) and 80% (Zha et al., 2004), respectively.

### 4.7    Response to water availability

The analysis of VOD-GPP residuals with respect to FLUXCOM and MODIS revealed that GPPvodtemp largely showed a

similar behavior as the independent GPP data sets as demonstrated by the widespread non-significant correlations with SPEI. This result is further supported by the general agreement in interannual variability. In addition to the possible impact of NSC, occurrences of significant correlations between VOD-GPP residuals and SPEI may indicate different plant strategies for dealing with changes in dry or wet conditions. For negative correlations, this could be mainly related to differences in plant hydraulics, while for positive correlations, it might indicate shifts between above- and belowground carbon allocation.

Different plant strategies with regard to hydraulics can be expressed with the concept of isohydricity, which describes the regulation of stomatal control (Konings and Gentine, 2017; Giardina et al., 2018; Martínez-Vilalta and Garcia-Forner, 2017). At ecosystem level, this parameter can be obtained using the difference in twice daily overpasses of microwave observations (Konings and Gentine, 2017). Although Martínez-Vilalta and Garcia-Forner (2017) argue that the regulation of water potential may not necessarily be strongly coupled with the assimilation during drought, the degree of isohydricity may still be an

explanation for the observed variation in GPPvodtemp relative to GPPfluxcom and GPPmodis. Pronounced negative correlation for the analysis of GPP residuals were found in Argentina and the US corn belt, which are regions where Konings and Gentine (2017) observed high values of isohydricity. Corn, which exhibits isohydric behavior (Lambers and Oliveira, 2019; Martínez-Vilalta and Garcia-Forner, 2017), i.e. is maintaining water potential through strong regulation of stomata, additionally has the ability, like other grasses, to roll up leaves in response to drought for reducing the loss of water from the plant's cuticular (e.g.,

Ribaut et al., 2009). In conjunction with the isohydric behavior, this might be an explanation for the strong signal reduction of

GPPfluxcom and GPPmodis relative to GPPvodtemp observed over Argentina. Although our analysis is based on 8-daily time steps, characteristics of plant hydraulics which are retrieved from sub-daily data show similar features as for our analysis of residuals between source- and sink-driven GPP in response to changes in water availability.

In contrast to the isohydric behavior, anisohydric behavior should not lead to pronounced differences between GPPvodtemp and GPPfluxcom or GPPmodis as stomatal conductance and leaf water potential are both reduced in response to dry conditions (Lambers and Oliveira, 2019). The anisohydric behavior thus potentially relates to the non-significant correlations. Nevertheless, the degree of isohydricity may also vary between wet and dry season (Konings and Gentine, 2017), which also needs to be taken into account for the interpretation of the residuals.

The observed positive correlations, i.e. reductions of GPPvodtemp relative to GPPfluxcom or GPPmodis, could be associated with a stronger shift of assimilates to belowground plant organs. Different studies have shown that root growth may increase in face of drought to maintain water access (Sanaullah et al., 2012; Burri et al., 2014) and consequently also nutrient supply (Lambers and Oliveira, 2019). Since VOD observations only detect above-ground living vegetation, a shift towards belowground plant organs may lead to apparently lower GPPvodtemp. Nevertheless, also the inverse, i.e. an increase of allocation to shoots, was observed in the presence of legume species during drought (Sanaullah et al., 2012) and for tropical forest plots after drought (Doughty et al., 2015).

Comparisons of GPPvodtemp with in situ observations of vegetation properties during such extreme events like drought, however, may be needed to improve the understanding of the plant's response to changes in environmental conditions at the ecosystem to global scale.

## 5   Conclusions

The VOD-GPP model was analyzed with regard to the impact of adding temperature as model input in order to account for the temperature dependency of autotrophic respiration. The resulting GPP estimates, GPPvodtemp, showed a high consistency with GPPfluxcom and GPPmodis for the temporal dynamic both at intra- and interannual time scale. For bias and error, the addition of temperature resulted in a regionally diverse response with a general improvement for temperate and cold regions and a decrease in performance mainly in the tropics. The improvement upon adding temperature, however, was less than might have been expected, which indicates that the previous lack of temperature dependency in the model formulation can only partly account for the observed differences between the global GPP datasets. Nevertheless, this result demonstrates that an improvement by adding temperature is possible but might require further model constraints for a more robust estimation of GPP.

The analysis of the VOD-GPP residuals revealed that GPPvodtemp largely yields a similar behavior as GPPfluxcom and GPPmodis with respect to SPEI. This highlights that the relationship between VOD and GPP generally may be valid even under varying conditions of water availability. For some regions, where significant correlations were observed, the observed differences between GPPvodtemp and GPPfluxcom or GPPmodis may indicate different plant strategies for dealing with drought conditions.

Overall, our results showed that GPPvodtemp potentially contains information on plant characteristics that may be relevant for large-scale ecological studies that are addressing the response to varying environmental conditions.

*Data availability.* VODCA products are available at https://doi.org/10.5281/zenodo.2575599. FLUXCOM products are available from http://www.fluxcom.org or on request to Martin Jung (mjung@bgc-jena.mpg.de). MODIS GPP estimates can be accessed at https://lpdaac.usgs.gov/products/mod17a2hv006/. Data form the FLUXNET network is available at https://fluxnet.org/data/fluxnet2015-dataset/.

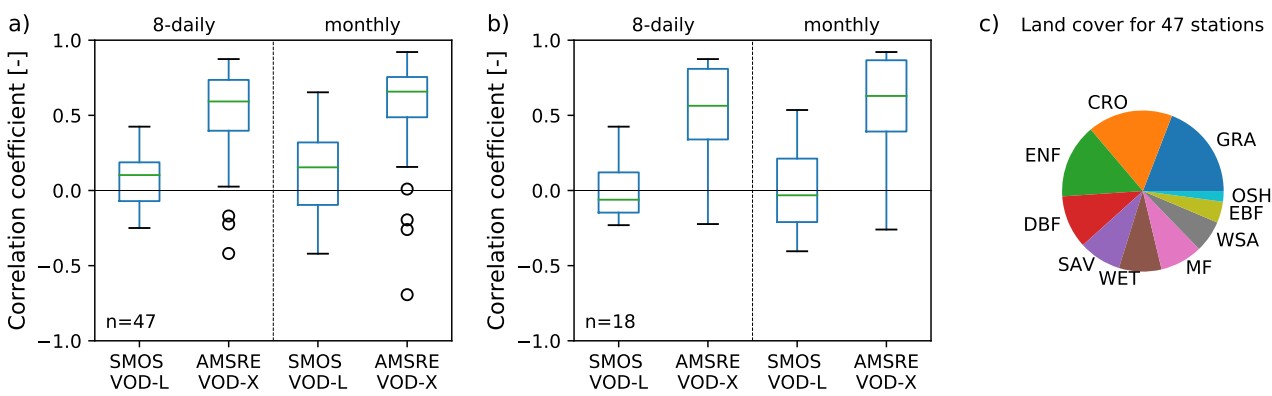

**Figure A1.** Pre-analysis of correlation between in situ FLUXNET GPP and single sensor VOD from L- and X-band. (a): Pearson correlation between FLUXNET GPP (mean of GPP_DT_VUT_REF and GPP_NT_VUT_REF) and L-band VOD (SMOS VOD-L, 7/2010–12/2014) and X-band VOD (AMSR-E VOD-X, 1/2007–9/2011). Data were resampled to 8-daily or monthly values. The analysis was conducted only for stations where both of the VOD data set are available (47 stations). For details about the VOD datasets and their data processing, see Teubner et al. (2018). (b): As in (a) but for the subset of forest land cover classes (ENF, DBF, EBF and MF). (c): Composition of IGBP land cover classes for the stations used in this pre-analysis. Abbreviations: GRA (Grasslands), CRO (Croplands), ENF (Evergreen Needleleaf Forests), DBF (Deciduous Broadleaf Forests), EBF (Evergreen Broadleaf Forests), SAV (Savannas), MF (Mixed Forests), WET (Permanent Wetlands), WSA (Woody Savannas) and OSH (Open Shrublands).

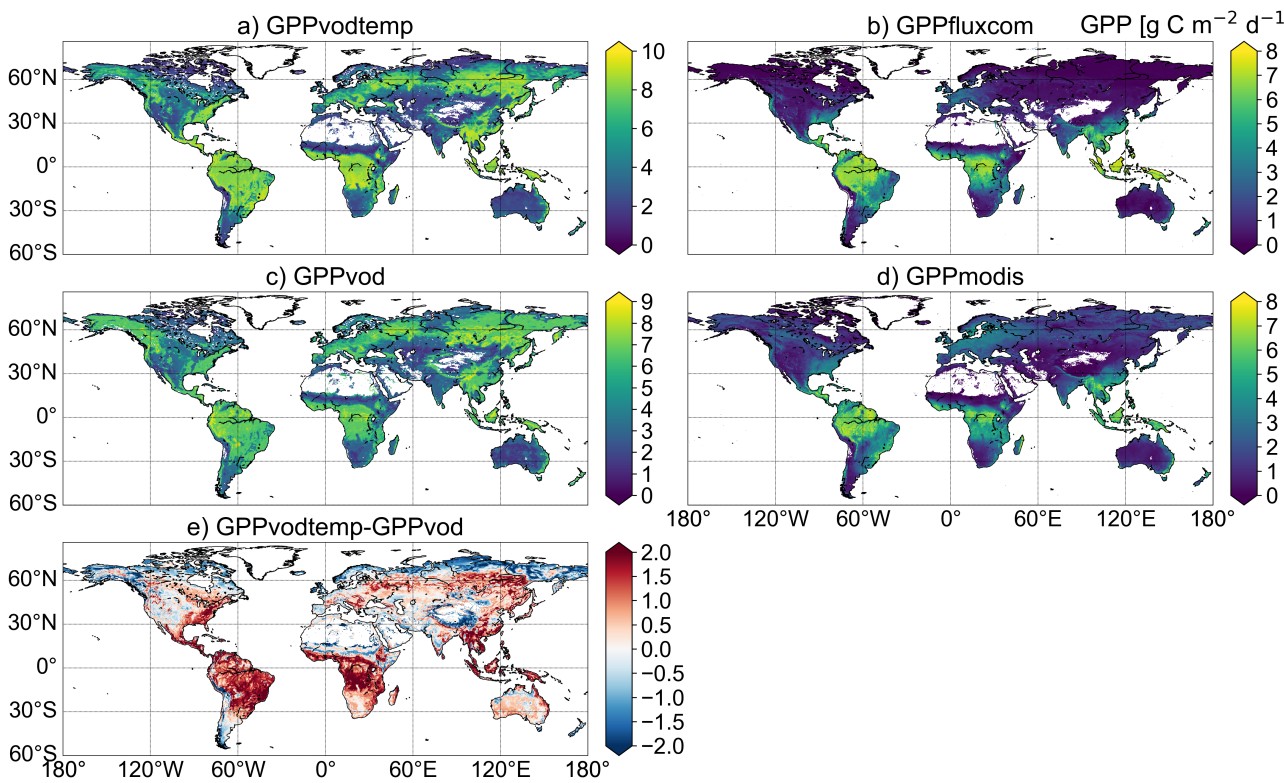

**Figure A2.** Temporal median maps for (a) GPPvodtemp, (b) GPPfluxcom, (c) GPPvod, (d) GPPmodis and (e) difference between the median maps of GPPvodtemp and GPPvod. For GPPvodtemp and GPPvod, areas where both GPPfluxcom and GPPmodis are missing were masked, since these data were not used during the analysis. Data were computed over the whole study period (2003-2015).

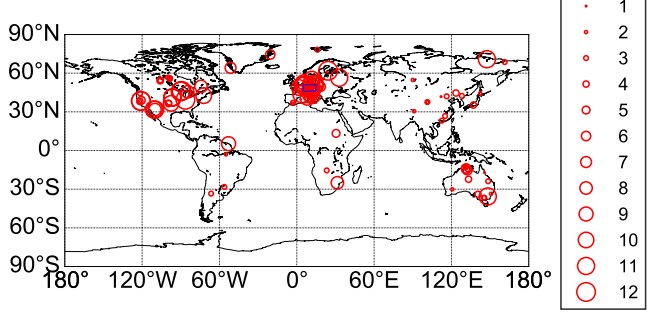

**Figure A3.** Location of FLUXNET Tier1 v1 stations within the period 2003 to 2014. The size of the circles represents the number of available years for each station. The blue rectangle denotes the location of the region in Europe used Figure 5.

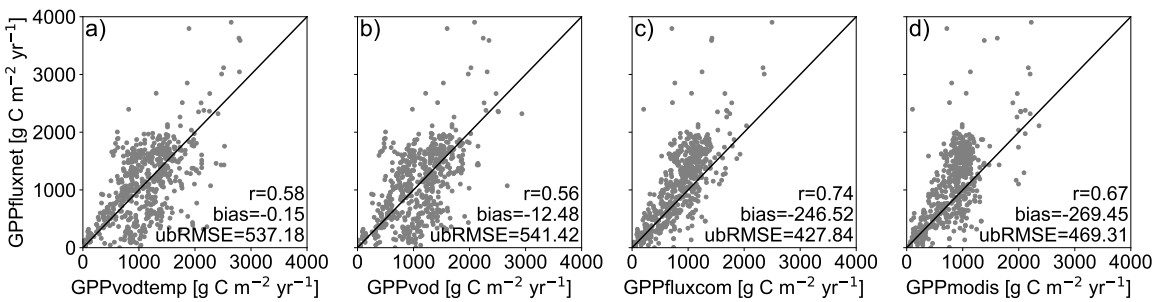

**Figure A4.** Scatterplot of annual GPP for GPPfluxnet versus (a) GPPvodtemp, (b) GPPvod, (c) GPPfluxcom and (d) GPPmodis. Annual values were calculated from 8-daily GPP for each data set and cover the FLUXNET period 2003-2014.

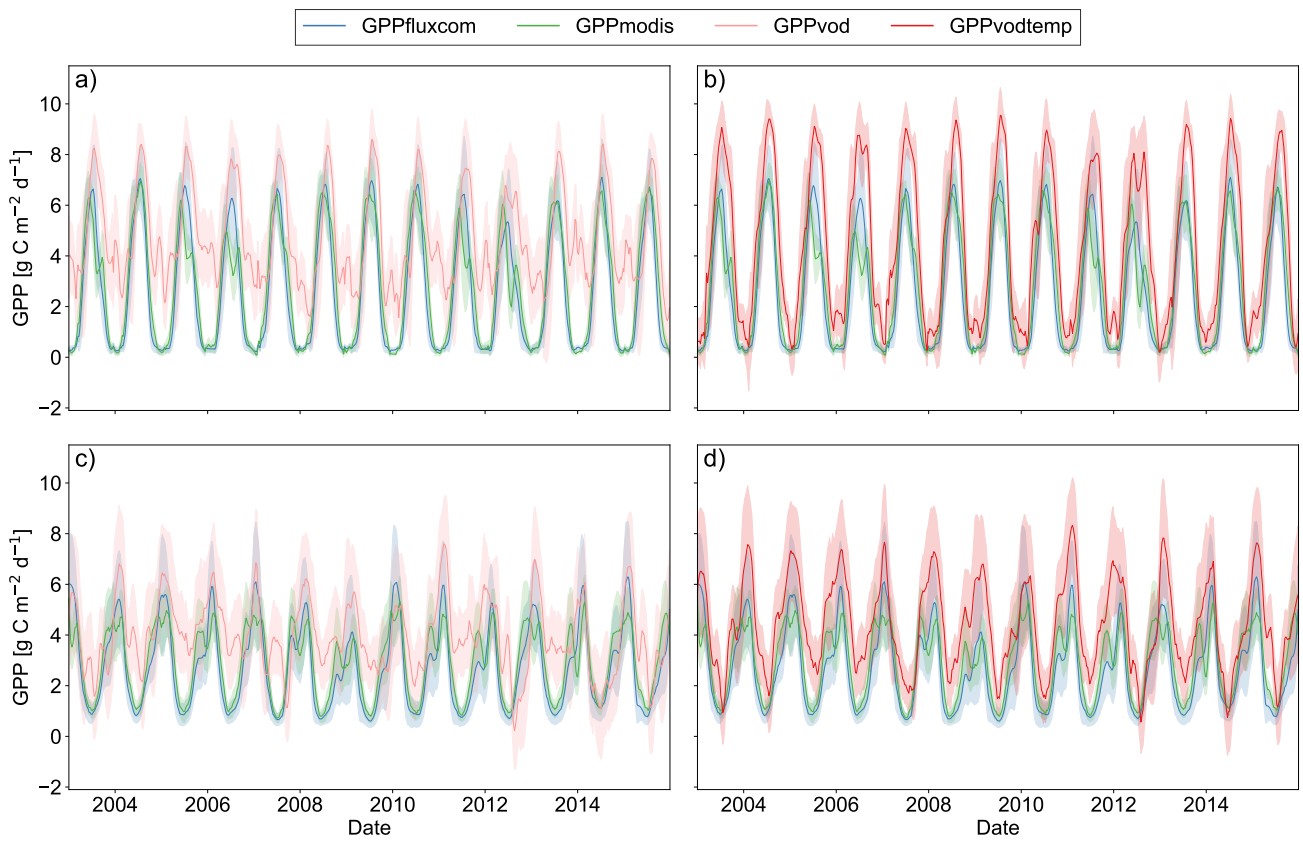

**Figure A5.** Time series plot of spatially aggregated GPP estimates for GPPfluxcom, GPPmodis and (a,c) GPPvod or (b,d) GPPvodtemp for the two regions US cornbelt (a,b; region A) and Argentina (c,d; region B) from Figures 7, 8 and A7. The analysis is based on the study period 2003-2015. Shaded areas represent the standard deviation over the aggregated grid cells. 8-daily data were smoothed to aid visual comparison.

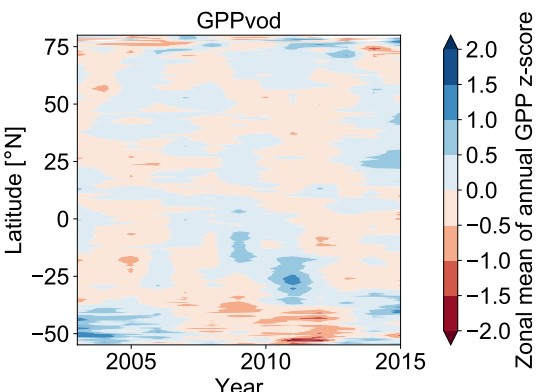

**Figure A6.** Hovmöller diagramm for zonal means of annual GPP anomalies (z-scores) for GPPvod over the study period 2003-2015.

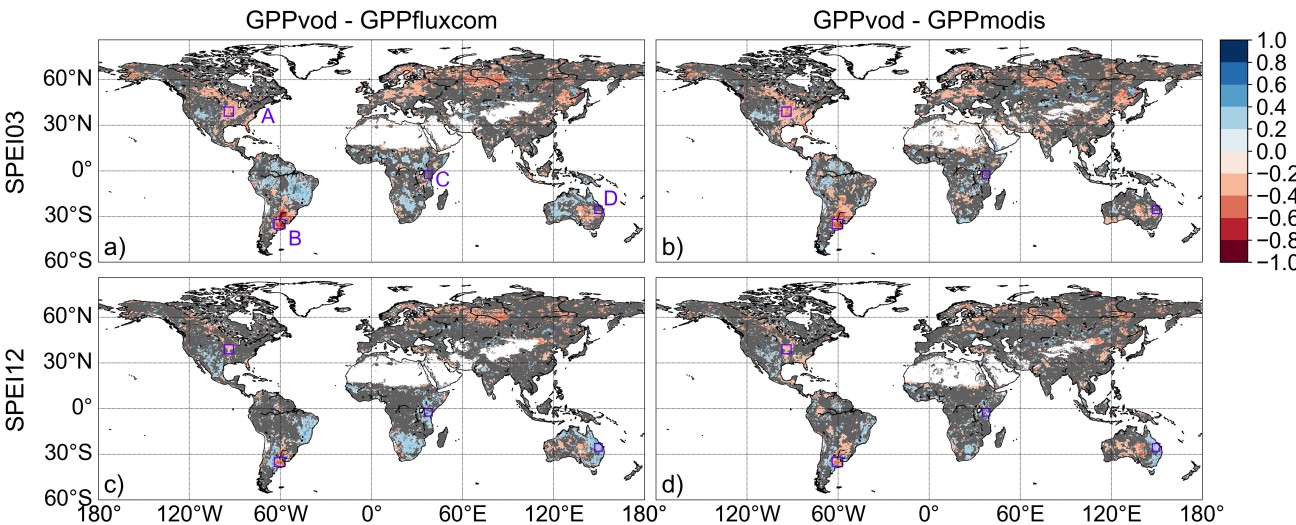

**Figure A7.** Correlation between residuals of standardized GPP (GPPvod-GPPfluxcom and GPPvod-GPPmodis) and SPEI. Non-significant correlations are indicated in grey. (a,c): GPPvod-GPPfluxcom, (b,d): GPPvod-GPPmodis, (a,b): SPEI03 (short-term response), (c,d): SPEI12 (long-term response). Regions A-D: US cornbelt (A), Argentina (B), Eastern Africa (C) and Eastern Australia (D). Results are computed based on the study period 2003-2015.

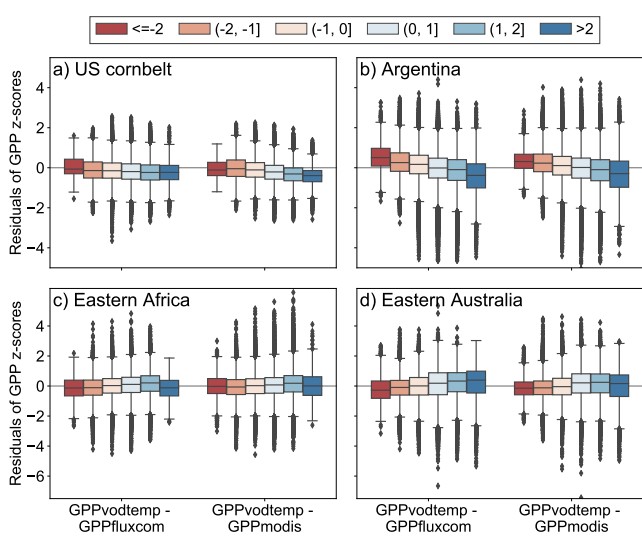

**Figure A8.** Boxplot of residuals between standardized GPP values of GPPvodtemp and GPPfluxcom or GPPmodis along SPEI12 categories for the data in Figure 8. The intervals for the different SPEI12 categories are given in the legend. Box whiskers indicate 1.5 of the interquartile range. The analysis is based on the whole study period (2003-2015).

**Table A1.** Overview of FLUXNET Tier1 v1 stations within the period 2003 to 2014. Land cover from IGBP (International Geosphere–Biosphere Programme) is obtained from the FLUXNET station metadata. Land cover abbreviations and number of stations per land cover class sorted by station number: ENF (Evergreen Needleleaf Forests; 23), GRA (Grasslands; 22), DBF (Deciduous Broadleaf Forests; 14), CRO (Croplands; 11), EBF (Evergreen Broadleaf Forests; 9), WET (Permanent Wetlands; 9), OSH (Open Shrublands; 7), MF (Mixed Forests; 6), SAV (Savannas; 6), WSA (Woody Savannas; 4) and CSH (Closed Shrublands; 1).

| FLUXNET-ID | Name | Lon [° E] | Lat [° N] | Years used | Land cover |
|---|---|---|---|---|---|
| AR-SLu | San Luis | -66.46 | -33.46 | 2009-2011 | MF |
| AR-Vir | Virasoro | -56.19 | -28.24 | 2010-2012 | ENF |
| AT-Neu | Neustift | 11.32 | 47.12 | 2003-2012 | GRA |
| AU-ASM | Alice Springs | 133.25 | -22.28 | 2010-2013 | ENF |
| AU-Ade | Adelaide River | 131.12 | -13.08 | 2007-2009 | WSA |
| AU-Cpr | Calperum | 140.59 | -34.00 | 2010-2013 | SAV |
| AU-Cum | Cumberland Plains | 150.72 | -33.61 | 2012-2013 | EBF |
| AU-DaP | Daly River Savanna | 131.32 | -14.06 | 2008-2013 | GRA |
| AU-DaS | Daly River Cleared | 131.39 | -14.16 | 2008-2013 | SAV |
| AU-Dry | Dry River | 132.37 | -15.26 | 2008-2013 | SAV |
| AU-Emr | Emerald, Queensland, Australia | 148.47 | -23.86 | 2011-2013 | GRA |
| AU-Fog | Fogg Dam | 131.31 | -12.55 | 2006-2008 | WET |
| AU-GWW | Great Western Woodlands, Western Australia, Australia | 120.65 | -30.19 | 2013-2014 | SAV |
| AU-RDF | Red Dirt Melon Farm, Northern Territory | 132.48 | -14.56 | 2011-2013 | WSA |
| AU-Rig | Riggs Creek | 145.58 | -36.65 | 2011-2013 | GRA |
| AU-Rob | Robson Creek, Queensland, Australia | 145.63 | -17.12 | 2014-2014 | EBF |
| AU-Tum | Tumbarumba | 148.15 | -35.66 | 2003-2013 | EBF |
| AU-Whr | Whroo | 145.03 | -36.67 | 2011-2013 | EBF |
| BE-Bra | Brasschaat | 4.52 | 51.31 | 2004-2013 | MF |
| BE-Lon | Lonzee | 4.75 | 50.55 | 2004-2014 | CRO |
| BE-Vie | Vielsalm | 6.00 | 50.31 | 2003-2014 | MF |
| BR-Sa3 | Santarem-Km83-Logged Forest | -54.97 | -3.02 | 2003-2004 | EBF |
| CA-NS1 | UCI-1850 burn site | -98.48 | 55.88 | 2003-2005 | ENF |
| CA-NS3 | UCI-1964 burn site | -98.38 | 55.91 | 2003-2005 | ENF |
| CA-NS4 | UCI-1964 burn site wet | -98.38 | 55.91 | 2003-2005 | ENF |
| CA-NS5 | UCI-1981 burn site | -98.49 | 55.86 | 2003-2005 | ENF |
| CA-NS6 | UCI-1989 burn site | -98.96 | 55.92 | 2003-2005 | OSH |
| CA-NS7 | UCI-1998 burn site | -99.95 | 56.64 | 2003-2005 | OSH |
| CA-Qfo | Quebec - Eastern Boreal, Mature Black Spruce | -74.34 | 49.69 | 2003-2010 | ENF |

| FLUXNET-ID | Name | Lon [° E] | Lat [° N] | Years used | Land cover |
|---|---|---|---|---|---|
| CA-SF1 | Saskatchewan - Western Boreal, forest burned in 1977 | -105.82 | 54.49 | 2003-2006 | ENF |
| CA-SF2 | Saskatchewan - Western Boreal, forest burned in 1989 | -105.88 | 54.25 | 2003-2005 | ENF |
| CA-SF3 | Saskatchewan - Western Boreal, forest burned in 1998 | -106.01 | 54.09 | 2003-2006 | OSH |
| CH-Cha | Chamau | 8.41 | 47.21 | 2006-2012 | GRA |
| CH-Fru | Früebüel | 8.54 | 47.12 | 2006-2012 | GRA |
| CH-Oe1 | Oensingen grassland | 7.73 | 47.29 | 2003-2008 | GRA |
| CN-Cha | Changbaishan | 128.10 | 42.40 | 2003-2005 | MF |
| CN-Cng | Changling | 123.51 | 44.59 | 2007-2010 | GRA |
| CN-Dan | Dangxiong | 91.07 | 30.50 | 2004-2005 | GRA |
| CN-Din | Dinghushan | 112.54 | 23.17 | 2003-2005 | EBF |
| CN-Du2 | Duolun_grassland (D01) | 116.28 | 42.05 | 2006-2008 | GRA |
| CN-Ha2 | Haibei Shrubland | 101.33 | 37.61 | 2003-2005 | WET |
| CN-HaM | Haibei Alpine Tibet site | 101.18 | 37.37 | 2003-2004 | GRA |
| CN-Qia | Qianyanzhou | 115.06 | 26.74 | 2003-2005 | ENF |
| CN-Sw2 | Siziwang Grazed (SZWG) | 111.90 | 41.79 | 2010-2012 | GRA |
| CZ-BK1 | Bily Kriz forest | 18.54 | 49.50 | 2003-2008 | ENF |
| CZ-BK2 | Bily Kriz grassland | 18.54 | 49.49 | 2004-2006 | GRA |
| DE-Akm | Anklam | 13.68 | 53.87 | 2009-2014 | WET |
| DE-Gri | Grillenburg | 13.51 | 50.95 | 2004-2014 | GRA |
| DE-Hai | Hainich | 10.45 | 51.08 | 2003-2012 | DBF |
| DE-Kli | Klingenberg | 13.52 | 50.89 | 2004-2014 | CRO |
| DE-Lkb | Lackenberg | 13.30 | 49.10 | 2009-2013 | ENF |
| DE-Obe | Oberbärenburg | 13.72 | 50.78 | 2008-2014 | ENF |
| DE-RuS | Selhausen Juelich | 6.45 | 50.87 | 2011-2014 | CRO |
| DE-Spw | Spreewald | 14.03 | 51.89 | 2010-2014 | WET |
| DE-Tha | Tharandt | 13.57 | 50.96 | 2003-2014 | ENF |
| DK-NuF | Nuuk Fen | -51.39 | 64.13 | 2008-2014 | WET |
| DK-Sor | Soroe | 11.64 | 55.49 | 2003-2012 | DBF |
| DK-ZaH | Zackenberg Heath | -20.55 | 74.47 | 2003-2008 | GRA |
| ES-LgS | Laguna Seca | -2.97 | 37.10 | 2007-2009 | OSH |
| ES-Ln2 | Lanjaron-Salvage logging | -3.48 | 36.97 | 2009-2009 | OSH |
| FI-Hyy | Hyytiala | 24.30 | 61.85 | 2003-2014 | ENF |
| FI-Jok | Jokioinen | 23.51 | 60.90 | 2003-2003 | CRO |
| FR-Gri | Grignon | 1.95 | 48.84 | 2004-2013 | CRO |

| FLUXNET-ID | Name | Lon [° E] | Lat [° N] | Years used | Land cover |
|---|---|---|---|---|---|
| FR-Pue | Puechabon | 3.60 | 43.74 | 2003-2013 | EBF |
| GF-Guy | Guyaflux (French Guiana) | -52.92 | 5.28 | 2004-2012 | EBF |
| IT-CA1 | Castel d'Asso 1 | 12.03 | 42.38 | 2011-2013 | DBF |
| IT-CA2 | Castel d'Asso 2 | 12.03 | 42.38 | 2011-2013 | CRO |
| IT-CA3 | Castel d'Asso 3 | 12.02 | 42.38 | 2011-2013 | DBF |
| IT-Cp2 | Castelporziano 2 | 12.36 | 41.70 | 2012-2013 | EBF |
| IT-Isp | Ispra ABC-IS | 8.63 | 45.81 | 2013-2014 | DBF |
| IT-Lav | Lavarone | 11.28 | 45.96 | 2003-2012 | ENF |
| IT-Noe | Arca di Noé - Le Prigionette | 8.15 | 40.61 | 2004-2012 | CSH |
| IT-PT1 | Parco Ticino forest | 9.06 | 45.20 | 2003-2004 | DBF |
| IT-Ren | Renon | 11.43 | 46.59 | 2003-2013 | ENF |
| IT-Ro1 | Roccarespampani 1 | 11.93 | 42.41 | 2003-2008 | DBF |
| IT-Ro2 | Roccarespampani 2 | 11.92 | 42.39 | 2003-2012 | DBF |
| IT-SR2 | San Rossore 2 | 10.29 | 43.73 | 2013-2014 | ENF |
| IT-SRo | San Rossore | 10.28 | 43.73 | 2003-2012 | ENF |
| IT-Tor | Torgnon | 7.58 | 45.84 | 2008-2013 | GRA |
| JP-MBF | Moshiri Birch Forest Site | 142.32 | 44.39 | 2003-2005 | DBF |
| JP-SMF | Seto Mixed Forest Site | 137.08 | 35.26 | 2003-2006 | MF |
| NL-Hor | Horstermeer | 5.07 | 52.24 | 2004-2011 | GRA |
| NL-Loo | Loobos | 5.74 | 52.17 | 2003-2013 | ENF |
| NO-Adv | Adventdalen | 15.92 | 78.19 | 2012-2014 | WET |
| RU-Che | Cherski | 161.34 | 68.61 | 2003-2005 | WET |
| RU-Cok | Chokurdakh | 147.49 | 70.83 | 2003-2013 | OSH |
| RU-Fyo | Fyodorovskoye | 32.92 | 56.46 | 2003-2013 | ENF |
| RU-Ha1 | Hakasia steppe | 90.00 | 54.73 | 2003-2004 | GRA |
| SD-Dem | Demokeya | 30.48 | 13.28 | 2005-2009 | SAV |
| US-AR1 | ARM USDA UNL OSU Woodward Switchgrass 1 | -99.42 | 36.43 | 2009-2012 | GRA |
| US-AR2 | ARM USDA UNL OSU Woodward Switchgrass 2 | -99.60 | 36.64 | 2009-2012 | GRA |
| US-ARM | ARM Southern Great Plains site- Lamont | -97.49 | 36.61 | 2003-2012 | CRO |
| US-Blo | Blodgett Forest | -120.63 | 38.90 | 2003-2007 | ENF |
| US-Ha1 | Harvard Forest EMS Tower (HFR1) | -72.17 | 42.54 | 2003-2012 | DBF |
| US-Los | Lost Creek | -89.98 | 46.08 | 2003-2014 | WET |
| US-MMS | Morgan Monroe State Forest | -86.41 | 39.32 | 2003-2014 | DBF |
| US-Me6 | Metolius Young Pine Burn | -121.61 | 44.32 | 2010-2012 | ENF |

| FLUXNET-ID | Name | Lon [° E] | Lat [° N] | Years used | Land cover |
|---|---|---|---|---|---|
| US-Myb | Mayberry Wetland | -121.77 | 38.05 | 2011-2014 | WET |
| US-Ne1 | Mead - irrigated continuous maize site | -96.48 | 41.17 | 2003-2013 | CRO |
| US-Ne2 | Mead - irrigated maize-soybean rotation site | -96.47 | 41.16 | 2003-2013 | CRO |
| US-Ne3 | Mead - rainfed maize-soybean rotation site | -96.44 | 41.18 | 2003-2013 | CRO |
| US-SRM | Santa Rita Mesquite | -110.87 | 31.82 | 2004-2014 | WSA |
| US-Syv | Sylvania Wilderness Area | -89.35 | 46.24 | 2003-2014 | MF |
| US-Ton | Tonzi Ranch | -120.97 | 38.43 | 2003-2014 | WSA |
| US-Tw3 | Twitchell Alfalfa | -121.65 | 38.12 | 2013-2014 | CRO |
| US-UMd | UMBS Disturbance | -84.70 | 45.56 | 2007-2014 | DBF |
| US-Var | Vaira Ranch- Ione | -120.95 | 38.41 | 2003-2014 | GRA |
| US-WCr | Willow Creek | -90.08 | 45.81 | 2003-2014 | DBF |
| US-Whs | Walnut Gulch Lucky Hills Shrub | -110.05 | 31.74 | 2007-2014 | OSH |
| US-Wkg | Walnut Gulch Kendall Grasslands | -109.94 | 31.74 | 2004-2014 | GRA |
| ZA-Kru | Skukuza | 31.50 | -25.02 | 2003-2010 | SAV |
| ZM-Mon | Mongu | 23.25 | -15.44 | 2007-2009 | DBF |

**Table A2.** Leave-site-out cross validation for GPPvodtemp and GPPvod. The analysis was conducted for the full signal as well as for the anomalies from the mean seasonal cycle. Anomalies were calculated after model application. Values represent mean and standard deviation of the metrics over the cross validation results for each site.

| | Pearson r [-] | UbRMSE [gC m$^{-2}$ d$^{-1}$] | Bias [gC m$^{-2}$ d$^{-1}$] |
|---|---|---|---|
| GPPvod | 0.40 ± 0.32 | 2.57 ± 1.14 | -0.04 ± 2.01 |
| GPPvodtemp | 0.54 ± 0.31 | 2.30 ± 1.01 | -0.08 ± 2.01 |
| GPPvod anomalies | 0.18 ± 0.22 | 1.57 ± 0.78 | -0.00 ± 0.00 |
| GPPvodtemp anomalies | 0.22 ± 0.19 | 1.53 ± 0.76 | 0.00 ± 0.00 |

*Author contributions.* IT conceived the study, carried out the analysis and drafted the manuscript with contributions from WD and MF on
the study design. BW contributed to data preparation. LM provided VOD estimates from VODCA. All authors discussed the results and commented on the manuscript.

*Competing interests.* The authors declare that they have no conflict of interest.

*Acknowledgements.* Wouter Dorigo and Benjamin Wild are supported through the EOWAVE project funded by the TU Wien Wissenschaftspreis 2015, which was awarded to Wouter Dorigo (http://climers.geo.tuwien.ac.at/eowave/). The authors acknowledge the TU Wien Bibliothek for
financial support through its Open Access Funding Program. This work used eddy covariance data acquired and shared by the FLUXNET community, including these networks: AmeriFlux, AfriFlux, AsiaFlux, CarboAfrica, CarboEuropeIP, CarboItaly, CarboMont, ChinaFlux, Fluxnet-Canada, GreenGrass, ICOS, KoFlux, LBA, NECC, OzFlux-TERN, TCOS-Siberia, and USCCC. The FLUXNET eddy covariance data processing and harmonization was carried out by the ICOS Ecosystem Thematic Center, AmeriFlux Management Project and Fluxdata project of FLUXNET, with the support of CDIAC, and the OzFlux, ChinaFlux and AsiaFlux offices.

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
