# Peer review of "Impact of temperature and water availability on microwave-derived gross primary production"

_Biogeosciences, 2020_

## Referee Comment (RC1) · Jean-Pierre Wigneron (Referee) · 18 Dec 2020

Impact of temperature and water availability on microwave-derived gross primary production by Irene E. Teubner

This study evaluates the capabilities of VOD to provide new information on the changes in vegetation productivity at a global scale. Specific improvements obtained by accounting for temperature effects on autotrophic respiration are analyzed.

I found that interesting results are presented. However, I think significant improvements should be made. As I am not familiar with the studies by the authors on this topic, I found it is difficult to understand many points in this manuscript, unless, maybe, I read in detail all papers published before. Basic elements of the modeling approach

published before should be given here, so that the paper is "more autonomous". I present below many points to be improved, so that readers who are not familiar with the papers published by the authors, may understand the results and the discussion

I have 4 main comments which should be accounted for before publication

1) I think, the lack of improvement in the tropics could be related to the low sensitivity of X-VOD to biomass changes, which was found in many regions of the world but particularly in the tropics. This should be better discussed and accounted for throughout the manuscript.

For instance, many references discussing the capabilities of VOD to monitor biomass are missing. Cf below references on this topic including applications on biomass changes / productivity monitoring, to better account for and reflect the published literature on this topic (Brandt et al., 2018, Fan et al., 2019; Al -Yaari et al., 2020; Lei et al., 2020; Frappart et al., 2020).

line 14: "regions outside the tropics", many studies have shown that X-band and C-band VOD present saturation vs biomass (close to 200 t/ha). So, how do you expect to monitor GPP from VOD indices that saturate over dense vegetation forests, which represent a large fraction of the vegetation cover in those regions.

line 72-74, I did not review these previous papers, but I think it is quite surprising that X-VOD provide best agreement with GPP by considering "sink terms" related to biomass changes. X-VOD is better related to LAI/NDVI (and thus to photosynthesis and "source terms"), while L-VOD is better related to biomass changes (see Li et al., 2020).

Still based on L-VOD, Tian et al., 2018, found a decoupling between seasonal changes in VOD and in the leafy/biomass component in dry tropical forests. This should explain some errors too in the tropics, when attempting to relate VOD changes to vegetation productivity (?)

2) I found it is very difficult to understand section 2.2, except if you are an expert in

this specific modelling approach Temperature is an important parameter in Ra but also in other key processes such as photosynthesis. How can you be sure that only the Ra(T) dependence was accounted for here? Because I cannot see any deterministic equations relating Ra to temperature: in Eq 1 and 2, is it fully a machine learning approach that you used, isn't it?

As I'm not expert of this kind of regressions and many terms are unclear to me. Maybe, it is very specific to me, but maybe it will apply to many other readers: Better explain what is "VOD time series", "delta VOD", "mdn VOD": over which time step? considering daily , monthly or yearly values? do you compute mean of delta VOD, etc...? What is the time step of Eq 1 & 2: daily? "spline terms for representing 2-dim functions": what do you mean? which 2-dim parameters are considered here? "smoothing factor"?, etc.

3) Did GPP-VOD-temp showed improvements vs GPP-VOD, when considering correlation of residuals with SPEI? Since the present study focuses on analyzing possible improvement of the new GPP-VOD-temp, intercomparing residuals vs SPEI with GPP-VOD is key and should be added in this manuscript. The present description of results is a bit lengthy and should be reduced to the profit of the above inter-comparison.

4) I found that conclusions are much more nuanced considering the relative improvement obtained with the new GPP-VOD-temp product. This should be better reflected in the abstract which I found too optimistic.

Minor line 16- 20; it seems to me the two sentences are a bit contradictory line 25-30, Cf above remarks on C- and X-VOD saturation, many papers were published based on SMOS L-VOD and none is mentioned in this short review. This short review should be more "opened" line 35: "VOD as a proxy of AGB": I guess very few FLUXNET sites are available in relatively dense vegetation sites, and more generally in the tropics. The VOD-derived GPP is manly calibrated based on data in temperate climate? Line 43: " as a necessity": what do you mean here; not so convincing as a scientific term. Lien 50 define what is Q10? Line 94: why not using FLUXCOM RS + Meteo , which

has more input and could be more reliable. I do not understand the reason given here "our approach is mainly based on RS..". This is a not a good reason to me (?) Please provide more information on Fluxnet sites used here (maps of locations, main vegetation types, etc.) Fig. 1 saturation for VOD > 0.6, can this be related to the saturation of the VOD / Biomass relationship? (0.6 corresponds very well to the saturation level of X-band) Line 179, add that you consider 8-daily values. Line 218 "increase of" ? -check grammar in line 223-225. -Figure 3 and 4: is this based on the whole study period (please add the information caption and check throuhout)? -Figure 4, the overestimation in the tropics seems to be much more significant than the very small decrease outside -35°, +60°. Can we really consider this is "improvement"? -add site for Fig 5 in the Fluxnet map. Why selecting this site: is it representative of more general results (specific canopy types, climate?)? -line 239; "holds true" is too strong; here you only find no contradiction on a specific point; mathematically it is not right at all to say that the hypothesis is validated. It is only one indication you go the right way...

References: Li, X. et al.., Global-scale assessment and inter-comparison of recently developed / reprocessed microwave satellite vegetation optical depth products, Remote Sens. Env., DOI: 10.1016/j.rse.2020.112208

Al‐Yaari, et al. Asymmetric responses of ecosystem productivity to rainfall anomalies vary inversely with mean annual rainfall over the conterminous U.S, Global Change Biology, 00:1–15, https://doi.org/10.1111/gcb.15345.

Tian, F., et al. "Coupling of ecosystem-scale plant water storage and leaf phenology observed by satellite", Nature Ecology & Evolution, Vol 2, Spt. 2018, 1428–1435, https://doi.org/10.1038/s41559-018-0630-3, 2, 1428–1435, 2018.

Brandt M., "Satellite passive microwaves reveal recent climate-induced carbon losses in African drylands", Nature Ecology and Evolution., https://doi.org/10.1038/s41559-018-0530-6, 2, 827–835 2018.

Fan L., et al., Satellite observed pantropical carbon dynamics, Nature Plants, 5, 944–951, July 2019, https://doi.org/10.1038/s41477-019-0478-9.

Frappart F., et al., Global monitoring of the vegetation dynamics from the Vegetation Optical Depth (VOD): a review, Remote Sensing, 12, 2915, 2020 doi:10.3390/rs12182915

---

## Referee Comment (RC2) · David Chaparro (Referee) · 13 Jan 2021

This work presents a model to estimate Gross Primary Production (GPP) globally from a carbon sink driven approach. In particular, the paper aims at improving previous modelling of GPP as a function of the vegetation optical depth (VOD; Teubner et al., 2019) by including the effect of temperature on the autotrophic respiration. Authors explain that the model is based on the fact that VOD is a good proxy of above-ground biomass (AGB). The link between residuals of the model and the drought index SPEI is also analysed. The results presented show an improvement of model performance in terms of temporal dynamics, especially in non-tropical regions. Interestingly, results also report that the presented model does not require complementary information from precipitation or drought indicators.

[Figure]

Despite that the results presented are consistent with previous works and show interesting contribution to GPP modelling, I have important concerns that have to be addressed before publication. The most relevant are related to the lack of penetration through the vegetation canopy of the X-band VOD (which is not a good proxy of biomass if compared to other frequency bands), and to the need of further explaining the modelling framework both in the introduction and the methods sections. These comments and other major and minor proposals for improving the paper are detailed hereafter:

**Major comments**

1. Although the paper can be well understood if the reader knows previous literature on this topic published by the authors, it is necessary that the modelling approach (i.e., main ideas and equations from previous works) and the implementation in the current paper are explained in more detail. In particular:

a. I suggest that first paragraphs of the introduction provide a more detailed description of the framework explained in Teubner et al. (2019), including if necessary some equations (e.g., equations 4 to 6 in Teubner et al., 2019).

b. Please, provide more detail on how you are computing and including into the model the different variables (Section 2.2). For instance, does the term "VOD" refer to VOD time-series? If so, how is the time-domain processed (raw data, smoothing, etc...)? How is the variable computed?

In summary, please extend the text to provide enough information for readers that do not know your previous work.

2. The basis of the work is the fact that VOD is a good proxy of AGB. Nevertheless, it is very important to note that X-band VOD (hereafter XVOD) has poor capacity to penetrate the vegetation canopy, and therefore it is very limited to accurately track AGB in regions with dense vegetation. While the AGB - L-band VOD relationship shows low

saturation in tropical regions, X-band is not the most appropriate frequency to be used in these areas as a proxy of biomass (e.g., Brandt et al., 2018; Rodríguez-Fernández et al., 2018; Chaparro et al., 2019). Actually, even in low carbon density areas, XVOD is more representative of vegetation cover than it is for biomass, while lower frequencies (L-band VOD; hereafter LVOD) still have improved capacity to track AGB in these areas (e.g., see Fig. 9 in Chaparro et al., 2019).

It is very likely that this limitation explains the lack of improvement of the model in the tropical regions (Fig. 3b) and the low correlation between the model and the benchmark datasets in regions such as the Amazon (Fig. 3a). In addition, this could also explain the saturation of the partial dependency plots (Fig. 1) at high VOD and T2M values (darkest lines in the first panel, probably representing vegetation-temperature conditions in the tropics) and at mdnVOD values above 0.4 (i.e., dense vegetation; third panel).

The application of XVOD is justified in the paper by the higher correlation between XVOD and GPP if compared to the LVOD-GPP correlation (Teubner et al., 2018). Nevertheless, it is important to note that the GPP benchmark datasets have an important contribution from visible-infrared (VIS-IR) indices (EVI, LAI, MIR, NDVI and NDWI, as stated in l. 96). I think it is expected that GPP datasets based on VIS-IR indices show a greater correlation with XVOD than with lower frequency VOD data, because both of them capture the same layer of vegetation (top of the canopy). In contrast, I would expect greater correlations with in situ GPP FluxNet data (Fig. 2a and Fig. 2b) if GPPvod and GPPvodtemp were computed using L-band VOD. Although it is not a global dataset, FluxNet in situ information is not conditioned by physical properties of remote sensing sensor measurements, so it is probably the most "independent" tool the authors have for evaluating the accuracy of the GPP estimates.

For all these reasons, it would be very interesting if the authors include new GPPvod and GPPvodtemp models based on L-band VOD and validate their accuracy using in situ FluxNet data (i.e., including them in Fig. 2). They could show (and compare) the

resulting GPP estimates between different frequencies and, importantly to preserve the scope of the paper, between GPPvod and GPPvodtemp models. However, I am aware that this could move the work slightly beyond its initial scope, as it adds another factor (i.e., frequencies) in the comparisons. I encourage the authors to work on this possibility, although it is up to them to finally incorporate this change or to keep only XVOD in the paper. In any case, they must discuss all the possible implications of using XVOD. Within the discussion, they should address at least the following points/questions:

- It is stated that "the VOD-GPP model relies on estimating carbon sink terms, [. . .], based on VOD as a proxy of aboveground living biomass" (l. 35-37). To what point is this true, according to the facts that XVOD is more representative of vegetation cover than of biomass, and that lower VOD frequencies have enhanced capacities to capture biomass? Please clearly explain the possible limitations of the approach.

- Please, discuss about the saturation effects in Fig. 1 (first and third panel; see my comment above). Are they likely to be linked to XVOD saturation in the tropics? If so, which are the implications?

- In l. 197, it is mentioned (referring to tropical regions) that "[in these regions] sensitivity to temperature is also low, which makes the interaction term mainly controlled by VOD." If I correctly understood plots in Fig. 1, would it be more precise to affirm that it is mostly controlled by , as other dependencies (VOD, mdnVOD) saturate in tropical regions?

3. The GPP estimates (GPPvod and GPPvodtemp) are calibrated using FluxNet in situ data. Also, both FluxNet and FLUXCOM data (an upscaling of FluxNet) are used as reference datasets for evaluating GPP estimates. I think that, consequently, reference datasets may not be fully independent from GPP estimates. To what extent? Which is the contribution of FluxNet data in the reference datasets and in the estimates? This has to be acknowledged and possible implications discussed.

In addition, authors should try to guarantee at least one year of "fully independent" comparison between estimates and FluxNet/FLUXCOM data. I suggest they could

calibrate the model by leaving one year of data apart (e.g., use 2004-2014 for calibration) and apply the remaining data (2003) for fully independent comparisons. They can show these new comparisons in supplementary materials and refer to them to show consistency/inconsistency with the full-period comparisons of Figures 2 to 6.

**Minor comments**

- Lines 5-6: "VOD-GPP model generally showed good agreement" → Please quantify (e.g. correlation coefficient).

- L. 6: "tended to overestimate" → By how much? Please quantify.

- L. 13: "Our results reveal an improvement" → Please quantify this improvement (e.g., increase on the average correlation).

- L. 14: "This increase in temporal dynamic" → "This improvement in temporal dynamics."

- L. 19: can you mention which are these regions?

- L. 19: "between [. . .] with" → "between [. . .] and"

- L. 25: provide → provides

- L. 25 to 30: you may want to include other references which are explicitly linked to water content: e.g., Feldman et al., 2018; Tian et al., 2018.

- L. 28: Chaparro et al., 2019 → Chaparro et al., 2018 (this is different from the "carbon-stocks work" in Chaparro et al., 2019). Add the new reference to the references list if you want to keep it in the text.

- L. 70: maybe saying "only a few years" is a bit excessive (e.g., SMOS spans >10 years). Try using another expression, please.

- L. 83-85: "During data processing. . ." → Please move these lines to the methods section.
- L. 118: "T2M was used in our analysis, since this parameter is most common for describing the temperature dependency" → Please add some references to show the common use of this variable.

- L. 121-122: "aggregated to 8-daily estimates" → Please, specify that this is done to match MODIS time-steps in case it was your intention.

- L. 162: "savitzky-golay" → "Savitzky-Golay."

- L. 170: "are consistent the previous" → "Are consistent with the previous."

- Figure 1: Please add marginal distribution rug plots to each panel. Name the panels as "a" to "d" and complete the figure caption with explanation of each panel.

- Figure 2: add GPPvod as another panel for comparison with GPPvodtemp and benchmark data. Also, I find that the blue to yellow colorbar shows very low contrasts. To me, it is difficult to appreciate color gradients in the figure. You could use other colors (e.g., blue to red?) or saturate the colorbar at the (e.g., 95th, 99th) percentile to improve contrast.

- L. 218: "For a region in Europe" → Please add coordinates also here, as well as the general situation (e.g., "Central Europe") to help the reader.

- L. 218: "increase" → "increase in".

- L. 227-229: there is no a verb in this sentence, please rephrase.

- L. 233: "between [. . .] with" → "between [. . .] and".

- Figures 4 and 6: please define in the text what are "zonal means".

- Figures 5 and 8: please, could you make each panel wider? Then there is more place for seeing interannual variability in the figures.

- Figure 6: Add GPPvod as another panel for comparison with GPPvodtemp and benchmark data.

- L. 236: "given that correlations in these regions are high" → Authors probably mean correlations between GPPvodtemp and benchmark data. Please specify this, because as you explained correlations of residuals in the previous sentence, it can be confusing.

- Figure 7: to improve the boxes for regions, you could use colors different than those from the colorbar (e.g., light green instead of blue?).

- L. 282: "increase" → Do you mean "improvement"?

- L. 338: "between with" → "with".

- Figure A1: please add GPPvod as well as a map of the median differences between GPPvod and GPPvodtemp. Also, note that the contrast in the blue to yellow colorbar could be improved, or colors changed to a blue-red scale.

- Figure A3: please add GPPvod.

- Figure A4: this seems an interesting result, but I do not fully understand what do you mean by "scaled latitudinal" distribution. Could you please explain this?

- Table A1: it could be useful to detail the dominant land cover in each station. Then, the reader will be able to see which vegetation types have been included in calibration of GPP estimates.

**References**

Brandt, M., Wigneron, J. P., Chave, J., Tagesson, T., Penuelas, J., Ciais, P., ... Fensholt, R. (2018). Satellite passive microwaves reveal recent climate-induced carbon losses in African drylands. Nature ecology evolution, 2(5), 827-835.

Chaparro, D., Piles, M., Vall-Llossera, M., Camps, A., Konings, A. G., Entekhabi, D. (2018). L-band vegetation optical depth seasonal metrics for crop yield assessment. Remote sensing of environment, 212, 249-259.

Chaparro, D., Duveiller, G., Piles, M., Cescatti, A., Vall-Llossera, M., Camps, A., Entekhabi, D. (2019). Sensitivity of L-band vegetation optical depth to carbon stocks in tropical forests: a comparison to higher frequencies and optical indices. Remote sensing of environment, 232, 111303.

Feldman, A. F., Gianotti, D. J. S., Konings, A. G., McColl, K. A., Akbar, R., Salvucci, G. D., Entekhabi, D. (2018). Moisture pulse-reserve in the soil-plant continuum observed across biomes. Nature plants, 4(12), 1026-1033.

Rodríguez-Fernández, N. J., Mialon, A., Mermoz, S., Bouvet, A., Richaume, P., Al Bitar, A., ... Wigneron, J. P. (2018). An evaluation of SMOS L-band vegetation optical depth (L-VOD) data sets: high sensitivity of L-VOD to above-ground biomass in Africa. Biogeosciences, 15(14), 4627-4645.

Teubner, I. E., Forkel, M., Jung, M., Liu, Y. Y., Miralles, D. G., Parinussa, R., ... Dorigo, W. A. (2018). Assessing the relationship between microwave vegetation optical depth and gross primary production. International journal of applied earth observation and geoinformation, 65, 79-91.

Teubner, I. E., Forkel, M., Camps-Valls, G., Jung, M., Miralles, D. G., Tramontana, G., ... Dorigo, W. A. (2019). A carbon sink-driven approach to estimate gross primary production from microwave satellite observations. Remote Sensing of Environment, 229, 100-113.

Tian, F., Wigneron, J. P., Ciais, P., Chave, J., Ogée, J., Penuelas, J., ... Fensholt, R. (2018). Coupling of ecosystem-scale plant water storage and leaf phenology observed by satellite. Nature ecology evolution, 2(9), 1428-1435.

---

## Author Comment (AC1) · 2 Feb 2021

Impact of temperature and water availability on microwave-derived gross primary production

by Irene E. Teubner

This study evaluates the capabilities of VOD to provide new information on the changes in vegetation productivity at a global scale. Specific improvements obtained by accounting for temperature effects on autotrophic respiration are analyzed.

I found that interesting results are presented. However, I think significant improvements should be made. As I am not familiar with the studies by the authors on this topic, I found it is difficult to understand many points in this manuscript, unless, maybe, I read in detail all papers published before. Basic elements of the modeling approach published before should be given here, so that the paper is "more autonomous". I present below many points to be improved, so that readers who are not familiar with the papers published by the authors, may understand the results and the discussion

Response: Dear Jean-Pierre Wigneron,

many thanks for your detailed and very constructive review. We agree with you that the manuscript is hard to understand without knowledge about our previous publications. Therefore, we will include the main equations and assumptions from Teubner et al. (2019) in the revised version of the manuscript.

I have 4 main comments which should be accounted for before publication

1) I think, the lack of improvement in the tropics could be related to the low sensitivity of X-VOD to biomass changes, which was found in many regions of the world but particularly in the tropics. This should be better discussed and accounted for throughout the manuscript.

Response: We agree that X-band VOD is less sensitive to changes in total biomass in the tropical forests, since L-band VOD has been shown to be less prone to saturation at high biomass values (e.g., Chaparro et al., 2019; Li et al., 2021). This may thus suggest that the use of L-band VOD would yield better results in estimating GPP than X-band VOD. However, various studies show that X-band VOD generally yields a higher agreement with GPP (Teubner et al., 2018) or proxies for productivity like NDVI (Li et al., 2021) than L-band VOD. This finding that L-band VOD agrees less with GPP than X-band VOD was also observed for the land cover classes with higher biomass, i.e., forest land cover classes (Teubner et al., 2018). The reason for the general higher suitability of X-band VOD for the estimation of GPP in forests and other land cover types with a large woody component can be explained by differences in metabolic activity of various plant parts. Plant parts with a higher metabolic activity, like leaves and roots, are a better proxy for GPP than large structural components like branches and trunks

(Litton et al., 2007). For further details, see discussion section 6.2 in Teubner et al. (2019). In addition, our approach is based on the change in biomass between two successive time steps (mostly a few days). At this short time interval, leaf biomass - to which higher frequency microwaves are more sensitive (Woodhouse, 2005) - is expected to potentially show changes, while woody above-ground components - to which lower frequency microwaves like L-band are more sensitive (Woodhouse, 2005) - may not.

However, in general the correlations between all VOD bands and existing global GPP products are low in evergreen broad-leaved tropical forests (Teubner et al., 2018). This disagreement can be due to multiple reasons. First, also data-driven large-scale products of GPP like FLUXCOM are based on only few station observations in tropical forests and thus have large uncertainty in this region (Jung et al., 2020). Second, the optical remote sensing input data to produce these datasets are highly affected by cloud cover (Zhao et al., 2005).

To make the choice of X-band in our approach clearer, we will add the issue of high- vs low-frequency VOD in the discussion.

For instance, many references discussing the capabilities of VOD to monitor biomass are missing. Cf below references on this topic including applications on biomass changes / productivity monitoring, to better account for and reflect the published literature on this topic (Brandt et al., 2018, Fan et al., 2019; Al -Yaari et al., 2020; Lei et al., 2020; Frappart et al., 2020).

Response: Thank you for these suggestions and we are well aware about these publications. However, some of the papers that you co-authored (Frappart et al., 2020; Lei et al., 2020) were published just before or after the submission of our manuscript and hence we could not include it. We will incorporate these references where appropriate.

line 14: "regions outside the tropics", many studies have shown that X-band and Cband VOD present saturation vs biomass (close to 200 t/ha). So, how do you expect to monitor GPP from VOD indices that saturate over dense vegetation forests, which represent a large fraction of the vegetation cover in those regions.

Response: See our reply above about our previous findings on the correlation between VOD bands and GPP.

line 72-74, I did not review these previous papers, but I think it is quite surprising that XVOD provide best agreement with GPP by considering "sink terms" related to biomass changes. X-VOD is better related to LAI/NDVI (and thus to photosynthesis and "source terms"), while L-VOD is better related to biomass changes (see Li et al., 2020).

Response: X-band VOD is indeed more similar to LAI/NDVI which represent commonly used inputs for source-driven approaches. Although high-frequency VOD is closely related to small vegetation parts like leaves and twigs (Woodhouse, 2005) and therefore closely related to LAI or NDVI, the method can still be regarded as a sink-driven approach. If X-band VOD was better suited for a source-driven approach, the use of VOD alone should be enough for estimating GPP. However, Teubner et al. (2019) could demonstrate that the combination of VOD and change in VOD (computed as the change over consecutive time steps) outperforms the use of each VOD variable taken separately. We think that this performance is related to allometric relationships between the biomass in canopies and biomass in stems. A certain long-term increase in canopy biomass should correspond to an increase in stem diameter and tree height. Hence, we assume that the sensitivity of X-band VOD to the canopy serves also as proxy for changes in total above-ground biomass. The sensitivity of high-frequency VOD to

allometric patterns in vegetation is also supported by the overall medium to high correlation between VOD and canopy height which is partly higher for X-band than for L-band (Li et al., 2020). Hence, we assume that our sink-driven approach is even valid if the used VOD band is not directly sensitive to the changes in total above-ground biomass.

Still based on L-VOD, Tian et al., 2018, found a decoupling between seasonal changes in VOD and in the leafy/biomass component in dry tropical forests. This should explain some errors too in the tropics, when attempting to relate VOD changes to vegetation productivity (?)

Response: Such temporal shifts between VOD and LAI (or GPP) based on optical data may occur and may result in negative correlations between VOD and GPP (Teubner et al., 2018). A potential reason could be that water availability and photosynthesis are negatively correlated in tropical forests (Green et al., 2020), although the hypothesis of increasing photosynthesis under drought conditions in the Amazon is highly controversial (Koren et al., 2018; Huete et al., 2006; Morton et al., 2014). Hence the sensitivity of VOD to the vegetation water content might explain why VOD is negatively correlated to GPP in tropical forests. As we do not specifically account for this, it will have an effect on our estimates of GPP in tropical forests as also discussed before in Teubner et al. (2019).

2) I found it is very difficult to understand section 2.2, except if you are an expert in this specific modelling approach Temperature is an important parameter in Ra but also in other key processes such as photosynthesis. How can you be sure that only the Ra(T) dependence was accounted for here? Because I cannot see any deterministic equations relating Ra to temperature: in Eq 1 and 2, is it fully a machine learning approach that you used, isn't it?

Response: Although our approach is based on a machine learning method, we make use of knowledge about the relationships between VOD and biomass and between biomass and GPP. Equations 1 and 2 present the machine learning formulas. For the temperature dependency of Ra, we did not present a formula per se since it is difficult to express a formula which combines both the instantaneous temperature response (which may not necessarily apply to 8-daily values) and the acclimation effect; an issue which we addressed in the introduction. As described in the methods section, this was also the reason for choosing Generalized Additive Models as modelling approach, since the relationship is not required to be known a piori but instead estimated from the input data (Hastie and Tibshirani, 1987). We will improve the description of our method in the revised version of the manuscript by including the formulas for GPP, on which our approach is based on.

As I'm not expert of this kind of regressions and many terms are unclear to me. Maybe, it is very specific to me, but maybe it will apply to many other readers: Better explain what is "VOD time series", "delta VOD", "mdn VOD": over which time step? Considering daily , monthly or yearly values? do you compute mean of delta VOD, etc: : :? What is the time step of Eq 1 & 2: daily? "spline terms for representing 2-dim functions": what do you mean? which 2-dim parameters are considered here? "smoothing factor"?, etc.

Response: Following our approach described in Teubner et al. (2019), the VOD time series are the VOD values that were aggregated to 8-daily time steps, ΔVOD is the change of VOD over consecutive time steps of the smoothed 8-daily VOD time series and mdnVOD is the median of the VOD time series computed over all time steps of the grid cell. The term "2-dimensional" refers to fact that the partial dependency for the spline-terms describe a 2-dimensional function between input and response variable. In contrast, the partial dependency for the tensor-term (with two input variables) is represented by a surface in the 3-dimensional space; the three dimensions being the two input

variables and one response variable. In Generalized Additive Models, a number of spline functions are fitted between the input data of the respective term and the response variable; and the resulting partial dependency function is further smoothed (Hastie and Tibshirani, 1987; Servén and Brummitt, 2018). The strength of the smoothing is controlled via the smoothing factor lambda. We will add more details for the mentioned terms in the revised version.

3) Did GPP-VOD-temp showed improvements vs GPP-VOD, when considering correlation of residuals with SPEI? Since the present study focuses on analyzing possible improvement of the new GPP-VOD-temp, intercomparing residuals vs SPEI with GPPVOD is key and should be added in this manuscript. The present description of results is a bit lengthy and should be reduced to the profit of the above inter-comparison.

Response: We will conduct the suggested analysis for the revised manuscript and place it (where appropriate) either in the results or the supplement.

4) I found that conclusions are much more nuanced considering the relative improvement obtained with the new GPP-VOD-temp product. This should be better reflected in the abstract which I found too optimistic.

Response: We will revise the abstract accordingly.

Minor

line 16- 20; it seems to me the two sentences are a bit contradictory

Response: Thank you for this comment. It was meant that the analysis reveals that the residuals largely do not correlate with SPEI, which indicates that the relationship largely is reliable with respect to variations in water availability. Exceptions from this rule are found in some areas which may point towards specific plant properties. We will clarify this in the revised version.

line 25-30, Cf above remarks on C- and X-VOD saturation, many papers were published based on SMOS L-VOD and none is mentioned in this short review. This short review should be more "opened"

Response: As the focus of our study is the estimation of GPP, we did not specially include papers on the relation between VOD and biomass. But we are open to add references in order to improve the argumentation.

line 35: "VOD as a proxy of AGB": I guess very few FLUXNET sites are available in relatively dense vegetation sites, and more generally in the tropics. The VOD-derived GPP is manly calibrated based on data in temperate climate?

Response: The spatial distribution of in situ stations may indeed be an issue which we therefore already addressed in the discussion (Lines 305-307). However, we will revise these lines to make it clearer.

Line 43: " as a necessity": what do you mean here; not so convincing as a scientific term.

Response: It is meant that dry cells, which do not contain water, will not contribute to the estimation of respiration, since these cells are not living. We will make this clearer in the revised version.

Lien 50 define what is Q10?

Response: The definition of Q10 is given in Lines 51-52.

Line 94: why not using FLUXCOM RS + Meteo , which has more input and could be more reliable. I do not understand the reason given here "our approach is mainly based on RS..". This is a not a good reason to me (?)

Response: In contrast to the FLUXCOM RS setup, the RS + Meteo setup includes only the mean seasonal cycle of remote sensing data as input but does not use any information from remote sensing data that could be sensitive to inter-annual changes such as under dry conditions. Our approach mainly relies on remote sensing data, namely VOD estimates. For a fair comparison, we thus choose the FLUXCOM RS setup.

Please provide more information on Fluxnet sites used here (maps of locations, main vegetation types, etc.)

Response: A map of FLUXNET stations is already given in the Appendix (Figure A2). We will add information on the vegetation type in the overview of the FLUXNET stations (Table A1).

Fig. 1 saturation for VOD > 0.6, can this be related to the saturation of the VOD / Biomass relationship? (0.6 corresponds very well to the saturation level of X-band)

Response: This is an interesting point; however, if saturation had a strong impact here, the relationship should yield more an exponential-like behavior rather than the optimum curve-like behavior, which we observed in the partial dependency plots for VOD and T2M.

Line 179, add that you consider 8-daily values.

Response: We will add this in the revised version.

Line 218 "increase of" ?

Response: Thank you, we will revise this.

– check grammar in line 223-225.

Response: We will revise these lines.

– Figure 3 and 4: is this based on the whole study period (please add the information caption and check throuhout)?

Response: The analyses in both figures are based the whole study period. We will add this information to the captions.

– Figure 4, the overestimation in the tropics seems to be much more significant than the very small decrease outside -35_, +60_. Can we really consider this is "improvement"?

Response: At first glance, it might appear contradictory that this result may be considered an improvement. However, when looking at the relative latitudinal distribution in Figure A1, the setup with temperature shows closer agreement with the other two data sets.

- add site for Fig 5 in the Fluxnet map. Why selecting this site: is it representative of more general results (specific canopy types, climate?)?

Response: The area was selected as an example of a region where the residuals between GPPvodtemp and GPPfluxcom/GPPmodis are not correlated with SPEI. We will add this information and the information about the location used in Figure 5.

– line 239; "holds true" is too strong; here you only find no contradiction on a specific point; mathematically it is not right at all to say that the hypothesis is validated. It is only one indication you go the right way …

Response: We will rephrase it.

References:

Li, X. et al.., Global-scale assessment and inter-comparison of recently developed / reprocessed microwave satellite vegetation optical depth products, Remote Sens. Env., DOI: 10.1016/j.rse.2020.112208

AlâˇAˇRYaari, et al. Asymmetric responses of ecosystem productivity to rainfall anomalies vary inversely with mean annual rainfall over the conterminous U.S, Global Change Biology, 00:1–15, https://doi.org/10.1111/gcb.15345.

Tian, F., et al. "Coupling of ecosystem-scale plant water storage and leaf phenology observed by satellite", Nature Ecology & Evolution, Vol 2, Spt. 2018, 1428–1435, https://doi.org/10.1038/s41559-018-0630-3, 2, 1428–1435, 2018.

Brandt M., "Satellite passive microwaves reveal recent climate-induced carbon losses in African drylands", Nature Ecology and Evolution., https://doi.org/10.1038/s41559-018-0530-6, 2, 827–835 2018.

Fan L., et al., Satellite observed pantropical carbon dynamics, Nature Plants, 5, 944–951, July 2019, https://doi.org/10.1038/s41477-019-0478-9.

Frappart F., et al., Global monitoring of the vegetation dynamics from the Vegetation Optical Depth (VOD): a review, Remote Sensing, 12, 2915, 2020 doi:10.3390/rs12182915

References (mentioned in the response):

Chaparro, D., Duveiller, G., Piles, M., Cescatti, A., Vall-Llossera, M., Camps, A., & Entekhabi, D. (2019). Sensitivity of L-band vegetation optical depth to carbon stocks in tropical forests: a comparison to higher frequencies and optical indices. Remote sensing of environment, 232, 111303.

Green, J. K., Berry, J., Ciais, P., Zhang, Y., & Gentine, P. (2020). Amazon rainforest photosynthesis increases in response to atmospheric dryness. Science advances, 6(47), eabb7232.

Hastie, T. and Tibshirani, R. (1987). Generalized additive models: some applications, Journal of the American Statistical Association, 82, 371–386.

Huete, A. R., Didan, K., Shimabukuro, Y. E., Ratana, P., Saleska, S. R., Hutyra, L. R., … & Myneni, R. (2006). Amazon rainforests green-up with sunlight in dry season. Geophysical research letters, 33(6).

Jung, M., Schwalm, C., Migliavacca, M., Walther, S., Camps-Valls, G., Koirala, S., ... & Reichstein, M. (2020). Scaling carbon fluxes from eddy covariance sites to globe: synthesis and evaluation of the FLUXCOM approach. Biogeosciences, 17(5), 1343-1365.

Litton, C. M., Raich, J. W., Ryan, M. G. (2007). Carbon allocation in forest ecosystems. Glob. Chang. Biol. 13, 2089–2109. URL https://onlinelibrary.wiley.com/doi/abs/10. 1111/j.1365-2486.2007.01420.x, doi:10.1111/j.1365-2486.2007.01420.x

Koren, G., van Schaik, E., Araújo, A. C., Boersma, K. F., Gärtner, A., Killaars, L., ... & Peters, W. (2018). Widespread reduction in sun-induced fluorescence from the Amazon during the 2015/2016 El Niño. Philosophical Transactions of the Royal Society B: Biological Sciences, 373(1760), 20170408.

Morton, D. C., Nagol, J., Carabajal, C. C., Rosette, J., Palace, M., Cook, B. D., ... & North, P. R. (2014). Amazon forests maintain consistent canopy structure and greenness during the dry season. Nature, 506(7487), 221-224.

Li, X., Wigneron, J. P., Frappart, F., Fan, L., Ciais, P., Fensholt, R., ... & Moisy, C. (2021). Global-scale assessment and inter-comparison of recently developed/reprocessed microwave satellite vegetation optical depth products. Remote Sensing of Environment, 253, 112208.

Servén, D., & Brummitt, C. (2018). pyGAM: generalized additive models in python. Zenodo. DOI, 10.

Teubner, I. E., Forkel, M., Jung, M., Liu, Y. Y., Miralles, D. G., Parinussa, R., ... & Dorigo, W. A. (2018). Assessing the relationship between microwave vegetation optical depth and gross primary production. International journal of applied earth observation and geoinformation, 65, 79-91.

Teubner, I. E., Forkel, M., Camps-Valls, G., Jung, M., Miralles, D. G., Tramontana, G., ... & Dorigo, W. A. (2019). A carbon sink-driven approach to estimate gross primary production from microwave satellite observations. Remote Sensing of Environment, 229, 100-113.

Woodhouse, I.H. (2005). Introduction to Microwave Remote Sensing. CRC Press.

Zhao, M., Heinsch, F. A., Nemani, R. R., & Running, S. W. (2005). Improvements of the MODIS terrestrial gross and net primary production global data set. Remote sensing of Environment, 95(2), 164-176.

---

## Author Comment (AC2) · 2 Feb 2021

This work presents a model to estimate Gross Primary Production (GPP) globally from a carbon sink driven approach. In particular, the paper aims at improving previous modelling of GPP as a function of the vegetation optical depth (VOD; Teubner et al., 2019) by including the effect of temperature on the autotrophic respiration. Authors explain that the model is based on the fact that VOD is a good proxy of above-ground biomass (AGB). The link between residuals of the model and the drought index SPEI is also analysed. The results presented show an improvement of model performance in terms of temporal dynamics, especially in non-tropical regions. Interestingly, results also report that the presented model does not require complementary information from precipitation or drought indicators.

Despite that the results presented are consistent with previous works and show interesting contribution to GPP modelling, I have important concerns that have to be addressed before publication. The most relevant are related to the lack of penetration through the vegetation canopy of the X-band VOD (which is not a good proxy of biomass if compared to other frequency bands), and to the need of further explaining the modelling framework both in the introduction and the methods sections. These comments and other major and minor proposals for improving the paper are detailed hereafter:

Response: We thank David Chaparro for his detailed review of the manuscript and are happy to address his comments.

**Major comments**

1. Although the paper can be well understood if the reader knows previous literature on this topic published by the authors, it is necessary that the modelling approach (i.e., main ideas and equations from previous works) and the implementation in the current paper are explained in more detail. In particular:

a. I suggest that first paragraphs of the introduction provide a more detailed description of the framework explained in Teubner et al. (2019), including if necessary some equations (e.g., equations 4 to 6 in Teubner et al., 2019).

Response: Thank you for this suggestion. This was also suggested by the first referee. We will add this information in the revised manuscript.

b. Please, provide more detail on how you are computing and including into the model the different variables (Section 2.2). For instance, does the term "VOD" refer to VOD time-series? If so, how is the time-domain processed (raw data, smoothing, etc…)?

How is the variable computed?

In summary, please extend the text to provide enough information for readers that do not know your previous work.

Response: The term VOD refers to VOD time series. The VOD data were resampled to 8-daily values using the mean over 8 days and then used as input to the model, as stated in Lines 121-121 in Section 2.1. The temporal resolution of 8-daily values was chosen to match the resolution of FLUXCOM and MODIS. We will also include this information in Section 2.2 to make this clearer.

2. The basis of the work is the fact that VOD is a good proxy of AGB. Nevertheless, it is very important to note that X-band VOD (hereafter XVOD) has poor capacity to penetrate the vegetation canopy, and therefore it is very limited to accurately track AGB in regions with dense vegetation. While the AGB - L-band VOD relationship shows low saturation in tropical regions, X-band is not the most appropriate frequency to be used in these areas as a proxy of biomass (e.g., Brandt et al., 2018; Rodríguez-Fernández et al., 2018; Chaparro et al., 2019). Actually, even in low carbon density areas, XVOD is more representative of vegetation cover than it is for biomass, while lower frequencies (L-band VOD; hereafter LVOD) still have improved capacity to track AGB in these areas (e.g., see Fig. 9 in Chaparro et al., 2019).

Response: The reviewer is right: VOD from L-band has been demonstrated to be a suitable proxy for biomass which is less prone to saturate at high biomass than VOD from X-band (e.g., Chaparro et al., 2019; Li et al., 2021). This is well in line with theory, since L-band VOD is more sensitive to large plant parts while X-band VOD is more sensitive to small vegetation parts as leaves and twigs (Woodhouse, 2005). However, in our current study we do not estimate AGB but GPP and in a previous study we demonstrated that VOD from L-band (SMOS) yields the lowest correlation with GPP in vast areas of the world (Teubner et al., 2018).

It is very likely that this limitation explains the lack of improvement of the model in the tropical regions (Fig. 3b) and the low correlation between the model and the benchmark datasets in regions such as the Amazon (Fig. 3a).

Response: The poor agreement of our model in tropical regions may have several sources of uncertainty. First, our model can only be trained based on a few FLUXNET sites in tropical regions as most measurement sites are located in mid-latitudes. The same is also true for the FLUXCOM and MODIS GPP datasets. Second, the FLUXCOM and MODIS GPP products rely on optical satellite data which is highly affected by cloud cover in tropical regions (Zhao et al., 2005). Hence, correlation analyses with biophysical vegetation properties from optical sensors generally suffer from higher uncertainty of these datasets over tropical regions (Zhao et al., 2005).

In addition, this could also explain the saturation of the partial dependency plots (Fig. 1) at high VOD and T2M values (darkest lines in the first panel, probably representing vegetation-temperature conditions in the tropics) and at mdnVOD values above 0.4 (i.e., dense vegetation; third panel).

The application of XVOD is justified in the paper by the higher correlation between XVOD and GPP if compared to the LVOD-GPP correlation (Teubner et al., 2018). Nevertheless, it is important to note that the GPP benchmark datasets have an important contribution from visible-infrared (VIS-IR) indices (EVI, LAI, MIR, NDVI and NDWI, as stated in l. 96). I think it is expected that GPP datasets based on VIS-IR indices show a greater correlation with XVOD than with lower frequency VOD data, because both of them capture the same layer of vegetation (top of the canopy).

Response: Yes, this is true. As both VIS-IR-based indices or biophysical properties and XVOD are sensitive to the upper vegetation canopy, the layer where photosynthesis predominantly takes place, they can also well capture the temporal dynamic of GPP. This is exactly the motivation of using higher frequency microwave observations in our model. This has been further corroborated by

correlation analyses carried out in Teubner et al. (2018). Despite the close agreement between optical indices and XVOD, we would like to note that our model still presents a sink-driven approach. This is demonstrated by the higher performance for the combination of VOD and ΔVOD compared to each input variable taken separately (Teubner et al., 2019).

In contrast, I would expect greater correlations with in situ GPP FluxNet data (Fig. 2a and Fig. 2b) if GPPvod and GPPvodtemp were computed using L-band VOD. Although it is not a global dataset, FluxNet in situ information is not conditioned by physical properties of remote sensing sensor measurements, so it is probably the most "independent" tool the authors have for evaluating the accuracy of the GPP estimates.

Response: During our previous analyses, LVOD consistently showed lower performance than higher microwave frequencies. At the global scale, among frequencies from L- to X-band, LVOD resulted in lowest correlations (Teubner et al., 2018). For in situ stations, a similar result, i.e., a closer agreement between GPP and XVOD than between GPP and LVOD, is obtained for the correlation of FLUXNET GPP with LVOD and XVOD (Figure AC1). A classification of the results into land cover classes shows that also a fair amount of forest stations is included in this analysis (~38%). In addition to the analysis of 8-daily values, we repeated the analysis also for monthly values. This was done in order to exclude the possibility that the low correlations for LVOD are caused by the relatively high temporal resolution of 8 days. However, a similar result as for 8-daily values is obtained.

Based on these results, we concluded that LVOD may not be a good proxy for estimating GPP and did not include LVOD in further analyses.

[Figure]

Figure AC1: Left: Correlation between FLUXNET GPP (mean of GPP_DT_VUT_REF and GPP_NT_VUT_REF) and VOD from two different frequencies, L-band VOD (SMOS VOD-L, 7/2010–12/2014) and X-band VOD (AMSR-E VOD-X, 1/2007–9/2011). Data were resampled to 8-daily or monthly values. The analysis was conducted only for stations where both of the VOD data set were available (47 stations). Right: Composition of land cover classes for the stations used in the analysis. Abbreviations: GRA (Grasslands), CRO (Croplands), ENF (Evergreen Needleleaf Forests), DBF (Deciduous Broadleaf Forests), EBF (Evergreen Broadleaf Forests), SAV (Savannas), MF (Mixed Forests), WET (Permanent Wetlands), WSA (Woody Savannas) and OSH (Open Shrublands).

For all these reasons, it would be very interesting if the authors include new GPPvod and GPPvodtemp models based on L-band VOD and validate their accuracy using in situ FluxNet data (i.e., including them in Fig. 2). They could show (and compare) the resulting GPP estimates between different frequencies and, importantly to preserve the scope of the paper, between GPPvod and GPPvodtemp models. However, I am aware that this could move the work slightly beyond its initial scope, as it adds another factor (i.e., frequencies) in the comparisons. I encourage the authors to work on this possibility, although it is up to them to finally incorporate this change or to keep only XVOD in the paper. In any case, they must discuss all the possible implications of using XVOD.

Response: Thank you for your suggestion. However, based on our above analysis results that show a poor performance of LVOD in predicting GPP, we will not add LVOD to the paper to not further complicate the paper. But we follow the suggestion of the referee and will improve our discussion with respect to the use of XVOD vs. LVOD data.

Within the discussion, they should address at least the following points/questions:

- It is stated that "the VOD-GPP model relies on estimating carbon sink terms, [...], based on VOD as a proxy of aboveground living biomass" (l. 35-37). To what point is this true, according to the facts that XVOD is more representative of vegetation cover than of biomass, and that lower VOD frequencies have enhanced capacities to capture biomass? Please clearly explain the possible limitations of the approach.

Response: Indeed, the use of VOD from X-band may appear counterintuitive and the use of LVOD might seem more appropriate. However, as LVOD cannot capture well the intra-annual temporal dynamics of canopies, it cannot serve as a good proxy for the high-frequency temporal dynamics in GPP. In addition, the reason for using X-band VOD (and not L-band VOD) for the estimation of GPP might be explained with differences in metabolic activity of various plant parts. Litton et al. (2007) showed that leaves and roots may present a better proxy for GPP than large structural components with a lower metabolic activity. For further details, see discussion section 6.2 in Teubner et al. (2019). On the other hand, LVOD might be a better predictor for the annual-integrated total carbon allocation (net primary production) as this is closely related to the annual increment in biomass. However, testing this hypothesis is beyond the scope of our study.

We will expand our discussion for using X-band VOD to make this clearer.

- Please, discuss about the saturation effects in Fig. 1 (first and third panel; see my comment above). Are they likely to be linked to XVOD saturation in the tropics? If so, which are the implications?

Response: Although we cannot rule out a saturation effect of VOD with regard to GPP, the partial dependency plots should, in this case, resemble an exponential curve and not the optimum curve, which we observed. The plots rather suggest that the in situ GPP estimates are limited at around 10 gC m-2 d-1.

- In l. 197, it is mentioned (referring to tropical regions) that "[in these regions] sensitivity to temperature is also low, which makes the interaction term mainly controlled by VOD."

If I correctly understood plots in Fig. 1, would it be more precise to affirm that it is mostly controlled by , as other dependencies (VOD, mdnVOD) saturate in tropical regions?

Response: Our statement is referring to the lack of a strong seasonality in the tropics, which might explain why the improvement in temporal correlation is absent or lower than in temperate regions because these regions do not experience large temperature differences.

3. The GPP estimates (GPPvod and GPPvodtemp) are calibrated using FluxNet in situ data. Also, both FluxNet and FLUXCOM data (an upscaling of FluxNet) are used as reference datasets for evaluating GPP estimates. I think that, consequently, reference datasets may not be fully independent from GPP estimates. To what extent? Which is the contribution of FluxNet data in the reference datasets and in the estimates? This has to be acknowledged and possible implications discussed.

Response: Yes, the FLUXCOM product has been trained against the FLUXNET station data (Tramontana et al., 2016; Jung et al., 2020). In comparison to the FLUXCOM product, we could use much fewer station data (only Tier 1 data). Also, the MOD17 GPP product has been partly calibrated

to some FLUXNET stations (Running et al., 1999). At large/global scales, there is no alternative to constrain absolute estimates of GPP than using FLUXNET data. We will add this point to the discussion.

In addition, authors should try to guarantee at least one year of "fully independent" comparison between estimates and FluxNet/FLUXCOM data. I suggest they could calibrate the model by leaving one year of data apart (e.g., use 2004-2014 for calibration) and apply the remaining data (2003) for fully independent comparisons. They can show these new comparisons in supplementary materials and refer to them to show consistency/inconsistency with the full-period comparisons of Figures 2 to 6.

Response: As a common approach (that was also used in FLUXCOM), we performed leave-site out cross-validation and we will add these cross-validation results for GPPvod and GPPvodtemp in the supplement.

**Minor comments**

- Lines 5-6: "VOD-GPP model generally showed good agreement" → Please quantify (e.g. correlation coefficient).

Response: A quantification of the previous correlation coefficient would imply that the results can be directly compared. However, different aspects of the model have changed. In the current study, we are using a merged VOD (VODCA X-band), while we previously only considered single sensor VOD (AMSRE or AMSR2 X-band). In addition, we analyzed a longer period, which also makes the training dataset different from the previous studies. Therefore, we would not add information of correlation coefficients from previous studies in the abstract.

For the direct comparison of previous (GPPvod) and current (GPPvodtemp) model results under similar conditions, we would like to refer to our results presented in section 3.3.

- L. 6: "tended to overestimate" → By how much? Please quantify.

Response: The overestimation was not quantified in the previous study; therefore, we cannot provide a specific number here.

- L. 13: "Our results reveal an improvement" → Please quantify this improvement (e.g., increase on the average correlation).

Response: Thank you for this suggestion, we will add the increase in correlation coefficient from the results section also in the abstract.

- L. 14: "This increase in temporal dynamic" → "This improvement in temporal dynamics."

Response: We will revise this.

- L. 19: can you mention which are these regions?

Response: We will add this information.

- L. 19: "between […] with" → "between […] and"

Response: We will revise this.

- L. 25: provide → provides

Response: We will revise this.

- L. 25 to 30: you may want to include other references which are explicitly linked to water content: e.g., Feldman et al., 2018; Tian et al., 2018.

Response: Thank you for this suggestion, we will add these references.

- L. 28: Chaparro et al., 2019→Chaparro et al., 2018 (this is different from the "carbonstocks work" in Chaparro et al., 2019). Add the new reference to the references list if you want to keep it in the text.

Response: Thank you, we will revise it.

- L. 70: maybe saying "only a few years" is a bit excessive (e.g., SMOS spans >10 years). Try using another expression, please.

Response: It was meant it in the sense that the use of single sensor VOD may hamper the comparison of longer time periods. We will rephrase it.

- L. 83-85: "During data processing…" → Please move these lines to the methods section.

Response: We will move this sentence to section 2.1 Data processing.

- L. 118: "T2M was used in our analysis, since this parameter is most common for describing the temperature dependency" → Please add some references to show the common use of this variable.

Response: We will add references regarding the use of T2M for describing the temperature dependency of autotrophic respiration.

- L. 121-122: "aggregated to 8-daily estimates" → Please, specify that this is done to match MODIS time-steps in case it was your intention.

Response: We will add this information.

- L. 162: "savitzky-golay" → "Savitzky-Golay."

Response: We will revise this.

- L. 170: "are consistent the previous" → "Are consistent with the previous."

Response: We will revise it.

- Figure 1: Please add marginal distribution rug plots to each panel. Name the panels as "a" to "d" and complete the figure caption with explanation of each panel.

Response: Thank you for this suggestion. We will add rug plots and include the information about each panel in the caption. Panel names are already included in the bottom of each panel.

- Figure 2: add GPPvod as another panel for comparison with GPPvodtemp and benchmark data. Also, I find that the blue to yellow colorbar shows very low contrasts. To me, it is difficult to appreciate color gradients in the figure. You could use other colors (e.g., blue to red?) or saturate the colorbar at the (e.g., 95th, 99th) percentile to improve contrast.

Response: We will add GPPvod and adjust the color bar.

- L. 218: "For a region in Europe" → Please add coordinates also here, as well as the general situation (e.g., "Central Europe") to help the reader.

Response: We will include the information about the location given in the caption of Figure 5 also in the text.

- L. 218: "increase" → "increase in".

Response: We will revise this.

- L. 227-229: there is no a verb in this sentence, please rephrase.

Response: We will revise this.

- L. 233: "between […] with" → "between […] and".

Response: We will revise this

- Figures 4 and 6: please define in the text what are "zonal means".

Response: We will include this information.

- Figures 5 and 8: please, could you make each panel wider? Then there is more place for seeing interannual variability in the figures.

Response: We will update the figure layout.

- Figure 6: Add GPPvod as another panel for comparison with GPPvodtemp and benchmark data.

Response: Although this would be possible in general, we would rather keep the main part as it is. The reason for this is that we narrowed the two setups down to one setup (i.e. GPPvodtemp) based on the results in the preceding section. But in order to provide the information, we will put the figure for GPPvod in the appendix.

- L. 236: "given that correlations in these regions are high" → Authors probably mean correlations between GPPvodtemp and benchmark data. Please specify this, because as you explained correlations of residuals in the previous sentence, it can be confusing.

Response: We will clarify this.

- Figure 7: to improve the boxes for regions, you could use colors different than those from the colorbar (e.g., light green instead of blue?).

Response: We will adjust the color.

- L. 282: "increase" → Do you mean "improvement"?

Response: We will revise it.

- L. 338: "between with" → "with".

Response: We will revise this.

- Figure A1: please add GPPvod as well as a map of the median differences between GPPvod and GPPvodtemp. Also, note that the contrast in the blue to yellow colorbar could be improved, or colors changed to a blue-red scale.

Response: We will add GPPvod and a difference map between GPPvod and GPPvodtemp.

- Figure A3: please add GPPvod.

Response: Consistent with the scatter plots, we will add GPPvod to this figure.

- Figure A4: this seems an interesting result, but I do not fully understand what do you mean by "scaled latitudinal" distribution. Could you please explain this?

Response: The scaled latitudinal distribution is obtained by dividing the data by the maximum of the latitudinal distribution. We will rephrase it to make it clearer.

- Table A1: it could be useful to detail the dominant land cover in each station. Then, the reader will be able to see which vegetation types have been included in calibration of GPP estimates.

Response: We will include the information about land cover in the table.

---

## Referee Report (RR1)

The reviewed manuscript has been clearly improved from the previous version, and authors have addressed all the comments with a very thorough revision. Both the introduction and methods sections have been completed with further details on the previous literature which is the basis of the work. The fact of using X-band VOD instead of L-band VOD has been explained and discussed consciously. The manuscript will be ready for publication after addressing some minor comments that may provide a more complete understanding and interpretation of results, as well as after checking other small (text and spelling) issues:

- Discussion on X-band and L-band is appropriate and very thorough. Figure A1 supports it very well. A last point for improving Figure A1 would be to provide the same analysis only for forests (i.e., a third panel in Fig. A1, similar to panel *a*, only containing forest stations).
- In L. 262 you comment that *"For a region in Europe (5 to 15°E and 46 to 51°N), where we generally did observe an increase in all three performance metrics, we find that for GPPvod mainly winter time estimates of GPP are too high compared to GPPfluxcom and GPPmodis (Figure 5). By adding temperature as input to the model, winter observations are markedly dampened and summer observations are only slightly increased."* This is a very nice result to show, nevertheless it refers only to one region. It will provide more consistency to your results if you include two or three similar regions and see if the winter observations also dampen when adding temperature to the model. This can be included in the supplement.
- Figure A5: this figure provides an interesting conclusion, as you mention in L. 260. In my opinion, this is relevant enough to be moved to the main part of the paper. I suggest to show it as a second panel in Figure 4. Also, in the discussion, you can emphasize the fact that adding temperature improves the relative distribution of GPP.
- Section 4.4: please, add a sentence to highlight that the leave-site-out cross validation shown in Table A2 confirms that, although the benchmark GPP datasets are not fully independent, this does not impact the results.

Other minor comments are:

- L. 75: "have been demonstrated" → "has been demonstrated".
- L. 79-81: in these lines you describe Figure A1, but data needed for this figure have not been described before. At least you should include a sentence explaining in which sections the reader can find a description of the data.
- L. 93-98: these sentences are redundant. Can you join frequencies (l. 93-94) with time overpasses (l. 95-97) to avoid writing the satellite names twice?
- L. 177: "pyGAM" → "The pyGAM".
- L. 272: "GPP data set" → "GPP datasets"
- L. 380: "sensitive" → "sensitive to"

---

## Author Response (AR2)

Impact of temperature and water availability on microwave-derived gross primary production

by Irene E. Teubner

This study evaluates the capabilities of VOD to provide new information on the changes in vegetation productivity at a global scale. Specific improvements obtained by accounting for temperature effects on autotrophic respiration are analyzed.

I found that interesting results are presented. However, I think significant improvements should be made. As I am not familiar with the studies by the authors on this topic, I found it is difficult to understand many points in this manuscript, unless, maybe, I read in detail all papers published before. Basic elements of the modeling approach published before should be given here, so that the paper is "more autonomous". I present below many points to be improved, so that readers who are not familiar with the papers published by the authors, may understand the results and the discussion

Response: Dear Jean-Pierre Wigneron,

many thanks for your detailed and very constructive review. We agree that the manuscript is hard to understand without knowledge of our previous publications. Therefore, we provided a concise repetition of the methodology, the main equations and assumptions from Teubner et al. (2019) in section 2.4.

Lines 149-162: "The approach of estimating GPP based on microwave radiation and the corresponding equations are described in detail in Teubner et al. (2019). In short, the VOD-GPP model uses VOD as a proxy of above-ground living biomass (Equation 1). It determines GPP by estimating sinks for carbohydrates, i.e. the sum of NPP and Ra, which are represented through different VOD-derived variables: 1) time series of the bulk VOD signal ($VOD$; 8-daily aggregated native VOD time series), 2) time series of the temporal change in VOD ($\Delta VOD$; $\Delta VOD_t = VOD_t - VOD_{t-1}$ computed from the smoothed 8-daily aggregated VOD time series) and 3) the grid cell median of VOD ($mdnVOD$; calculated over the entire VOD time series of the grid cell; used as a proxy for vegetation cover). While NPP is related to $\Delta VOD$, Ra is related to both $VOD$ and $\Delta VOD$ using the concept proposed by Ryan et al. (1997) of dividing Ra into maintenance and growth respiration (Equation 2). By assuming that belowground biomass terms are proportional to above-ground biomass (i.e. biomass $B$ can be expressed through above ground biomass $AGB$) and by adding a static term $c$ supporting the conversion in Equation 2, GPP can be represented through a differential equation with VOD as input (Equation 3).

$$AGB = f(VOD) = \widehat{VOD} \tag{1}$$

$$GPP = NPP + Ra = \left(\frac{dB}{dt} + loss\ terms\right) + \left(a_0 \frac{dB}{dt} + b_0 B\right) \sim a\frac{dB}{dt} + b\,B \tag{2}$$

$$GPP = a \frac{d\widehat{VOD}}{dt} + b \widehat{VOD} + c \qquad\qquad (3)$$
"

I have 4 main comments which should be accounted for before publication

1) I think, the lack of improvement in the tropics could be related to the low sensitivity of X-VOD to biomass changes, which was found in many regions of the world but particularly in the tropics. This should be better discussed and accounted for throughout the manuscript.

Response: We agree that X-band VOD is less sensitive to changes in total biomass in the tropical forests, since L-band VOD has been shown to be less prone to saturation at high biomass values (e.g., Chaparro et al., 2019; Li et al., 2021). This may thus suggest that the use of L-band VOD would yield better results in estimating GPP than X-band VOD. However, various studies show that X-band VOD generally yields a higher agreement with GPP (Teubner et al., 2018) or proxies for productivity like NDVI (Li et al., 2021) than L-band VOD. This finding that L-band VOD agrees less with GPP than X-band VOD was also observed for the land cover types with higher biomass, i.e. forest (Teubner et al., 2018). The reason for the general higher suitability of X-band VOD for the estimation of GPP in forests and other land cover types with a large woody component can be explained by differences in metabolic activity of various plant parts. Plant parts with a higher metabolic activity, like leaves and roots, are a better proxy for GPP than large structural components like branches and trunks (Litton et al., 2007). For further details, see discussion section 6.2 in Teubner et al. (2019). In addition, our approach is based on the change in biomass between two successive time steps (mostly a few days). At this short time interval, leaf biomass - to which higher frequency microwaves are more sensitive (Woodhouse, 2005) - is expected to potentially show changes, while woody above-ground components - to which lower frequency microwaves like L-band are more sensitive (Woodhouse, 2005) - may not.

However, in general the correlations between all VOD bands and existing global GPP products are low in evergreen broad-leaved tropical forests (Teubner et al., 2018). This disagreement can be due to multiple reasons. First, also data-driven large-scale products of GPP like FLUXCOM are based on only few station observations in tropical forests and thus have large uncertainty in this region (Jung et al., 2020). Second, the optical remote sensing input data to produce these datasets are highly affected by cloud cover (Zhao et al., 2005).

To make the choice of X-band in our approach clearer, we added the issue of high- vs low-frequency VOD in the beginning of the data & methods section as well as in the discussion. In addition, we added a graph for correlation analysis between in situ GPP and VOD from L- and X-band in the supplement (Figure A1).

Lines 73-84: "2.1 Choice of microwave frequency

[revised manuscript text omitted]

For instance, many references discussing the capabilities of VOD to monitor biomass are missing. Cf below references on this topic including applications on biomass changes / productivity monitoring, to better account for and reflect the published literature on this topic (Brandt et al., 2018, Fan et al., 2019; Al -Yaari et al., 2020; Lei et al., 2020; Frappart et al., 2020).

Response: Thank you for these suggestions and we are well aware about these publications. However, some of the papers that you co-authored (Frappart et al., 2020; Li et al., 2020) were published just before or after the submission of our manuscript and hence we could not include it. We incorporated these references in introduction and discussion where appropriate.

line 14: "regions outside the tropics", many studies have shown that X-band and Cband VOD present saturation vs biomass (close to 200 t/ha). So, how do you expect to monitor GPP from VOD indices that saturate over dense vegetation forests, which represent a large fraction of the vegetation cover in those regions.

Response: See our reply above about our previous findings on the correlation between VOD bands and GPP.

line 72-74, I did not review these previous papers, but I think it is quite surprising that XVOD provide best agreement with GPP by considering "sink terms" related to biomass changes. X-VOD is better related to LAI/NDVI (and thus to photosynthesis and "source terms"), while L-VOD is better related to biomass changes (see Li et al., 2020).

Response: X-band VOD is indeed more similar to LAI/NDVI which represent commonly used inputs for source-driven approaches. Although high-frequency VOD is closely related to small vegetation parts like leaves and twigs (Woodhouse, 2005) and therefore closely related to LAI or NDVI, our method can still be regarded as a sink-driven approach. If X-band VOD was better suited for a source-driven approach, the use of VOD alone should be enough for estimating GPP. However, Teubner et al. (2019) demonstrated that the combination of VOD and change in VOD (computed as the change over consecutive time steps) outperforms the use of each VOD variable taken separately. We think that this performance is related to allometric relationships between the biomass in canopies and biomass in stems. A certain long-term increase in canopy biomass should correspond to an increase in stem diameter and tree height. Hence, we assume that the sensitivity of X-band VOD to the canopy serves also as proxy for changes in total above-ground biomass. The sensitivity of high-frequency VOD to allometric patterns in vegetation is also supported by the overall medium to high correlation between VOD and canopy height which is partly higher for X-band than for L-band (Li et al., 2020). Hence, we assume that our sink-driven approach is even valid if the used VOD band is not directly sensitive to the changes in total above-ground biomass.

Still based on L-VOD, Tian et al., 2018, found a decoupling between seasonal changes in VOD and in the leafy/biomass component in dry tropical forests. This should explain some errors too in the tropics, when attempting to relate VOD changes to vegetation productivity (?)

Response: Such temporal shifts between VOD and LAI (or GPP) based on optical data may occur and may result in negative correlations between VOD and GPP (Teubner et al., 2018). A potential reason could be that water availability and photosynthesis are negatively correlated in tropical forests (Green et al., 2020), although the hypothesis of increasing photosynthesis under drought conditions in the Amazon is highly controversial (Koren et al., 2018; Huete et al., 2006; Morton et al., 2014). Hence the sensitivity of VOD to the vegetation water content might explain why VOD is negatively correlated to GPP in tropical forests. As we do not specifically account for this, it will have an effect on our estimates of GPP in tropical forests as also discussed before in Teubner et al. (2019).

2) I found it is very difficult to understand section 2.2, except if you are an expert in this specific modelling approach Temperature is an important parameter in Ra but also in other key processes such as photosynthesis. How can you be sure that only the Ra(T) dependence was accounted for here? Because I cannot see any deterministic equations relating Ra to temperature: in Eq 1 and 2, is it fully a machine learning approach that you used, isn't it?

Response: Although our approach is based on a machine learning method, we make use of knowledge about the relationships between VOD and biomass and between biomass and GPP. Equations 1 and 2 present the machine learning formulas. For the temperature dependency of Ra, we did not present a formula per se since it is difficult to express a formula which combines both the instantaneous temperature response (which may not necessarily apply to 8-daily values) and the acclimation effect. We addressed this issue in the introduction. As described in the methods section, this was also the reason for choosing Generalized Additive Models as modelling approach, since the relationship is not required to be known a piori but instead estimated from the input data (Hastie and Tibshirani, 1987).

We improved the description of our method by including the formulas for GPP, on which our approach is based on. See our response above.

As I'm not expert of this kind of regressions and many terms are unclear to me. Maybe, it is very specific to me, but maybe it will apply to many other readers: Better explain what is "VOD time series", "delta VOD", "mdn VOD": over which time step? Considering daily , monthly or yearly values? do you compute mean of delta VOD, etc: : :? What is the time step of Eq 1 & 2: daily? "spline terms for representing 2-dim functions": what do you mean? which 2-dim parameters are considered here? "smoothing factor"?, etc.

Response: Following our approach described in Teubner et al. (2019), the VOD time series are the VOD values that were aggregated to 8-daily time steps, ΔVOD is the change of VOD over consecutive time steps of the smoothed 8-daily VOD time series and mdnVOD is the median of the VOD time series computed over all time steps of the grid cell. As input to the model, 8-daily values were used as stated in section 2.3. The term "2-dimensional" refers to fact that the partial dependency for the spline-terms describe a 2-dimensional function between input and response variable. In contrast, the partial dependency for the tensor-term (with two input variables) is represented by a surface in the 3-dimensional space; the three dimensions being the two input variables and one response variable. In Generalized Additive Models, a number of spline functions are fitted between the input data of the respective term and the response variable; and the resulting partial dependency function is further smoothed (Hastie and Tibshirani, 1987; Servén and Brummitt, 2018). The strength of the smoothing is controlled via the smoothing factor lambda.

We added more details for the mentioned terms or modified the sentences to make it clearer in the revised manuscript.

Lines 144-147: "These 8-daily values were then used as input to the VOD-GPP model and for further analysis throughout the study. GPPfluxcom and GPPmodis were aggregated to 0.25° to match the spatial sampling of VODCAX. For the comparison with SPEI, 8-daily GPP estimates were further resampled to monthly resolution while SPEI was spatially resampled to 0.25° using the nearest neighbour method."

Lines 150-155: "It determines GPP by estimating sinks for carbohydrates, i.e. the sum of NPP and Ra, which are represented through different VOD-derived variables: 1) time series of the bulk VOD signal (*VOD*; 8-daily aggregated native VOD time series), 2) time series of the temporal change in VOD (*ΔVOD*; $\Delta VOD_t = VOD_t - VOD_{t-1}$ computed from the smoothed 8-daily aggregated VOD time series) and 3) the grid cell median of VOD (*mdnVOD*; calculated over the entire VOD time series of the grid cell; used as a proxy for vegetation cover)."

Lines 166-167: "where s denotes spline terms for representing the functions between each input variable and the response variable GPP in the 2-dimensional space."

Lines 175-176: "where te stands for a tensor term, which represents the interaction between VOD and temperature and spans a surface in the 3-dimensional space."

Lines 182-185: "In GAM, a number of basis spline functions are fitted to the data and the resulting function is further smoothed to obtain the final response function (Servén and Brummitt, 2018). The degree of smoothing is determined by the smoothing factor, which yields strong smoothing for high values and low smoothing for low values. For the current models we used a smoothing factor of 2, which is lower than for the model in Teubner et al. (2019)."

3) Did GPP-VOD-temp showed improvements vs GPP-VOD, when considering correlation of residuals with SPEI? Since the present study focuses on analyzing possible improvement of the new GPP-VOD-temp, intercomparing residuals vs SPEI with GPPVOD is key and should be added in this manuscript. The present description of results is a bit lengthy and should be reduced to the profit of the above inter-comparison.

Response: We conducted the suggested analysis for the revised manuscript. Since it did not add much information to the main story, we place it in the supplement.

Lines 296-299: "The analysis of GPPvod residuals reveals a similar result as for GPPvodtemp (Figure A7). For GPPvod, however, the number of grid cells with non-significant correlations in the four analyses is lower by about 2 to 4 % than for GPPvodtemp, while the global average correlation is nearly identical. The higher number of non-significant correlations for GPPvodtemp than for GPPvod is expected, because the addition of temperature accounts for some variation in the VOD-based GPP estimation."

Supplement:

"

Figure A7. Correlation between residuals of standardized GPP (GPPvod-GPPfluxcom and GPPvod-GPPmodis) and SPEI. Non-significant correlations are indicated in grey. (a,c): GPPvod-GPPfluxcom, (b,d): GPPvod-GPPmodis, (a,b): SPEI03 (short-term response), (c,d): SPEI12 (long-term response). Regions A-D: US cornbelt (A), Argentina (B), Eastern Africa (C) and Eastern Australia (D). Results are computed based on the study period 2003-2015."

4) I found that conclusions are much more nuanced considering the relative improvement obtained with the new GPP-VOD-temp product. This should be better reflected in the abstract which I found too optimistic.

Response: We revised the abstract accordingly. It now reads:

Lines 13-25: "Our results reveal an improvement in model performance for correlation when including the temperature dependency of autotrophic respiration (average correlation increase of 0.18). This improvement in temporal dynamic is larger for temperate and cold regions than for the tropics. For ubRMSE and bias, the results are regionally diverse and are compensated in the global average. Improvements are observed in temperate and cold regions while decreases in performance are obtained mainly in the tropics. The overall improvement when adding temperature was less than expected and thus may only partly explain previously observed differences between the global GPP datasets. On interannual time scales, estimates of the VOD-GPP model agree well with GPP from FLUXCOM and MODIS. We further find that the residuals between VOD-based GPP estimates and the other data sets do not significantly correlate with SPEI which demonstrates that the VOD-GPP model can capture responses of GPP to water availability even without including additional information on precipitation, soil moisture or evapotranspiration. Exceptions from this rule were found in some regions: significant negative correlations between VOD-GPP residuals and SPEI were observed in the US corn belt, Argentina, Eastern Europe, Russia and China, while significant positive correlations were obtained in South America, Africa and Australia. In these regions, the significant correlations may indicate different plant strategies for dealing with variations in water availability."

Minor

line 16- 20; it seems to me the two sentences are a bit contradictory

Response: Thank you for this comment. It was meant that the analysis reveals that the residuals largely do not correlate with SPEI, which indicates that the relationship largely is reliable with respect to variations in water availability. Exceptions from this rule are found in some areas which may point towards specific plant properties. We modified these sentences.

Lines 19-25: "We further find that the residuals between VOD-based GPP estimates and the other data sets do not significantly correlate with SPEI which demonstrates that the VOD-GPP model can capture responses of GPP to water availability even without including additional information on precipitation, soil moisture or evapotranspiration. Exceptions from this rule were found in some regions: significant negative correlations between VOD-GPP residuals and SPEI were observed in the US corn belt, Argentina, Eastern Europe, Russia and China, while significant positive correlations were obtained in South America, Africa and Australia. In these regions, the significant correlations may indicate different plant strategies for dealing with variations in water availability."

line 25-30, Cf above remarks on C- and X-VOD saturation, many papers were published based on SMOS L-VOD and none is mentioned in this short review. This short review should be more "opened"

Response: As the focus of our study is the estimation of GPP, we did not specially include papers on the relation between VOD and biomass. We added the issue of the choice of microwave frequency for GPP estimation in the introduction and discussion and included a correlation analysis for L- and X-band versus in situ GPP in the supplement. See our Response above.

line 35: "VOD as a proxy of AGB": I guess very few FLUXNET sites are available in relatively dense vegetation sites, and more generally in the tropics. The VOD-derived GPP is manly calibrated based on data in temperate climate?

Response: The spatial distribution of in situ stations may indeed be an issue which we therefore already addressed in the discussion section 4.2. However, we revised these lines to make it clearer.

Lines 353-358: "Since the interaction term between *VOD* and *T2M* represents a relationship in the 3-dimensional space, certain combinations of *VOD* and *T2M* intervals in the parameter space may not be well represented by the training data. FLUXNET stations are not evenly distributed around the globe, as the majority of stations are located in the temperate region. This may have caused the model to be not well constrained in certain regions, e.g. where temperature and *VOD* are very high, and thus might have contributed to the increase in bias in the tropics."

Line 43: " as a necessity": what do you mean here; not so convincing as a scientific term.

Response: It is meant that dry cells, which do not contain water, will not contribute to the estimation of respiration, since these cells are not living. We made this clearer in the revised version.

Lines 48-51: "Although different studies are tackling the question of how much information on biomass is actually contained in the VOD signal (Momen et al., 2017; Vreugdenhil et al., 2018; Zhang et al., 2019), it might be worth noting that the water content can be seen as an important aspect in our model approach since it presents the living part of the vegetation and only living cells, which contain water, are able to respire."

Lien 50 define what is Q10?

Response: The definition of Q10 is given in Lines 69-71.

Line 94: why not using FLUXCOM RS + Meteo , which has more input and could be more reliable. I do not understand the reason given here "our approach is mainly based on RS..". This is a not a good reason to me (?)

Response: In contrast to the FLUXCOM RS setup, the RS + Meteo setup includes only the mean seasonal cycle of remote sensing data as input but does not use any information from remote sensing data that could be sensitive to inter-annual changes such as under dry conditions. Our approach mainly relies on remote sensing data, namely VOD estimates. For a fair comparison, we thus choose the FLUXCOM RS setup. We modified the sentence to read:

Lines 113-114: "Since our approach is mainly based on remote sensing data, i.e. VOD observations, we used FLUXCOM RS in our analysis."

Please provide more information on Fluxnet sites used here (maps of locations, main vegetation types, etc.)

Response: A map of FLUXNET stations is already given in the Appendix (Figure A2). We added information on the vegetation type and the number of stations for each land cover type in the overview of the FLUXNET stations (Table A1).

Supplement: "Table A1. Overview of FLUXNET Tier1 v1 stations within the period 2003 to 2014. Land cover from IGBP (International Geosphere–Biosphere Programme) is obtained from the FLUXNET station metadata. Land cover abbreviations and number of stations per land cover class sorted by station number: ENF (Evergreen Needleleaf Forests; 23), GRA (Grasslands; 22), DBF (Deciduous Broadleaf Forests; 14), CRO (Croplands; 11), EBF (Evergreen Broadleaf Forests; 9), WET (Permanent Wetlands; 9), OSH (Open Shrublands; 7), MF (Mixed Forests; 6), SAV (Savannas; 6), WSA (Woody Savannas; 4) and CSH (Closed Shrublands; 1).

| FLUXNET-ID | Name | Lon [° E] | Lat [° N] | Years used | Land cover |
|---|---|---|---|---|---|
| AR-SLu | San Luis | -66.46 | -33.46 | 2009-2011 | MF |
| AR-Vir | Virasoro | -56.19 | -28.24 | 2010-2012 | ENF |

…

"

Fig. 1 saturation for VOD > 0.6, can this be related to the saturation of the VOD / Biomass relationship? (0.6 corresponds very well to the saturation level of X-band)

Response: This is an interesting point; however, if saturation had a strong impact here, the relationship should yield more an exponential-like behavior rather than the optimum curve-like behavior, which we observed in the partial dependency plots for VOD and T2M. We addressed this in the discussion section 4.3.

Lines 369-375: "Nonetheless, the impact of such potential saturation with biomass on the estimation of GPP is less trivial, especially with regard to densely vegetated areas like the tropics. Non-linearity in the conversion between VOD and AGB should ideally be reflected in the partial dependency plot of GAM, which was also the reason for choosing this type of modelling approach. Scatterplots of the resulting GPPvodtemp estimates did not show clear signs of saturation at high in situ GPP. The FLUXNET training data set, however, only has few stations in the tropics and thus the robustness of the model may be limited by the availability of in situ stations."

Line 179, add that you consider 8-daily values.

Response: We included this information in section 2.3.

Line 144: "These 8-daily values were then used as input to the VOD-GPP model and for further analysis throughout the study."

Line 218 "increase of" ?

Response: Thank you, we revised this.

Lines 262-264: "For a region in Europe (5 to 15°E and 46 to 51°N), where we generally did observe an increase in all three performance metrics, we find that for GPPvod mainly winter time estimates of GPP are too high compared to GPPfluxcom and GPPmodis (Figure 5)."

– check grammar in line 223-225.

Response: We revised these lines to read:

Lines 267-270: "In the remaining study, due to the observed bias (both at site-level and global scale), we are analyzing relative rather than absolute values for comparing interannual variability and the impact of water availability. In addition, we are focusing our further analysis on GPPvodtemp since this setup overall showed higher performance than GPPvod. Results for GPPvod are displayed in the supplement for comparison with GPPvodtemp."

– Figure 3 and 4: is this based on the whole study period (please add the information caption and check throuhout)?

Response: The analyses in both figures are based the whole study period. We added this information to all captions.

– Figure 4, the overestimation in the tropics seems to be much more significant than the very small decrease outside -35_, +60_. Can we really consider this is "improvement"?

Response: At first glance, it might appear contradictory that this result may be considered an improvement. However, when looking at the relative latitudinal distribution in Figure A5, the setup with temperature shows closer agreement with the other two data sets.

- add site for Fig 5 in the Fluxnet map. Why selecting this site: is it representative of more general results (specific canopy types, climate?)?

Response: The area was selected as an example of a region where the residuals between GPPvodtemp and GPPfluxcom/GPPmodis are not correlated with SPEI. We added this information to the caption and included the information about the location from Figure 5 in Figure A3.

Caption Figure 5: "Time series plot of spatially aggregated GPP estimates for GPPfluxcom, GPPmodis and (a) GPPvod or (b) GPPvodtemp over the whole study period (2003-2015). Shaded areas indicate the standard deviation over the aggregated grid cells. The region is located in Europe, 5 to 15°E and 46 to 51°N, and was selected as an example where the correlation analysis between GPP residuals and SPEI largely yield no significant correlations. 8-daily data were smoothed to aid visual comparison."

– line 239; "holds true" is too strong; here you only find no contradiction on a specific point; mathematically it is not right at all to say that the hypothesis is validated. It is only one indication you go the right way …

Response: We rephrased it.

Lines 38-39: "Despite the sensitivity of VOD to vegetation water content, the relationship between VOD and GPP has not yet been analyzed with regard to how the relationship responds to varying conditions of dry- or wetness."
Lines 281-284: "Given that correlations between GPPvodtemp and GPPfluxcom or GPPmodis are high in these regions, this demonstrates that GPPvodtemp shows a similar behavior as GPPfluxcom or GPPmodis in response to variations in dry or wet conditions. This finding thus provides a strong indication that the VOD-GPP-relationship generally remains similar under varying conditions of water availability."
Lines 457-459: "The analysis of the VOD-GPP residuals revealed that GPPvodtemp largely yields a similar behavior as GPPfluxcom and GPPmodis with respect to SPEI. This highlights that the relationship between VOD and GPP generally may be valid even under varying conditions of water availability."

This work presents a model to estimate Gross Primary Production (GPP) globally from a carbon sink driven approach. In particular, the paper aims at improving previous modelling of GPP as a function of the vegetation optical depth (VOD; Teubner et al., 2019) by including the effect of temperature on the autotrophic respiration. Authors explain that the model is based on the fact that VOD is a good proxy of above-ground biomass (AGB). The link between residuals of the model and the drought index SPEI is also analysed. The results presented show an improvement of model performance in terms of temporal dynamics, especially in non-tropical regions. Interestingly, results also report that the presented model does not require complementary information from precipitation or drought indicators.

Despite that the results presented are consistent with previous works and show interesting contribution to GPP modelling, I have important concerns that have to be addressed before publication. The most relevant are related to the lack of penetration through the vegetation canopy of the X-band VOD (which is not a good proxy of biomass if compared to other frequency bands), and to the need of further explaining the modelling framework both in the introduction and the methods sections. These comments and other major and minor proposals for improving the paper are detailed hereafter:

Response: We thank David Chaparro for his detailed review of the manuscript and are happy to address his comments.

**Major comments**

1. Although the paper can be well understood if the reader knows previous literature on this topic published by the authors, it is necessary that the modelling approach (i.e., main ideas and equations from previous works) and the implementation in the current paper are explained in more detail. In particular:

a. I suggest that first paragraphs of the introduction provide a more detailed description of the framework explained in Teubner et al. (2019), including if necessary some equations (e.g., equations 4 to 6 in Teubner et al., 2019).

Response: Thank you for this suggestion. We added this information in section 2.4 of the revised manuscript.

Lines 149-162: "The approach of estimating GPP based on microwave radiation and the corresponding equations are described in detail in Teubner et al. (2019). In short, the VOD-GPP model uses VOD as a proxy of above-ground living biomass (Equation 1). It determines GPP by estimating sinks for carbohydrates, i.e. the sum of NPP and Ra, which are represented through different VOD-derived variables: 1) time series of the bulk VOD signal ($VOD$; 8-daily aggregated native VOD time series), 2) time series of the temporal change in VOD ($\Delta VOD$; $\Delta VOD_t = VOD_t - VOD_{t-1}$ computed from the smoothed 8-daily aggregated VOD time series) and 3) the grid cell median of VOD ($mdnVOD$; calculated over the entire VOD time series of the grid cell; used as a proxy for vegetation cover). While NPP is related to $\Delta VOD$, Ra is related to both $VOD$ and $\Delta VOD$ using the concept proposed by Ryan et al. (1997) of dividing Ra into maintenance and growth respiration (Equation 2). By assuming that belowground biomass terms are proportional to above-ground biomass (i.e. biomass $B$ can be expressed through

above ground biomass *AGB*) and by adding a static term *c* supporting the conversion in Equation 2, GPP can be represented through a differential equation with VOD as input (Equation 3).

$$AGB = f(VOD) = \widehat{VOD} \tag{1}$$

$$GPP = NPP + Ra = \left(\frac{dB}{dt} + loss\ terms\right) + \left(a_0\frac{dB}{dt} + b_0 B\right) \sim a\frac{dB}{dt} + b\,B \tag{2}$$

$$GPP = a\frac{d\widehat{VOD}}{dt} + b\,\widehat{VOD} + c \tag{3}$$

"

b. Please, provide more detail on how you are computing and including into the model the different variables (Section 2.2). For instance, does the term "VOD" refer to VOD time-series? If so, how is the time-domain processed (raw data, smoothing, etc…)?

How is the variable computed?

In summary, please extend the text to provide enough information for readers that do not know your previous work.

Response: The term VOD refers to VOD time series. The VOD data were resampled to 8-daily values using the mean over 8 days and then used as input to the model, as stated in section 2.3. The temporal resolution of 8-daily values was chosen to match the resolution of FLUXCOM and MODIS. We included this information in data & methods section and used italic font for model input variables to make this clearer.

Lines 144-147: "These 8-daily values were then used as input to the VOD-GPP model and for further analysis throughout the study. GPPfluxcom and GPPmodis were aggregated to 0.25° to match the spatial sampling of VODCAX. For the comparison with SPEI, 8-daily GPP estimates were further resampled to monthly resolution while SPEI was spatially resampled to 0.25° using the nearest neighbour method."

Lines 150-155: "It determines GPP by estimating sinks for carbohydrates, i.e. the sum of NPP and Ra, which are represented through different VOD-derived variables: 1) time series of the bulk VOD signal (*VOD*; 8-daily aggregated native VOD time series), 2) time series of the temporal change in VOD ($\Delta VOD$; $\Delta VOD_t = VOD_t - VOD_{t-1}$ computed from the smoothed 8-daily aggregated VOD time series) and 3) the grid cell median of VOD (*mdnVOD*; calculated over the entire VOD time series of the grid cell; used as a proxy for vegetation cover)."

Lines 199-200: "For generating the smoothed time series in the calculation of $\Delta VOD$ and for aiding visual comparison in time series plots, we applied a Savitzky-Golay filter with window size of 11 data points."

2. The basis of the work is the fact that VOD is a good proxy of AGB. Nevertheless, it is very important to note that X-band VOD (hereafter XVOD) has poor capacity to penetrate the vegetation canopy, and therefore it is very limited to accurately track AGB in regions with dense vegetation. While the AGB - L-band VOD relationship shows low saturation in tropical regions, X-band is not the most appropriate frequency to be used in these areas as a proxy of biomass (e.g., Brandt et al., 2018; Rodríguez-Fernández et al., 2018; Chaparro et al., 2019). Actually, even in low carbon density areas, XVOD is more representative of vegetation cover than it is for biomass, while lower frequencies (L-band VOD; hereafter LVOD) still have improved capacity to track AGB in these areas (e.g., see Fig. 9 in Chaparro et al., 2019).

Response: The reviewer is right: VOD from L-band has been demonstrated to be a suitable proxy for biomass which is less prone to saturate at high biomass than VOD from X-band (e.g., Chaparro et al.,

2019; Li et al., 2021). This is well in line with theory, since L-band VOD is more sensitive to large plant parts while X-band VOD is more sensitive to small vegetation parts as leaves and twigs (Woodhouse, 2005). However, in our current study we do not estimate AGB but GPP and in a previous study we demonstrated that VOD from L-band (SMOS) yields the lowest correlation with GPP in vast areas of the world (Teubner et al., 2018).

It is very likely that this limitation explains the lack of improvement of the model in the tropical regions (Fig. 3b) and the low correlation between the model and the benchmark datasets in regions such as the Amazon (Fig. 3a).

Response: The poor agreement of our model in tropical regions may have several sources of uncertainty. First, our model can only be trained based on a few FLUXNET sites in tropical regions as most measurement sites are located in mid-latitudes. The same is also true for the FLUXCOM and MODIS GPP datasets. Second, the FLUXCOM and MODIS GPP products rely on optical satellite data which is highly affected by cloud cover in tropical regions (Zhao et al., 2005). Hence, correlation analyses with biophysical vegetation properties from optical sensors generally suffer from higher uncertainty of these datasets over tropical regions (Zhao et al., 2005).

In addition, this could also explain the saturation of the partial dependency plots (Fig. 1) at high VOD and T2M values (darkest lines in the first panel, probably representing vegetation-temperature conditions in the tropics) and at mdnVOD values above 0.4 (i.e., dense vegetation; third panel).

The application of XVOD is justified in the paper by the higher correlation between XVOD and GPP if compared to the LVOD-GPP correlation (Teubner et al., 2018). Nevertheless, it is important to note that the GPP benchmark datasets have an important contribution from visible-infrared (VIS-IR) indices (EVI, LAI, MIR, NDVI and NDWI, as stated in l. 96). I think it is expected that GPP datasets based on VIS-IR indices show a greater correlation with XVOD than with lower frequency VOD data, because both of them capture the same layer of vegetation (top of the canopy).

Response: Yes, this is true. As both VIS-IR-based indices or biophysical properties and XVOD are sensitive to the upper vegetation canopy, the layer where photosynthesis predominantly takes place, they can also well capture the temporal dynamic of GPP. This is exactly the motivation of using higher frequency microwave observations in our model. This has been further corroborated by correlation analyses carried out in Teubner et al. (2018). Despite the close agreement between optical indices and XVOD, we would like to note that our model still presents a sink-driven approach. This is demonstrated by the higher performance for the combination of VOD and ΔVOD compared to each input variable taken separately (Teubner et al., 2019).

In contrast, I would expect greater correlations with in situ GPP FluxNet data (Fig. 2a and Fig. 2b) if GPPvod and GPPvodtemp were computed using L-band VOD. Although it is not a global dataset, FluxNet in situ information is not conditioned by physical properties of remote sensing sensor measurements, so it is probably the most "independent" tool the authors have for evaluating the accuracy of the GPP estimates.

Response: In our previous analyses, LVOD consistently showed lower performance than higher microwave frequencies. At the global scale, among frequencies from L- to X-band, LVOD resulted in lowest correlations (Teubner et al., 2018). For in situ stations, a similar result, i.e. a closer agreement between GPP and XVOD than between GPP and LVOD, is obtained for the correlation of FLUXNET GPP with LVOD and XVOD (Figure A1). A classification of the results into land cover classes shows that also a fair amount of forest stations is included in this analysis (~38%). In addition to the analysis of 8-daily values, we repeated the analysis also for monthly values. This was done in order to exclude the

possibility that the low correlations for LVOD are caused by the relatively high temporal resolution of 8 days. However, a similar result as for 8-daily values is obtained.

Based on these results, we concluded that LVOD may not be a good proxy for estimating GPP and we did not include LVOD in further analyses. We added the correlation pre-analysis in the supplement and included more information about the choice of microwave frequency in the beginning of the data & methods section, the discussion and the supplement.

Lines 73-84: "2.1 Choice of microwave frequency

[revised manuscript text omitted]

For all these reasons, it would be very interesting if the authors include new GPPvod and GPPvodtemp models based on L-band VOD and validate their accuracy using in situ FluxNet data (i.e., including them in Fig. 2). They could show (and compare) the resulting GPP estimates between different frequencies and, importantly to preserve the scope of the paper, between GPPvod and GPPvodtemp models. However, I am aware that this could move the work slightly beyond its initial scope, as it adds another factor (i.e., frequencies) in the comparisons. I encourage the authors to work on this possibility, although it is up to them to finally incorporate this change or to keep only XVOD in the paper. In any case, they must discuss all the possible implications of using XVOD.

Response: Thank you for your suggestion. However, based on our above analysis results that show a poor performance of LVOD in predicting GPP, we did not add LVOD to the paper to not further complicate the paper. But we followed the suggestion of the referee and included this aspect with regard to the use of XVOD vs. LVOD data in the introduction and discussion. Please see our response above.

Within the discussion, they should address at least the following points/questions:

- It is stated that "the VOD-GPP model relies on estimating carbon sink terms, [...], based on VOD as a proxy of aboveground living biomass" (l. 35-37). To what point is this true, according to the facts that XVOD is more representative of vegetation cover than of biomass, and that lower VOD frequencies have enhanced capacities to capture biomass? Please clearly explain the possible limitations of the approach.

Response: Indeed, the use of VOD from X-band may appear counterintuitive and the use of LVOD might seem more appropriate. However, as LVOD cannot capture well the intra-annual temporal dynamics of canopies, it cannot serve as a good proxy for the high-frequency temporal dynamics of GPP. In addition, the reason for using X-band VOD (and not L-band VOD) for the estimation of GPP might be explained with differences in metabolic activity of various plant parts. Litton et al. (2007) showed that

leaves and roots may present a better proxy for GPP than large structural components with a lower metabolic activity. For further details, see discussion section 6.2 in Teubner et al. (2019). On the other hand, LVOD might be a better predictor for the annual-integrated total carbon allocation (net primary production) as this is closely related to the annual increment in biomass. However, testing this hypothesis is beyond the scope of our study.

We expanded our discussion for using X-band VOD to make this clearer. See our response above.

- Please, discuss about the saturation effects in Fig. 1 (first and third panel; see my comment above). Are they likely to be linked to XVOD saturation in the tropics? If so, which are the implications?

Response: Although we cannot rule out a saturation effect of VOD with regard to GPP, the partial dependency plots should, in this case, resemble an exponential curve and not the optimum curve that we observed. The plots rather suggest that the in situ GPP estimates are limited at around 10 gC m-2 d-1. We addressed the issue of non-linearity and possible saturation in the discussion section 4.3.

Lines 369-375: "Nonetheless, the impact of such potential saturation with biomass on the estimation of GPP is less trivial, especially with regard to densely vegetated areas like the tropics. Non-linearity in the conversion between VOD and AGB should ideally be reflected in the partial dependency plot of GAM, which was also the reason for choosing this type of modelling approach. Scatterplots of the resulting GPPvodtemp estimates did not show clear signs of saturation at high in situ GPP. The FLUXNET training data set, however, only has few stations in the tropics and thus the robustness of the model may be limited by the availability of in situ stations."

- In l. 197, it is mentioned (referring to tropical regions) that "[in these regions] sensitivity to temperature is also low, which makes the interaction term mainly controlled by VOD."

If I correctly understood plots in Fig. 1, would it be more precise to affirm that it is mostly controlled by , as other dependencies (VOD, mdnVOD) saturate in tropical regions?

Response: Our statement is referring to the lack of a strong seasonality in the tropics, which might explain why the improvement in temporal correlation is absent or lower than in temperate regions because these regions do not experience large temperature differences.

3. The GPP estimates (GPPvod and GPPvodtemp) are calibrated using FluxNet in situ data. Also, both FluxNet and FLUXCOM data (an upscaling of FluxNet) are used as reference datasets for evaluating GPP estimates. I think that, consequently, reference datasets may not be fully independent from GPP estimates. To what extent? Which is the contribution of FluxNet data in the reference datasets and in the estimates? This has to be acknowledged and possible implications discussed.

Response: Yes, the FLUXCOM product has been trained against the FLUXNET station data (Tramontana et al., 2016; Jung et al., 2020). In comparison to the FLUXCOM product, we could use much fewer station data (only Tier 1 data). Also, the MOD17 GPP product has been partly calibrated to some FLUXNET stations (Running et al., 1999). At large/global scales, there is no alternative to constrain absolute estimates of GPP than using FLUXNET data. We added this point to the discussion.

Lines 383-389: "4.4 Independence of global GPP data sets
For the comparison with VOD-based GPP estimates, we used independent global data set from FLUXCOM and MODIS. Both data sets include to some extent information from FLUXNET data. FLUXCOM has been trained against FLUXNET data (Tramontana et al., 2016; Jung et al., 2020), however, with a larger number of stations than in the freely available Tier 1 data set that was used for our model. Also, MODIS has been partly calibrated to some FLUXNET stations (Running et al., 1999). Therefore, the FLUXCOM and MODIS may not be fully independent from our VOD-based GPP

estimates. Nevertheless, there is no alternative to constrain absolute GPP estimates at global scale than by using FLUXNET data."

In addition, authors should try to guarantee at least one year of "fully independent" comparison between estimates and FluxNet/FLUXCOM data. I suggest they could calibrate the model by leaving one year of data apart (e.g., use 2004-2014 for calibration) and apply the remaining data (2003) for fully independent comparisons. They can show these new comparisons in supplementary materials and refer to them to show consistency/inconsistency with the full-period comparisons of Figures 2 to 6.

Response: Thank you for this suggestion. As a common approach (that was also used in FLUXCOM), we did perform leave-site out cross-validation. We now added these cross-validation results for GPPvod and GPPvodtemp in the supplement.

Lines 192-194: "In addition, cross validation was computed for the above metrics using the leave-site-out method, where the model performance is evaluated at each site by omitting the respective site data from model training and then using the left-out data for computing the statistics. The analysis was carried out for the full signal and the anomalies from the mean seasonal cycle."

Lines 220-223: "Cross validation results in Table A2 further confirm a higher performance of GPPvodtemp compared to GPPvod. For the full signal as well as for the anomalies from the mean cycle, correlation, ubRMSE and bias generally yield higher performance for GPPvodtemp. The increase in performance is more pronounced for the full signal than for the anomalies."

Supplement: "Table A2. Leave-site-out cross validation for GPPvodtemp and GPPvod. The analysis was conducted for the full signal as well as for the anomalies from the mean seasonal cycle. Anomalies were calculated after model application. Values represent mean and standard deviation of the metrics over the cross validation results for each site.

|  | Pearson r [-] | UbRMSE [gC m-2 d-1] | Bias [gC m-2 d-1] |
|---|---|---|---|
| GPPvod | 0.40 ± 0.32 | 2.57 ± 1.14 | -0.04 ± 2.01 |
| GPPvodtemp | 0.54 ± 0.31 | 2.30 ± 1.01 | -0.08 ± 2.01 |
| GPPvod anomalies | 0.18 ± 0.22 | 1.57 ± 0.78 | -0.00 ± 0.00 |
| GPPvodtemp anomalies | 0.22 ± 0.19 | 1.53 ± 0.76 | 0.00 ± 0.00 |

"

**Minor comments**

- Lines 5-6: "VOD-GPP model generally showed good agreement" → Please quantify (e.g. correlation coefficient).

Response: A quantification of the previous correlation coefficient would imply that the results can be directly compared. However, different aspects of the model have changed. In the current study, we are using a merged VOD (VODCA X-band), while we previously only considered single sensor VOD (AMSRE or AMSR2 X-band). In addition, we analyzed a longer period, which also makes the training dataset different from the previous studies. Therefore, we would not add information of correlation coefficients from previous studies in the abstract.

For the direct comparison of previous (GPPvod) and current (GPPvodtemp) model results under similar conditions, we would like to refer to our results presented in section 3.3.

- L. 6: "tended to overestimate" → By how much? Please quantify.

Response: The overestimation was not quantified in the previous study; therefore, we cannot provide a specific number here.

- L. 13: "Our results reveal an improvement" → Please quantify this improvement (e.g., increase on the average correlation).

Response: Thank you for this suggestion, we added the increase in correlation coefficient from the results section also in the abstract.

Lines 13-14: "Our results reveal an improvement in model performance for correlation when including the temperature dependency of autotrophic respiration (average correlation increase of 0.18)."

- L. 14: "This increase in temporal dynamic" → "This improvement in temporal dynamics."

Response: We revised this.

Lines 14-15: "This improvement in temporal dynamic is larger for temperate and cold regions than for the tropics."

- L. 19: can you mention which are these regions?

Response: We added this information.

Lines 22-24: "Exceptions from this rule were found in some regions: significant negative correlations between VOD-GPP residuals and SPEI were observed in the US corn belt, Argentina, Eastern Europe, Russia and China, while significant positive correlations were obtained in South America, Africa and Australia."

- L. 19: "between […] with" → "between […] and"

Response: We revised this.

Lines 22-24: "Exceptions from this rule were found in some regions: significant negative correlations between VOD-GPP residuals and SPEI were observed in the US corn belt, Argentina, Eastern Europe, Russia and China, while significant positive correlations were obtained in South America, Africa and Australia."

- L. 25: provide → provides

Response: We revised this.

Lines 30-31: "Vegetation optical depth (VOD) from microwave satellite observations provides the opportunity for studying large-scale vegetation dynamics due to its sensitivity to the vegetation water content and above-ground biomass."

- L. 25 to 30: you may want to include other references which are explicitly linked to water content: e.g., Feldman et al., 2018; Tian et al., 2018.

Response: Thank you for this suggestion, we added these references.

Lines 31-36: "Different studies have employed VOD for deriving various plant properties or vegetation characteristics that can be related to the plant's water content, including biomass estimation (Liu et al., 2015; Brandt et al., 2018; Rodríguez-Fernández et al., 2018; Chaparro et al., 2019; Fan et al., 2019; Frappart et al., 2020; Wigneron et al., 2020; Li et al., 2021), crop yield (Chaparro et al., 2018), tree mortality (Rao et al., 2019; Sapes et al., 2019), analysis of burned area (Forkel et al., 2019), ecosystemscale isohydricity (Konings and Gentine, 2017), plant water uptake during dry downs (Feldman et al., 2018) and plant water storage (Tian et al., 2018)."

- L. 28: Chaparro et al., 2019→Chaparro et al., 2018 (this is different from the "carbonstocks work" in Chaparro et al., 2019). Add the new reference to the references list if you want to keep it in the text.

Response: Thank you, we revised it.

Lines 31-36: "Different studies have employed VOD for deriving various plant properties or vegetation characteristics that can be related to the plant's water content, including biomass estimation (Liu et al., 2015; Brandt et al., 2018; Rodríguez-Fernández et al., 2018; Chaparro et al., 2019; Fan et al., 2019; Frappart et al., 2020; Wigneron et al., 2020; Li et al., 2021), crop yield (Chaparro et al., 2018), tree mortality (Rao et al., 2019; Sapes et al., 2019), analysis of burned area (Forkel et al., 2019), ecosystem-scale isohydricity (Konings and Gentine, 2017), plant water uptake during dry downs (Feldman et al., 2018) and plant water storage (Tian et al., 2018)."

- L. 70: maybe saying "only a few years" is a bit excessive (e.g., SMOS spans >10 years). Try using another expression, please.

Response: It was meant it in the sense that the use of single sensor VOD may hamper the comparison of longer time periods. We rephrased it.

Line 91: "VOD retrievals from single sensors often span only a certain period in time, which may hamper the analysis of longer periods."

- L. 83-85: "During data processing…" → Please move these lines to the methods section.

Response: Actually, this sentence is in the right place because it belongs to the description of the VODCA dataset. We modified the sentence to make this clearer.

Lines 101-103: "During the processing of VODCAX, data are masked for radio frequency interference (RFI) (Moesinger et al., 2020) since RFI can introduce spurious retrievals (Li et al., 2004; Njoku et al., 2005)."

- L. 118: "T2M was used in our analysis, since this parameter is most common for describing the temperature dependency" → Please add some references to show the common use of this variable.

Response: We added references regarding the use of T2M for describing the temperature dependency of autotrophic respiration.

Lines 138-140: "*T2M* was used in our analysis, since this parameter is most common for describing the temperature dependency of autotrophic respiration for aboveground vegetation (e.g., Ryan et al., 1997; Running et al., 2000; Ceschia et al., 2002; Drake et al., 2016)."

- L. 121-122: "aggregated to 8-daily estimates" → Please, specify that this is done to match MODIS time-steps in case it was your intention.

Response: We added this information.

Lines 142-143: "VODCAX data were masked for low temperature (skin temperature < 0°C) and snow cover (snow depth > 0cm) and then aggregated to 8-daily estimates by computing the mean over 8 days to match the temporal resolution of GPPmodis and GPPfluxcom."

- L. 162: "savitzky-golay" → "Savitzky-Golay."

Response: We revised this.

Lines 199-200: "For generating the smoothed time series in the calculation of *ΔVOD* and for aiding visual comparison in time series plots, we applied a Savitzky-Golay filter with window size of 11 data points."

- L. 170: "are consistent the previous" → "Are consistent with the previous."

Response: We revised it.

Lines 207-209: "The partial dependencies for *ΔVOD* and *mdnVOD* (Figure 1b,c) are consistent with the previous model and yield a positive relationship with GPP for *ΔVOD* in the middle part of the value range and a general decreasing relationship for *mdnVOD*."

- Figure 1: Please add marginal distribution rug plots to each panel. Name the panels as "a" to "d" and complete the figure caption with explanation of each panel.

Response: Thank you for this suggestion. We added rug plots and included the more information about each panel in the caption. Panel names were already included in the bottom of each panel, but we moved them to the top and made the legend smaller to increase visibility.

"

[Figure]

Figure 1. Partial dependency plot for GPPvodtemp for each input variable: (a) *VOD*, (b) *ΔVOD*, (c) *mdnVOD* and (d) *T2M*. The model was trained with data from the period 2003-2014. Dashed lines in (b) and (c) denote the 95% confidence interval. The interaction between *VOD* and *T2M* (a,d), which represents a 3D surface, is displayed as projection on the 2D plane for each of the two input variables. For this, the parameter space was divided into 10 equally spaced bins between minimum and maximum of the respective variable. The bin edges are displayed as colored lines as indicated in the legend."

- Figure 2: add GPPvod as another panel for comparison with GPPvodtemp and benchmark data. Also, I find that the blue to yellow colorbar shows very low contrasts. To me, it is difficult to appreciate color gradients in the figure. You could use other colors (e.g., blue to red?) or saturate the colorbar at the (e.g., 95th, 99th) percentile to improve contrast.

Response: We added GPPvod and adjusted the color bar.

Lines 217-220: "At FLUXNET in situ stations, global GPP datasets overall show similar results (Figure 2). GPPvod exhibits a slight accumulation of GPP values at around 4 g C m-2 d-1, while the density for GPPvodtemp is relatively smooth and comparable to GPPfluxcom and GPPmodis. Both GPPvod and GPPvodtemp show a relatively high number of non-zero GPP at around zero GPPfluxnet, which is less pronounced for GPPvodtemp than for GPPvod."

"

[Figure]

Figure 2. Scatter plots of 8-daily in situ GPPfluxnet versus global GPP data sets (a) GPPvodtemp, (b) GPPvod, (c) GPPfluxcom and (d) GPPmodis for the period 2003-2014."

- L. 218: "For a region in Europe" → Please add coordinates also here, as well as the general situation (e.g., "Central Europe") to help the reader.

Response: We included the information about the location given in the caption of Figure 5 also in the text and added the location to Figure A3.

Lines 262-264: "For a region in Europe (5 to 15°E and 46 to 51°N), where we generally did observe an increase in all three performance metrics, we find that for GPPvod mainly winter time estimates of GPP are too high compared to GPPfluxcom and GPPmodis (Figure 5)."

Supplement: "

[Figure]

Figure A3. Location of FLUXNET Tier1 v1 stations within the period 2003 to 2014. The size of the circles represents the number of available years for each station. The blue rectangle denotes the location of the region in Europe used Figure 5."

- L. 218: "increase" → "increase in".

Response: We revised this.

Lines 262-264: "For a region in Europe (5 to 15°E and 46 to 51°N), where we generally did observe an increase in all three performance metrics, we find that for GPPvod mainly winter time estimates of GPP are too high compared to GPPfluxcom and GPPmodis (Figure 5)."

- L. 227-229: there is no a verb in this sentence, please rephrase.

Response: We revised this.

Lines 273-275: "Although differences exist between all data sets, key features are observed among all data sets, such as the positive anomalies at -55°N in 2003, at -30°N in 2011 or at +75°N in 2012 and the negative anomalies at +75°N in 2003 and 2015 and at around -40° in 2009 and 2011."

- L. 233: "between […] with" → "between […] and".

Response: We revised this

Lines 278-279: "For the correlation of the residuals between standardized GPP (GPPvodtemp-GPPfluxcom or GPPvodtemp-GPPmodis) and SPEI, we find that large areas show no significant correlation with SPEI03 (Figure 7a,b)."

- Figures 4 and 6: please define in the text what are "zonal means".

Response: We included this information in the captions of Figures 4 and 6.

Caption Figure 4: "Zonal mean of annual GPP for GPPfluxcom, GPPmodis, GPPvodtemp and GPPvod for the study period 2003-2015. To obtain zonal means, data were averaged over all grid points of the same latitude."

Caption Figure 6: "Hovmöller diagramm for zonal means of annual GPP anomalies (z-scores) for (a) GPPvodtemp, (b) GPPfluxcom and (c) GPPmodis over the study period. Zonal means were calculated by averaging data over all grid points of the same latitude."

- Figures 5 and 8: please, could you make each panel wider? Then there is more place for seeing interannual variability in the figures.

Response: We updated the figure layout.

- Figure 6: Add GPPvod as another panel for comparison with GPPvodtemp and benchmark data.

Response: Although this would be possible in general, we would rather keep the main part as it is. The reason for this is that we narrowed the two setups down to one setup (i.e. GPPvodtemp) based on the results in the preceding section. But in order to provide the information, we put the figure for GPPvod in the supplement.

Supplement: "

[Figure]

Figure A6. Hovmöller diagramm for zonal means of annual GPP anomalies (z-scores) for GPPvod over the study period 2003-2015."

- L. 236: "given that correlations in these regions are high" → Authors probably mean correlations between GPPvodtemp and benchmark data. Please specify this, because as you explained correlations of residuals in the previous sentence, it can be confusing.

Response: We clarified this.

Lines 281-284: "Given that correlations between GPPvodtemp and GPPfluxcom or GPPmodis are high in these regions, this demonstrates that GPPvodtemp shows a similar behavior as GPPfluxcom or GPPmodis in response to variations in dry or wet conditions. This finding thus provides a strong indication that the VOD-GPP-relationship generally remains similar under varying conditions of water availability."

- Figure 7: to improve the boxes for regions, you could use colors different than those from the colorbar (e.g., light green instead of blue?).

Response: We adjusted the color. Please note, that we tried different colors, also light green. However, light green may be problematic for color blind people to distinguish from the red areas and, in addition, it did not give enough discrimination with respect to the graph's colorbar when converting the figure to grey scale.

"

[Figure]

Figure 7. Correlation between residuals of standardized GPP (GPPvodtemp-GPPfluxcom and GPPvodtemp-GPPmodis) and SPEI. None significant correlations are indicated in grey. (a,c): GPPvodtemp-GPPfluxcom, (b,d): GPPvodtemp-GPPmodis, (a,b): SPEI03 (short-term response), (c,d): SPEI12 (long-term response). Regions A-D: US cornbelt (A), Argentina (B), Eastern Africa (C) and Eastern Australia (D). The analysis is based on the whole study period (2003-2015)."

- L. 282: "increase" → Do you mean "improvement"?

Response: We revised it.

Lines 332-333: "Our results showed that the improvement in temporal dynamic was mainly observed for temperate and cold regions."

- L. 338: "between with" → "with".

Response: We revised this.

Lines 412-413: "The analysis of VOD-GPP residuals with respect to FLUXCOM and MODIS revealed that GPPvodtemp largely showed a similar behavior as the independent GPP data sets as demonstrated by the widespread none significant correlations with SPEI."

- Figure A1: please add GPPvod as well as a map of the median differences between GPPvod and GPPvodtemp. Also, note that the contrast in the blue to yellow colorbar could be improved, or colors changed to a blue-red scale.

Response: We added GPPvod and a difference map between GPPvod and GPPvodtemp.

Supplement: "

[Figure]

Figure A2. Temporal median maps for (a) GPPvodtemp, (b) GPPfluxcom, (c) GPPvod, (d) GPPmodis and (e) difference between the median maps of GPPvodtemp and GPPvod. For GPPvodtemp and GPPvod, areas where both GPPfluxcom and GPPmodis are missing were masked, since these data were not used during the analysis. Data were computed over the whole study period (2003-2015)."

- Figure A3: please add GPPvod.

Response: Consistent with the 8-daily scatter plots, we added GPPvod to Figure A4.

Supplement: "

[Figure]

Figure A4. Scatterplot of annual GPP for GPPfluxnet versus (a) GPPvodtemp, (b) GPPvod, (c)

GPPfluxcom and (d) GPPmodis. Annual values were calculated from 8-daily GPP for each data set and cover the FLUXNET period 2003-2014."

- Figure A4: this seems an interesting result, but I do not fully understand what do you mean by "scaled latitudinal" distribution. Could you please explain this?

Response: The scaled latitudinal distribution is obtained by dividing the data by the maximum of the latitudinal distribution. We rephrased it to make it clearer.

Caption Figure A5: "Scaled latitudinal distribution of annual GPP for GPPvodtemp, GPPvod, GPPfluxcom and GPPmodis for the study period 2003-2015. Data are scaled by dividing the latitudinal distribution by the maximum of the latitudinal distribution for each data set."

- Table A1: it could be useful to detail the dominant land cover in each station. Then, the reader will be able to see which vegetation types have been included in calibration of GPP estimates.

Response: We included the information about land cover in the table.

[revised manuscript text omitted]

Woodhouse, I. H. (2017). Introduction to microwave remote sensing. CRC press.

Zhao, M., Heinsch, F. A., Nemani, R. R., & Running, S. W. (2005). Improvements of the MODIS terrestrial gross and net primary production global data set. Remote sensing of Environment, 95(2), 164-176.

**bg-2020-413-referee-report-2**

**David Chaparro (Referee)**

The reviewed manuscript has been clearly improved from the previous version, and authors have addressed all the comments with a very thorough revision. Both the introduction and methods sections have been completed with further details on the previous literature which is the basis of the work. The fact of using X-band VOD instead of L-band VOD has been explained and discussed consciously. The manuscript will be ready for publication after addressing some minor comments that may provide a more complete understanding and interpretation of results, as well as after checking other small (text and spelling) issues:

Response: Dear David Chaparro, thank you for this further review. We are happy to include the suggested changes and we updated the manuscript accordingly.

- Discussion on X-band and L-band is appropriate and very thorough. Figure A1 supports it very well. A last point for improving Figure A1 would be to provide the same analysis only for forests (i.e., a third panel in Fig. A1, similar to panel *a*, only containing forest stations).

  Response: We added the analysis of only forest stations.
  "

[Figure]

  Figure A1. Pre-analysis of correlation between in situ FLUXNET GPP and single sensor VOD from L- and X-band. (a): Pearson correlation between FLUXNET GPP (mean of GPP_DT_VUT_REF and GPP_NT_VUT_REF) and L-band VOD (SMOS VOD-L, 7/2010–12/2014) and X-band VOD (AMSR-E VOD-X, 1/2007–9/2011). Data were resampled to 8-daily or monthly values. The analysis was conducted only for stations where both of the VOD data set are available (47 stations). For details about the VOD datasets and their data processing, see Teubner et al. (2018). (b): As in (a) but for the subset of forest land cover classes (ENF, DBF, EBF and MF). (c): Composition of IGBP land cover classes for the stations used in this pre-analysis. Abbreviations: GRA (Grasslands), CRO (Croplands), ENF (Evergreen Needleleaf Forests), DBF (Deciduous Broadleaf Forests), EBF (Evergreen Broadleaf Forests), SAV (Savannas), MF (Mixed Forests), WET (Permanent Wetlands), WSA (Woody Savannas) and OSH (Open Shrublands)."

- In L. 262 you comment that *"For a region in Europe (5 to 15°E and 46 to 51°N), where we generally did observe an increase in all three performance metrics, we find that for GPPvod mainly winter time estimates of GPP are too high compared to GPPfluxcom and GPPmodis (Figure 5). By adding temperature as input to the model, winter observations are markedly dampened and summer observations are only slightly increased."* This is a very nice result to show, nevertheless it refers only to one region. It will provide more consistency to your results if you include two or three similar regions and see if the winter observations also dampen when adding temperature to the model. This can be included in the supplement.

  Response: We added time series plots for the two regions A and B from our analysis and included this graph in place of previous Figure A5.

  Lines 265-266: "A similar behavior is observed for other temperate regions (Figure A5)."

"

[Figure]

Figure A5. Time series plot of spatially aggregated GPP estimates for GPPfluxcom, GPPmodis and (a,c) GPPvod or (b,d) GPPvodtemp for the two regions US cornbelt (a,b; region A) and Argentina (c,d; region B) from Figures 7, 8 and A7. The analysis is based on the study period 2003-2015. Shaded areas represent the standard deviation over the aggregated grid cells. 8-daily data were smoothed to aid visual comparison."

- Figure A5: this figure provides an interesting conclusion, as you mention in L. 260. In my opinion, this is relevant enough to be moved to the main part of the paper. I suggest to show it as a second panel in Figure 4. Also, in the discussion, you can emphasize the fact that adding temperature improves the relative distribution of GPP.

Response: We moved previous Figure A5 to Figure 4 as a second panel.
"

[Figure]

Figure 4. Zonal mean of annual GPP for GPPfluxcom, GPPmodis, GPPvodtemp and GPPvod for the study period 2003-2015. (a): Absolute latitudinal distribution. (b): Scaled latitudinal distribution. To obtain zonal means, data were averaged over all grid points of the same latitude. Scaled data were computed by dividing the latitudinal distribution by the maximum of the latitudinal distribution for each data set."

- Section 4.4: please, add a sentence to highlight that the leave-site-out cross validation shown in Table A2 confirms that, although the benchmark GPP datasets are not fully independent, this does not impact the results.
  Response: We added the link to the cross validation results.
  Lines 389-391: "In addition, the agreement between GPP and VOD-based GPP estimates was also confirmed at site level using leave-site-out cross validation. Since this analysis is independent from the comparison with global data sets, it supports the use of VOD for deriving GPP."

Other minor comments are:
- L. 75: "have been demonstrated" → "has been demonstrated".
  Response: We revised this.
  Lines 75-77: "On the one hand, VOD from low microwave frequencies like L-band has been demonstrated to be better suited as proxy for mapping total above-ground biomass than high frequency VOD, i.e. X-band VOD, as L-band VOD saturates less at high biomass values (Chaparro et al., 2019; Frappart et al., 2020; Li et al., 2021)."

- L. 79-81: in these lines you describe Figure A1, but data needed for this figure have not been described before. At least you should include a sentence explaining in which sections the reader can find a description of the data.
  Response: We added this information from the figure caption also in the text.
  Lines 78-80: "In Figure A1 we further corroborated this observation by a correlation analysis between in situ GPP and VOD from L- and X-band, respectively (for details about the single sensor VOD datasets, see Teubner et al., 2018)."

- L. 93-98: these sentences are redundant. Can you join frequencies (l. 93-94) with time overpasses (l. 95-97) to avoid writing the satellite names twice?
  Response: We joined the two sentences.
  Lines 93-98: "VODCA (Moesinger et al., 2020) X-band (VODCAX) contains nighttime observations of passive VOD derived from TMI (10.7 GHz; variable overpass time), AMSR-E (10.7 GHz; descending 1:30 am),WindSat (10.7 GHz; descending 6:00 am) and AMSR2 (10.7 GHz; descending 1:30 am). The VOD input data are obtained from the Land Parameter Retrieval Model (LPRM; van der Schalie et al., 2017)."

- L. 177: "pyGAM" → "The pyGAM".
  Response: We updated this.
  Lines 176-177: "The pyGAM (Servén and Brummitt, 2018) version 0.8.0 provides the possibility of adding an interaction term."

- L. 272: "GPP data set" → "GPP datasets"
  Response: We revised this.
  Line 272: "The latitudinal distribution of annual GPP anomalies reveals a general agreement between the GPP datasets (Figures 6 and A6)."

- L. 380: "sensitive" → "sensitive to"
  Response: We updated this.
  Lines 380-382: "Since high frequency VOD is more sensitive to small plant parts like leaves and twigs (Woodhouse, 2017), this could be an explanation why X-band VOD might be better suited for the estimation of GPP and why saturation at high total above-ground biomass may be less of an issue here."